# High CDC20 levels increase sensitivity of cancer cells to MPS1 inhibitors

Siqi Zheng[1,6], Linoy Raz[2,6], Lin Zhou[1,6], Yael Cohen-Sharir[2,7], Ruifang Tian [iD][1,7], Marica Rosaria Ippolito[3], Sara Gianotti [iD][3,4], Ron Saad[2], Rene Wardenaar [iD][1,5], Mathilde Broekhuis[5], Maria Suarez Peredo Rodriguez[1], Soraya Wobben [iD][1], Anouk van den Brink [iD][1], Petra Bakker[1], Stefano Santaguida [iD][3,4], Floris Foijer [iD][1,5,8 ✉] & Uri Ben-David [iD][2,8 ✉]

## Abstract

**Spindle assembly checkpoint (SAC) inhibitors are a recently developed class of drugs, which perturb chromosome segregation during cell division, induce chromosomal instability (CIN), and eventually lead to cell death. The molecular features that determine cellular sensitivity to these drugs are not fully understood. We recently reported that aneuploid cancer cells are preferentially sensitive to SAC inhibition. Here we report that sensitivity to SAC inhibition by MPS1 inhibitors is largely driven by the expression of CDC20, a main mitotic activator of the anaphase-promoting complex (APC/C), and that the effect of CDC20 is larger than that of the APC/C itself. Mechanistically, we discovered that CDC20 depletion prolongs metaphase duration, diminishes mitotic errors, and reduces sensitivity to SAC inhibition. We found that aneuploid cells express higher basal levels of CDC20, which shortens the duration of metaphase and leads to multiple mitotic errors, resulting in increased long-term sensitivity to the additional CIN induced by SAC inhibition. Our findings propose high CDC20 expression as a molecular feature associated with the sensitivity to SAC inhibition therapy and as a potential aneuploidy-induced cellular vulnerability.**

**Keywords** Spindle Assembly Checkpoint (SAC); Cancer; Aneuploidy; Chromosomal Instability; Cell Cycle
**Subject Categories** Cancer; Cell Cycle

## Introduction

The spindle assembly checkpoint (SAC), also known as the mitotic checkpoint, is a key cell cycle regulator that ensures the fidelity of chromosome segregation during mitosis. In the presence of unattached kinetochores, which can lead to errors in chromosome segregation, the SAC is activated and arrests cell division, allowing time for error correction and restoration of normal division. The SAC is composed of a series of proteins that recruit an effector complex, the mitotic checkpoint complex (MCC). The MCC, which is composed of BUB1B, MAD2L1 and BUB3, prevents progression into anaphase by sequestering CDC20, the main activator of the anaphase-promoting complex (APC/C) (Curtis et al, 2020; Kops et al, 2005). When unbound by the MCC, CDC20 forms a complex with the E3 ubiquitin ligase APC/C, and the complex tags various substrates for degradation by ubiquitination. Notable CDC20-APC/C substrates are Securin, whose degradation allows sister chromatid separation; Cyclin B1, whose degradation also promotes mitosis culmination; and CDC20 itself, which is negatively regulated by the complex (Musacchio and Salmon, 2007). However, when the SAC is active, the MCC binds CDC20 and progression into anaphase is blocked, allowing time for error correction and restoration of normal division (Musacchio and Salmon, 2007). Nonetheless, continued cell cycle arrest or failure to divide normally will eventually lead either to cell death or to cell division despite the presence of an altered spindle and an active SAC (Sinha et al, 2019; Rossio et al, 2010), which would lead to chromosome missegregation and aneuploidization. Recent work has shown that the MCC can bind and inhibit a second molecule of CDC20 that has already bound and activated the APC/C (Izawa and Pines, 2015; Yamaguchi et al, 2016), pointing to additional complexity that extends the described mechanism.

MPS1 inhibitors are drugs that inhibit the SAC, leading to faulty mitoses and causing erroneous chromosome segregation. Consequently, these drugs induce chromosomal instability and lead to the acquisition of aberrant and unfit karyotypes in the treated cells (He et al, 2018; Mason et al, 2017; Kawakami et al, 2019). We recently showed that SAC inhibition is more detrimental for aneuploid cells than for diploid cells (Cohen-Sharir et al, 2021). SAC inhibitor drugs—specifically, MPS1 inhibitors—are currently in multiple

[1]European Research Institute for the Biology of Ageing, University of Groningen, 1, Antonius Deusinglaan, 9713 AV Groningen, The Netherlands. [2]Department of Human Molecular Genetics and Biochemistry, Faculty of Medicine, Tel Aviv University, Tel Aviv, Israel. [3]Department of Experimental Oncology, IEO European Institute of Oncology IRCCS, Milan, Italy. [4]Department of Oncology and Hemato-Oncology, University of Milan, Milan 20141, Italy. [5]Functional Genomics Center, University of Groningen, 1, Antonius Deusinglaan, 9713 AV Groningen, The Netherlands. [6]These authors contributed equally as first authors: Siqi Zheng, Linoy Raz, Lin Zhou. [7]These authors contributed equally as second authors: Yael Cohen-Sharir, Ruifang Tian. [8]These authors contributed equally as senior authors: Floris Foijer, Uri Ben-David. ✉E-mail: f.foijer@umcg.nl; ubendavid@tauex.tau.ac.il

clinical trials for the treatment of solid cancers (NCT02792465, NCT03568422, NCT05251714), either alone or in combination with microtubule-disrupting agents such as paclitaxel. Despite the emerging clinical utility of these drugs, no biomarker predicting patient response has yet been confirmed, and the molecular mechanism behind the response to these drugs is only partially understood. Several papers suggested that low expression levels or dysfunction of the APC/C are associated with resistance to SAC inhibition (Sansregret et al, 2017; Thu et al, 2018; Wild et al, 2016). However, in this work we refine the proposed mechanism underlying the response to SAC inhibition, suggesting that CDC20 is a major regulator of this response. We further propose that CDC20 underlies the differential sensitivity of aneuploid cells to these drugs.

# Results

## Cdc20 loss is associated with resistance to SAC inhibition

To identify genes and pathways that are involved in the response to SAC inhibition, we performed two independent CRISPR-Cas9 screens in 3T3 mouse fibroblasts, looking for genes whose depletion promoted cell proliferation under SAC inhibition, as shown in Figs. 1A and EV1A,B. Comprehensive lists of the resultant candidate genes can be found in Dataset EV1. Searching for cellular pathways enriched in the top ranking 10% of candidate genes from both CRISPR screens, we found that three out of ten pathways were directly related to anaphase promotion by the APC/C, and a fourth pathway contained the APC/C complex genes among other nuclear ubiquitin-ligases (Fig. 1B). Interestingly, the top significant gene whose loss conferred resistance to SAC inhibition in both screens was Cdc20, the protein co-activator of the APC/C during mitosis (Fig. 1C). While the core members of the APC/C were among the candidate genes, their effect was much weaker than Cdc20's in both of the screens (Fig. EV1C). To validate the role of Cdc20 levels in promoting resistance to SAC inhibition, we knocked down Cdc20 in 3T3 cells using shRNA (Fig. EV1D,E), and assessed sensitivity to the SAC inhibitor Reversine (Santaguida et al, 2010). Indeed, Cdc20 depletion by shRNA reduced sensitivity to SAC inhibition evidenced by increased cell viability (Fig. 1D,E). Together, these findings suggest a strong association between Cdc20 expression and sensitivity to SAC inhibition, with Cdc20 depletion leading to increased resistance to these drugs.

## CDC20 expression is a major predictor of the sensitivity to genetic and chemical inhibition of the SAC

Several studies suggested that resistance to SAC inhibition is associated with lower expression levels of the anaphase-promoting complex (APC/C) (Sansregret et al, 2017; Thu et al, 2018; Wild et al, 2016). While some of them included CDC20 within a broader APC/C signature, none have addressed its specific role in activating the APC/C and possibly driving the response. However, our CRISPR screen suggested that Cdc20 has a key role in determining the sensitivity to SAC inhibition, more than any APC/C core protein.

To further explore the respective roles of CDC20 and the APC/C in response to SAC inhibition, we analyzed genomic and transcriptomic datasets from over 1700 human cancer cell lines from the cancer Dependency Map (Tsherniak et al, 2017). We used an APC/C transcriptional signature previously reported to be associated with sensitivity to SAC inhibition (Thu et al, 2018) (Table EV1) to assign an APC/C expression score to each cell line and correlated the APC/C expression with sensitivity to genetic perturbation of the core SAC components MAD2L1 and BUB1B. Indeed, a lower APC/C expression score was correlated with lower sensitivity to genetic SAC perturbation (Fig. 2A,B, left panels). However, we noticed that the gene signature used by Thu et al to define the APC/C contained not only the core APC/C subunits but also CDC20, which acts as the main APC/C co-activator only during mitosis. Intriguingly, when we examined CDC20 and the APC/C subunits (Yamano, 2019) separately, we found that the predictive value of the signature could be largely attributed to the expression of CDC20 itself (Table EV1 and Fig. 2A,B, middle and right panels). Next, we used the human cell line data to study the association between CDC20 gene expression and sensitivity to chemical perturbation of the SAC. To this end, we compared the sensitivity to SAC inhibitor drugs between the quartiles of cell lines expressing the lowest and highest levels of CDC20. Here as well, increased CDC20 expression was correlated with increased sensitivity to SAC inhibitor drugs (MPS1 inhibitors MPI-0479605 and AZ3146) (Fig. 2C,D). Notably, the association between CDC20 expression and MPS1 inhibition is comparable to that seen for known drug targets and gene expression or gene-dependency of these targets. For example, CCNE1 expression and CDK2i ($R^2 = 0.068$), ATR dependency and ATR kinase inhibitor ($R^2 = 0.031$), BRD4 dependency and BRD4i ($R^2 = 0.022$), or TOP2A and TOP2Ai ($R^2 = 0.018$) (Cohen-Sharir et al, 2021), suggesting that this significant correlation is biologically meaningful.

To ask whether the association between CDC20 expression levels and the sensitivity to SAC inhibitors goes beyond a mere association of a cell cycle marker and the response to cell cycle inhibitors, we examined the association between CDC20 expression and the response to ~6500 drugs. We ranked the association between the CDC20 gene expression and the cellular response to all of the drugs included in the PRISM drug repurposing screen, relative to the association of all other genes to each drug ((Corsello et al, 2020); "Methods"), and found that the association of CDC20 expression with drug sensitivity was significantly stronger for MPS1 inhibitors than for all other classes of drugs (Fig. 2E), showing the specific importance of CDC20 in the response to SAC inhibition. Overall, the human cancer cell line data suggest that elevated levels of CDC20 expression are associated with increased sensitivity to both genetic and chemical disruption of the SAC.

To validate these findings in the context of human cancer cells, we knocked down CDC20 in the human colon cancer cell line HCT116, either by siRNA or by shRNA (Fig. EV2A–D), and assessed cell survival and proliferation under SAC inhibition. Similar to our observations in the mouse cells, CDC20 depletion using either shRNA (Fig. 2F) or siRNA (Fig. 2G) led to a substantial decrease in the cellular sensitivity of human cancer cells to SAC inhibition. These results suggest that both in mouse and in human cells, CDC20 expression as a single protein, regardless of the expression of other APC/C core components, is sufficient to mediate the cellular response to SAC inhibition.

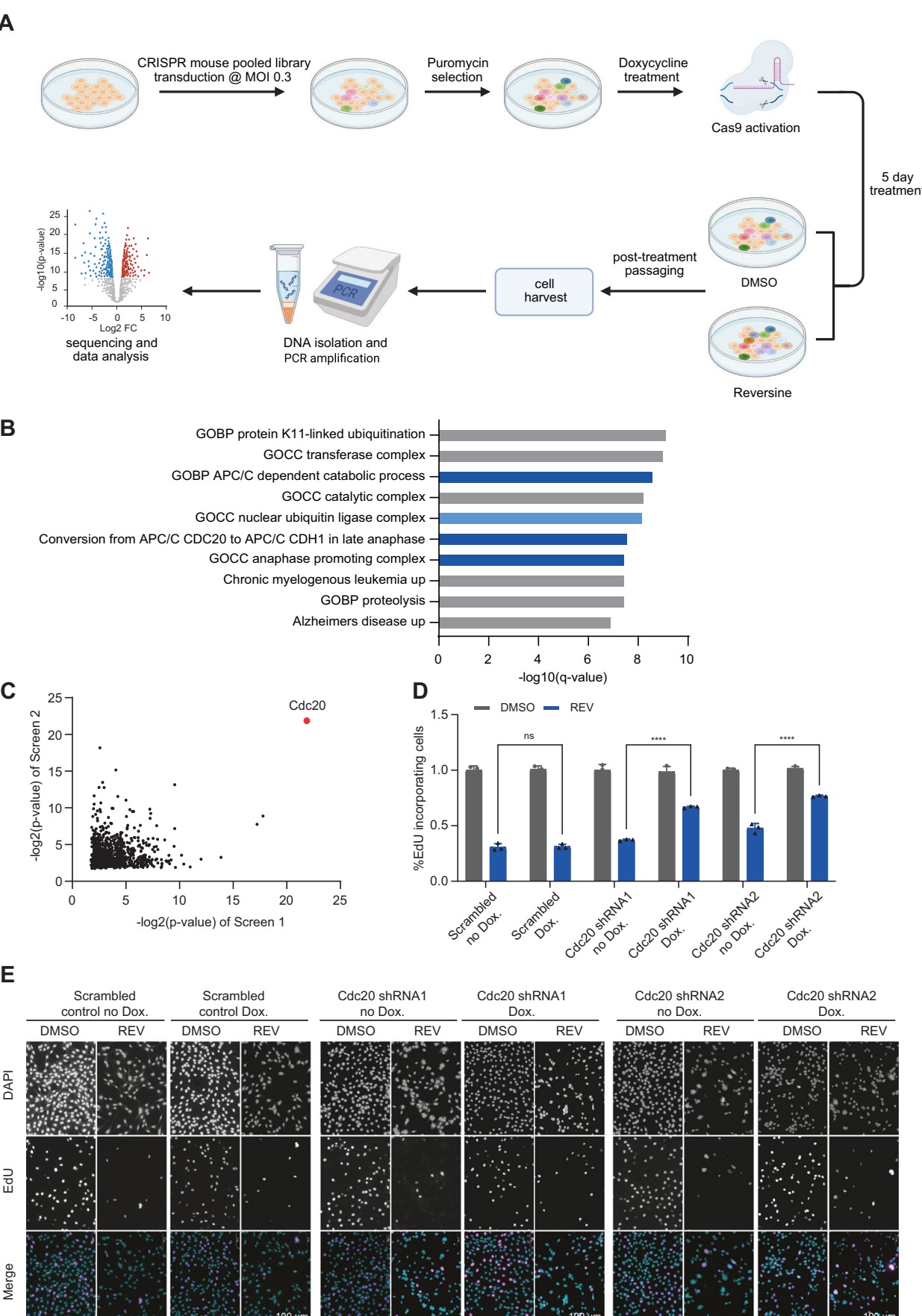

**Figure 1. Cdc20 loss is associated with resistance to SAC inhibition.**

(A) Schematic overview of the CRISPR screens performed to identify genes and pathways involved in resistance to SAC inhibition. The illustration was created using www.biorender.com. (B) Cellular pathways enriched in the top ranking 10% of genes in both CRISPR screens. (C) Correlation between the top-ranked 25% of genes in both CRISPR screens, based on their statistical significance. Cdc20 (red) is the top hit in both screens. *P* values were calculated using the RRA method (see "Methods"). (D, E) Quantification (D) and representative images (E) of an EdU incorporation assay in 3T3 cells after Cdc20 depletion by shRNA and treatment with 250 nM Reversine. Two-way ANOVA (*N*, number of biological replicates. *N* = 3; ns, *P* value = 0.9998; ****, *P* value < 0.0001). Error bars represent the standard deviation (SD) of the mean. Source data are available online for this figure.

## Increased CDC20 expression is associated with increased sensitivity of aneuploid cells to SAC inhibition

We recently found that aneuploid cells are initially more resistant than diploid cells to SAC inhibition but become more sensitive to the drug over time (Cohen-Sharir et al, 2021). We therefore wondered whether the expression levels of CDC20 differ between diploid and aneuploid cells, and if so, whether this underlies their differential drug sensitivity. We therefore compared CDC20 and APC/C mRNA expression levels between the top and bottom aneuploid quartiles of human cancer cell lines in the DepMap (Tsherniak et al, 2017; Zerbib et al, 2024). We found that highly aneuploid cells overexpress CDC20 mRNA in comparison to near-diploid cells (Fig. 3A). To evaluate whether the differential response of aneuploid human cancer cell lines to SAC inhibitors is indeed related to their increased CDC20 expression, we assessed the effect of CDC20 expression on the association between aneuploidy and drug response. Inclusion of CDC20 mRNA expression as a covariate in a linear regression model completely abolished the significant association between aneuploidy and the response to MPS1 inhibitors or to genetic disruption of SAC core components (Figs. 3B and EV3A, respectively). These results support the notion that CDC20 is a major determinant of the differential response of aneuploid cells to SAC inhibition.

To validate the increased expression of CDC20 in aneuploid cells and its direct role in determining their sensitivity to SAC inhibition, we quantified CDC20 protein expression levels in the human colon cancer cell line HCT116 and its highly aneuploid derivatives HPT1 and HPT2, as well as in the immortalized epithelial cell line RPE1 and its aneuploid derivatives, RPT1, RPT3 and RPT4. These cells became spontaneously aneuploid after induced tetraploidization as described in Kuznetsova et al

(Kuznetsova et al, 2015). As CDC20 levels fluctuate throughout the cell cycle (Foe et al, 2011), we first used immunofluorescence microscopy to quantify CDC20 expression levels at their peak, during metaphase, in single cells from the HCT-HPT and the RPE-RPT post-tetraploid isogenic-aneuploid cell systems (Figs. 3C and EV3B,C). To quantify CDC20 levels using an independent approach, we synchronized the cells at prometaphase using Nocodazole (Fig. EV3D) and quantified CDC20 expression by Western blots (Figs. 3D and EV3E). In both methods, highly aneuploid cells expressed significantly higher levels of CDC20 compared to their near-diploid counterparts. A recent work by Tsang et al has shown that there are several CDC20 protein isoforms that may affect mitotic duration (Tsang and Cheeseman, 2023). However, we did not observe a significant difference in the relative abundance of the isoforms between diploid and aneuploid cells (Fig. EV3F).

To verify the role of CDC20 in the increased sensitivity of aneuploid cells to SAC inhibition, we depleted it in the aneuploid HPT cells using siRNA (Fig. EV4A,B,E,F) or shRNA (Fig. EV4C,D). We found that CDC20 depletion, either by siRNA (Fig. 3E,F,H) or by shRNA (Fig. 3G,I) reduced the sensitivity to SAC inhibition. Next, we treated the near-diploid HCT116 cells with the MPS1 inhibitor Reversine to generate a heterogeneous aneuploid cell population and validated that this aneuploidization was indeed accompanied by increased CDC20 expression, using bulk quantification by Western blotting after cell synchronization (Fig. 3J). In this population of aneuploid cells, CDC20 depletion reduced the sensitivity to SAC inhibition as well (Figs. 3K and EV4G,H). Together, these results further demonstrate that CDC20 overexpression is a major contributor to the sensitivity of aneuploid cells to SAC inhibition.

**Figure 2. CDC20 expression predicts sensitivity to genetic and chemical SAC perturbation in human cancer cell lines.**

(A, B) Correlation between mRNA expression of an extended APC/C signature (left), an APC/C signature without CDC20 (middle), or CDC20 alone (right) and sensitivity to genetic disruption of the core SAC components BUB1B (A) and MAD2L1 (B) in human cancer cell lines from the DepMap. The "APC/C subunit-only" signature contains the 14 core APC/C subunits (Yamano, 2019), while the "extended APC/C signature" described by Thu et al (Thu et al, 2018) contains three additional APC/C co-factors, including CDC20. The genes included in each signature are listed in Table EV1. Shown are Spearman's correlation rho and *P* values. Spearman correlation (*N*, number of cell lines; *N* = 661 for MAD2L1 or BUB1B vs extended APC/C or subunit-only APC/C, *N* = 662 for MAD2L1 or BUB1B vs extended APC/C or subunit-only APC/C). RNAi dependency scores were obtained from the Achilles genome-wide RNAi screen, DepMap 22Q2 (Tsherniak et al, 2017). (C, D) Comparison of the sensitivity to two chemical MPS1 inhibitors—MPI-0479605 (C) and AZ3146 (D) between cell lines in the top vs. bottom mRNA expression quartiles of CDC20. Drug sensitivity data were obtained from PRISM repurposing primary CRISPR screen, DepMap 23Q2 (Corsello et al, 2020). Two-sided *t* test (*N* = 260 for CDC20 vs MPI-0479605, *N* = 198 for APC/C vs MPI-0479605, *N* = 274 for CDC20 vs AZ3146, *N* = 276 for APC/C vs AZ3146; *, *P* value = 0.0435; **, *P* value = 0.006). (E) Distribution of the correlation between CDC20 expression and sensitivity to ~6500 different drugs taken from the PRISM primary repurposing screen 2023, ordered by gene ranking percentile (see methods). Comparison between drug classes was performed by Student's *t* test. The ranking of CDC20 compared to all protein-coding genes in the response to MPS1 inhibitors is much higher than its average ranking in response to all other drugs. (F, G) Percent of EdU-incorporating HCT116 cells following SAC inhibition (125 nM Reversine), with and without CDC20 depletion by shRNA (F) or siRNA (G). CDC20 depletion increased the fraction of proliferating cells following drug treatment. Two-way ANOVA (*N*, number of biological replicates; *N* = 3; ****, *P* value < 0.0001). Error bars represent the standard deviation (SD) of the mean. Source data are available online for this figure.

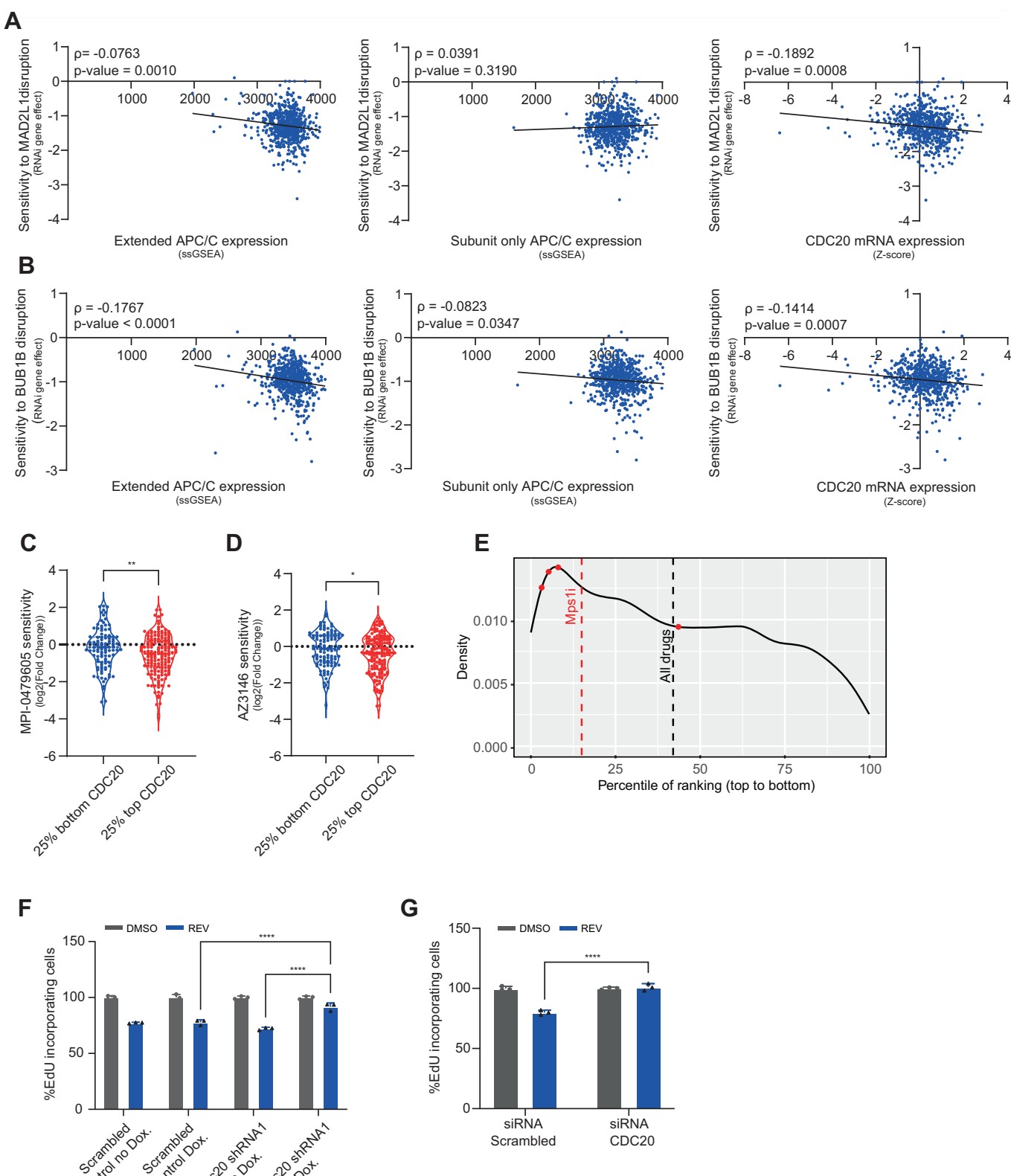

## CDC20 expression levels determine the prevalence

## of mitotic errors

Next, we set out to mechanistically explore the association between aneuploidy, CDC20 expression levels, and the response to SAC

inhibition. We previously found highly aneuploid cells to exhibit more mitotic errors and linked their CIN to their elevated response to SAC inhibition (Cohen-Sharir et al, 2021). We therefore hypothesized that the elevated CDC20 expression levels may be associated with the elevated CIN of the aneuploid cells. To test this,

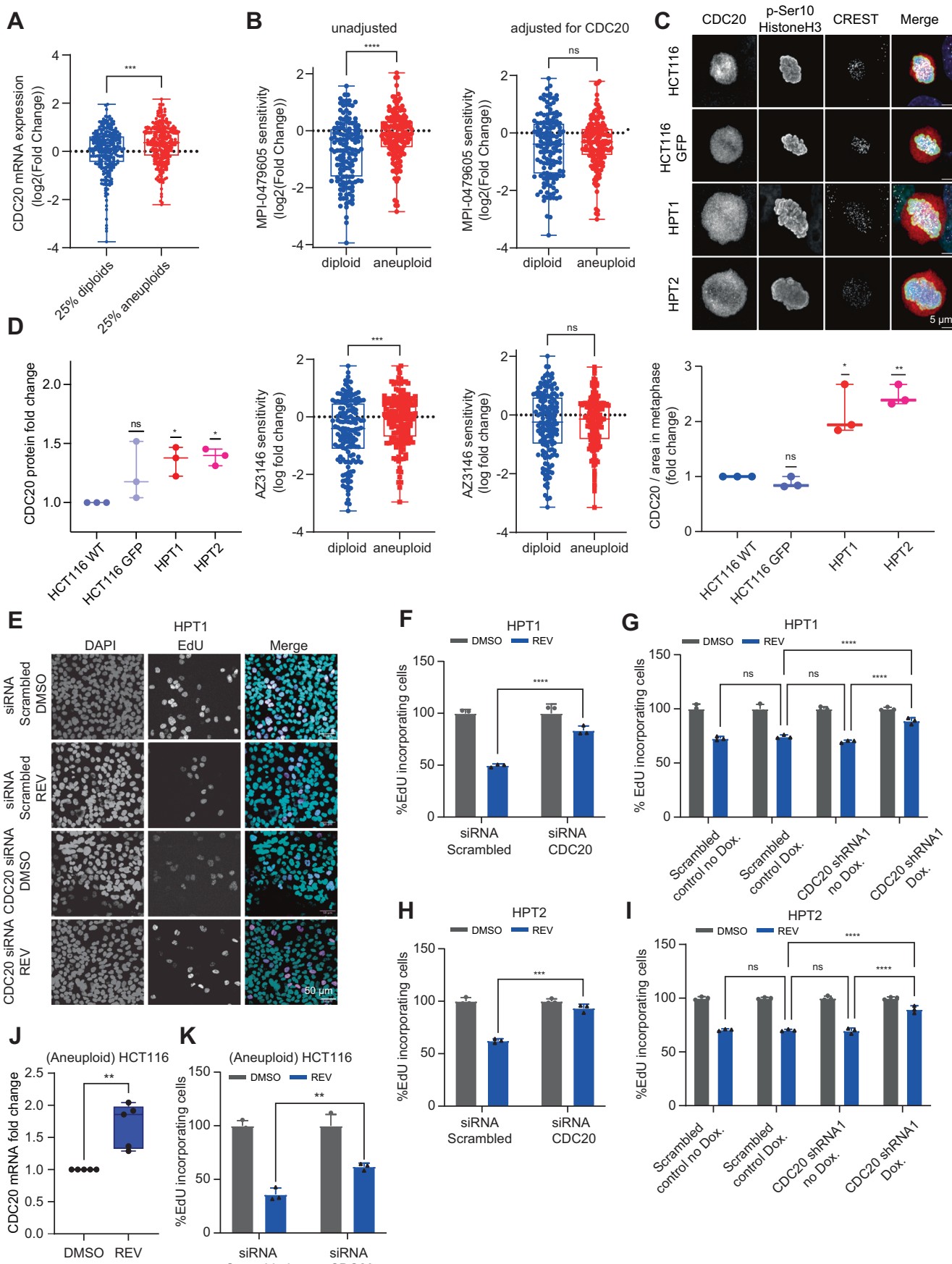

Figure 3.  Increased CDC20 expression is associated with the preferential response of aneuploid cells to SAC inhibition.

(A) Analysis of mRNA expression levels in ~1000 human cancer cell lines. Highly aneuploid cancer cell lines express significantly higher mRNA levels of CDC20 compared to near-diploid cell lines. Box plots show the median (middle line) and interquartile range (IQR; box edges). Whiskers extend to values within 1.5×IQR from the 25th and 75th percentiles. Two-sided t test (N, number of cell lines; N = 531; ***, P value = 0.0002). (B) Left: correlation between ploidy and the sensitivity to the MPS1 inhibitors MPI-0479605 (top) and AZ3164 (bottom). Right: same correlation with CDC20 removed as a covariate using a linear model. When removing the effect of CDC20 expression the trend becomes insignificant. Two-sided t test (ns, P value = 0.3051 for MPI-0479605 and P value = 0.6616 for AZ3146; ***, P value < 0.0009; ****, P value < 0.0001). (C) Representative images and quantification of CDC20 at metaphase in the isogenic-aneuploid system HCT116-HPT (Kuznetsova et al, 2015), after synchronization with 7.5 μM RO-3306 and 10 μM MG-132. Highly aneuploid cells express significantly higher levels of CDC20 than their diploid counterparts. One-sample t test (N, number of biological replicates; N = 3; ns, P value = 0.1894; *, P value = 0.0484; **, P value = 0.0053). Bars represent the data range. (D) Bulk quantification of CDC20 protein expression in the HCT116-HPT system during mitosis after synchronization with 330 nM Nocodazole. Here as well, the highly aneuploid cells express higher CDC20 protein levels than their diploid counterparts. One-sample t test (N = 3; ns, P value = 0.2261; *, P value = 0.0378 for HPT1 and P value = 0.0110 for HPT2). Bars represent the data range. (E) Representative images of EdU incorporating HPT1 cells with and without CDC20 siRNA depletion and Reversine treatment. (F, G) Percent of EdU incorporating cells in HPT1 cells after CDC20 depletion with siRNA (F) or shRNA (G), with and without treatment with 250 nM Reversine. Two-way ANOVA (N = 3; ns, P value = 0.8760 or P value = 0.2303 (from left to right); ***, P value = 0.0001; ****, P value < 0.0001). Error bars represent the standard deviation (SD) of the mean. (H, I) Percent of EdU incorporating cells in HPT2 cells after CDC20 depletion with siRNA (H) or shRNA (I). Two-way ANOVA (N = 3; ns, P value = 0.9966 or 0.9856 (from left to right); ****, P value < 0.0001). CDC20 depletion in both aneuploid cell lines significantly reduced sensitivity to SAC inhibition. (J) Bulk quantification of CDC20 in wild-type HCT116 and after following aneuploidy induction using 250 nM Reversine. Two-sided t test (N = 5; **, P value = 0.0019). (K) Percent of EdU incorporating cells in aneuploid HCT116 cells after CDC20 depletion with siRNA and treatment with 250 nM Reversine. Two-way ANOVA (N = 3; **, P value = 0.0029). In these cells too, CDC20 depletion decreases the sensitivity to SAC inhibition. Source data are available online for this figure.

we compared CDC20 protein expression in cells undergoing normal and erroneous cell divisions in the HCT116-based and RPE1-based isogenic-aneuploid systems and found that regardless of ploidy background, cells undergoing aberrant mitosis exhibited higher levels of CDC20 during metaphase (Figs. 4A,B and EV5A,B). These results suggest a likely link between increased CDC20 expression and chromosome missegregation, which may be both a cause and a consequence of the aneuploid state (Pfau and Amon, 2012; Ben-David and Amon, 2020; Holland and Cleveland, 2009).

We next characterized the effect of CDC20 levels on the overall rate and severity of mitotic errors in mouse and human cells using live cell imaging. To this end, we quantified mitotic aberrations in control and CDC20-knockdown cells, with or without SAC inhibition, and scored the mitotic aberrations according to their severity, as has been done in previous works ((Thu et al, 2018; Crozier et al, 2022; Huis in 't Veld et al, 2019); see Methods). In both mouse and human 3T3 and HCT116 cells subjected to SAC inhibition, CDC20 depletion by siRNA or shRNA significantly alleviated Reversine-induced CIN, reducing the prevalence of severe mitotic aberrations in cells exposed to SAC inhibition (Figs. 4C–F and EV5C,D). Similarly, CDC20 knockdown alleviated CIN in the highly aneuploid HPT1 and HPT2 cells, as well as in HCT116 cells in which aneuploidy was induced by Reversine (Fig. 4G–J,K,L respectively). Together, these findings demonstrate a significant association between CDC20 expression and the degree of CIN induced by SAC inhibition. Specifically, reduction in CDC20 levels is causally associated with decreased severity and overall rate of mitotic aberrations in mouse and human cells of various transformation and ploidy statuses. Increased expression of CDC20 is associated with the opposite phenotypes, but we could not demonstrate causality as we failed to overexpress CDC20 in our cells (Fig. EV6B–H).

## CDC20 expression levels determine the prevalence of mitotic errors by regulating mitosis duration

Metaphase duration is associated with the ability of cells to correct errors on the mitotic spindle, so that prolonged mitosis could reduce mitotic errors (Rieder and Maiato, 2004; Bloomfield et al, 2021). As high CDC20 levels promote the transition into anaphase (Greil et al, 2022), we hypothesized that low expression and activity of CDC20 will prolong mitosis, which may underlie the association that we observed between CDC20 and CIN levels. We thus quantified metaphase duration following gene knockdown in all cell lines. CDC20 knockdown significantly prolonged metaphase duration in 3T3 (Fig. 5A), HCT116 (Fig. 5B,C), HPT1 (Fig. 5D), HPT2 (Fig. 5E), and the Reversine-induced aneuploid HCT116 cells (Fig. 5F). The differences in mitotic phenotypes and duration in the 3T3, HCT116, HPT1, HPT2, aneuploid HCT116 cells with and without CDC20 can be observed in Movies EV1–17. These results suggest that CDC20 knockdown may reduce CIN by prolonging metaphase, thereby allowing more time for the correction of mitotic errors. To test whether the length of mitosis indeed influences the rate of CIN, we prolonged mitosis in the HCT116-HPT isogenic cells in a CDC20-independent manner by arresting the cells at G2/M and releasing them from arrest in the presence or absence of the proteasome inhibitor MG-132 (see "Methods"), and quantified the rate of mitotic aberrations using immunofluorescence microscopy. Highly aneuploid HPT cells exhibited more aberrant mitoses than their near-diploid counterparts, as previously reported (Fig. 4E,G,I,K (Cohen-Sharir et al, 2021)), but the prevalence of mitotic aberrations significantly decreased when metaphase duration was extended (Figs. 5G and EV6A).

We conclude that CDC20 depletion results in prolonged metaphase and reduced prevalence of mitotic aberrations across mouse and human cell lines with various ploidies. These results indicate that CDC20 is a key player in the cellular response to SAC inhibition, and that it acts by altering metaphase duration and affecting the overall level of chromosomal instability induced by SAC inactivation. These results provide a mechanistic explanation for the increased long-term sensitivity of aneuploid cells—which overexpress CDC20—to SAC inhibitors.

# Discussion

Aneuploidy and CIN are associated with cancer progression and drug response. CIN and aneuploidy were previously linked to

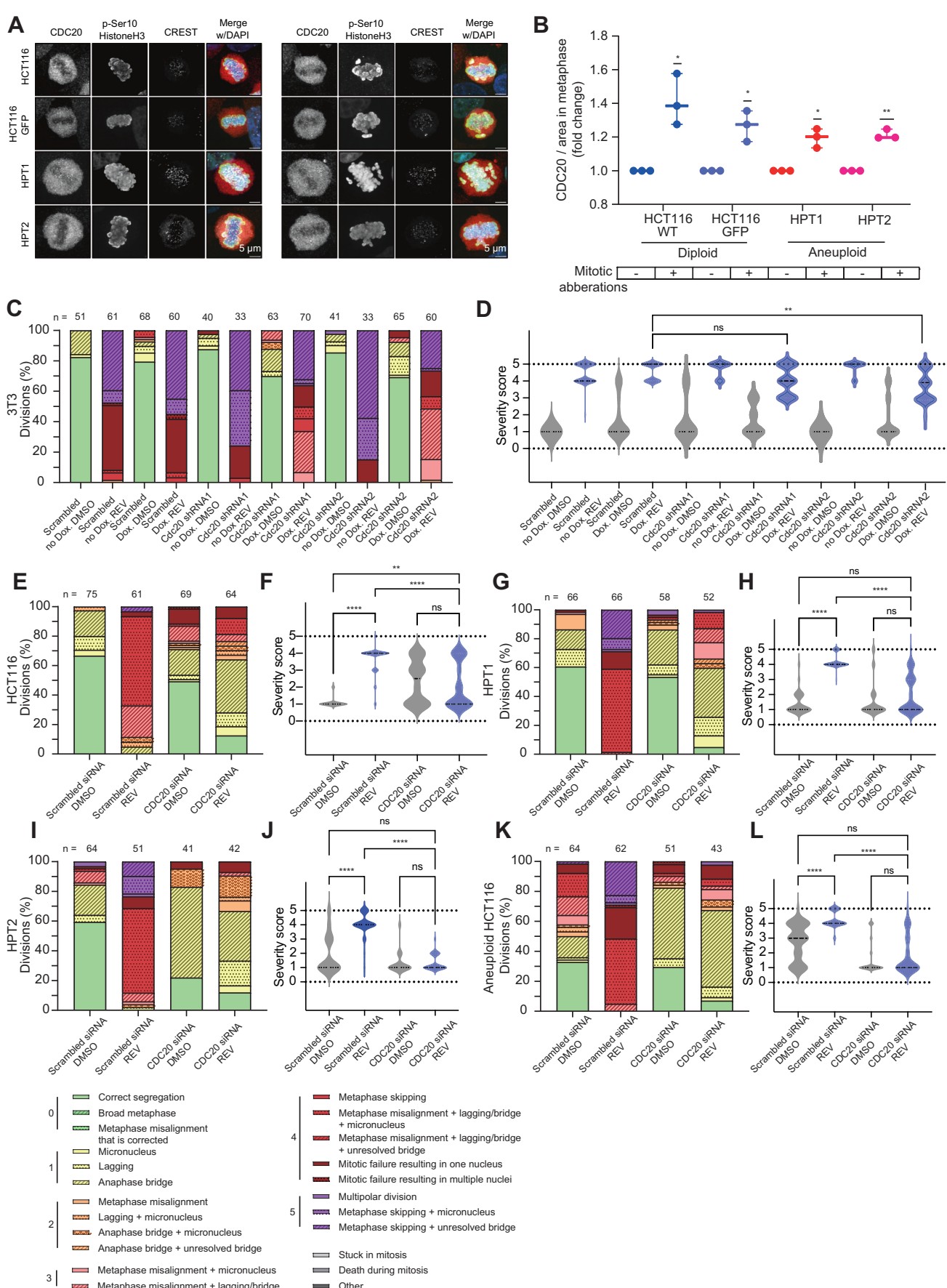

Figure 4.   CDC20 expression levels determine the prevalence of mitotic errors.

(A, B) Representative immunofluorescence images (A) and quantification (B) of CDC20 protein levels during normal and aberrant metaphases, in single HCT116 cells and their highly aneuploid derivatives after synchronization with 7.5 μM RO-3306. Regardless of ploidy background, cells with mitotic aberrations express significantly higher levels of CDC20 during metaphase than cells undergoing normal division. One-sample $t$ test ($N$, number of biological replicates, $N = 3$; *, $P$ value = 0.0427 for HCT116, $P$ value = 0.0367 for HCT116 GFP, $P$ value = 0.0268 for HPT1; **, $P$ value = 0.0061 for HPT2). Bars represent the data range. (C–L) Distribution (left) and severity quantification (right) of mitotic abnormalities in cells treated with 125 nM (or 250 nM) Reversine, under control conditions or CDC20 depletion by siRNA or shRNA. Mitotic aberrations were identified by live-cell imaging and scored on a severity scale of 0–5, then grouped and colored by score. The changes in severity distribution across samples were assessed using one-sided Kruskal–Wallis tests, as elaborated in "Methods". 3T3 (C, D) and HCT116 (E, F) cells treated with 250 nM or 125 nM Reversine, respectively, exhibit reduced mitotic aberrations after CDC20 depletion. Sample size (N) in (F, H, J, L) corresponds to those specified on the barplots in (E, G, I, K), respectively. One-sided Kruskal–Wallis test (ns, $P$ value = 0.1057 (D) or 0.9999 (F); **, $P$ value = 0.0017 (D) or 0.0067 (F); ****, $P$ value < 0.0001). (G–L) HPT1 (G, H), HPT2 (I, J) and cells from the third aneuploid HCT116 isogenic derivative (K, L) treated with 125 nM Reversine also exhibit significantly decreased mitotic aberrations after CDC20 depletion. One-sided Kruskal–Wallis test (ns, $P$ value > 0.9999 (H, J) and 0.0714 or $P$ value > 0.9999 (L, from left to right); ****, $P$ value < 0.0001). Source data are available online for this figure.

increased drug resistance to many anticancer drugs (Cohen-Sharir et al, 2021; Lee et al, 2011; Replogle et al, 2020; Lukow et al, 2021; Marquis et al, 2021), but they can also lead to increased sensitivity to specific therapies, such as SAC inhibition (Cohen-Sharir et al, 2021), KIF18A inhibition (Cohen-Sharir et al, 2021; Marquis et al, 2021), Src1 inhibition (Schukken et al, 2020), IL6-R inhibition (Hong et al, 2022), MAPK signaling inhibition (Zerbib et al, 2024) and proteasome inhibition (Ippolito et al, 2024). Understanding the molecular mechanisms that mediate the associations between CIN/aneuploidy and drug response can guide the development of new therapies and promote our basic understanding of cancer.

MPS1 inhibitors operate by inactivating the chromosome segregation-control mechanism, culminating in a cascade of chromosomal instability and aneuploidy that ultimately leads to cell cycle arrest and cell death. Biomarkers that enable predicting which patients respond well to MPS1 inhibitors are urgently needed and may arise from better understanding of the molecular mechanism(s) of action of these drugs. Previous studies have demonstrated that the response to SAC inhibition is driven by the APC/C (Sansregret et al, 2017; Thu et al, 2018; Wild et al, 2016). Here, we show that CDC20 is a major driver of the response to genetic and chemical SAC perturbation, and that its expression predicts drug sensitivity better than that of the APC/C subunits themselves. Our findings are supported by two functional genetic CRISPR genome-wide screens and the analyses of data from >1000 human cancer cell lines.

We show that CDC20 expression is significantly correlated with sensitivity to multiple forms of SAC inhibition and that CDC20 expression levels are elevated in aneuploid cells. Furthermore, we show that CDC20 depletion decreases the sensitivity to SAC inhibition in both mouse and human cells. Mechanistically, we show that increased expression of CDC20 is significantly associated with an increase in mitotic errors, while CDC20 depletion results in prolonged metaphases and decreased prevalence and severity of mitotic errors under SAC inhibition. We note that we demonstrated the association between mitotic errors and high levels of CDC20 using orthogonal approaches: (a) high-resolution immunofluorescence of single cells in mitosis; (b) Western blotting of the cell population following cell cycle synchronization and release into mitosis; and (c) mitosis synchronization by MG-132. However, these analyses are either fixed (IF) or performed on the bulk cell population (WB), and it will be interesting to genetically label endogenous CDC20 in the future and follow its levels in normal and abnormal mitosis using live cell imaging.

We show that prolonging mitosis duration decreases the prevalence of mitotic errors, which could explain the increased

tolerance to SAC inhibition upon CDC20 depletion. As cells with lower CDC20 levels delay anaphase onset and spend more time in metaphase, they have more time to correct spindle abnormalities, resulting in the observed decrease in mitotic error rate and severity, to the acquisition of fitter karyotypes, and to the overall better survival of such cells under SAC inhibition.

Our study provides a molecular link between aneuploidy and the sensitivity to SAC inhibition. We show for the first time that aneuploid cells overexpress CDC20 (but not the APC/C subunits), both across human cancer cell lines and in two experimental models of aneuploidy induction (by tetraploidization or by Reversine treatment), potentially explaining the increased inherent chromosomal instability in these cells and their enhanced sensitivity to the excess CIN caused by SAC inhibition. We hypothesize that the upregulation of CDC20 in aneuploid cells could be important for their adaptation to the aneuploid state, enabling them to bypass the SAC despite spindle abnormalities. However, at the same time this overexpression leads to their increased sensitivity to SAC inhibition. While we conclusively showed the impact of CDC20 depletion on chromosome segregation, mitotic timing and SACi sensitivity, we note that we were unable to overexpress CDC20 in our cells (Fig. EV6AB–H), probably due to high toxicity of constitutive activity of this cell cycle-controlled protein in our cells, and this prevented us from conclusively showing a causal association between the overexpression of CDC20 and the drug response. Our analysis of CDC20 overexpression is therefore limited to the natural increased expression of the protein (e.g., in aneuploid cells).

Effectively, SAC inhibition and CDC20 overexpression result in a similar outcome – they allow cell division to continue even in the presence of unattached kinetochores. More specifically, these conditions could cause cells to move through prometaphase and metaphase more quickly, thereby preventing the proper segregation of their chromosomes. Therefore, each of these two conditions individually leads to chromosomal instability and aneuploidization (Mason et al, 2017; Weaver and Cleveland, 2009; Mondal et al, 2007; Adell et al, 2023). Combined, we show that these conditions lead to excessive chromosomal instability and the formation of unfit karyotypes, which is poorly tolerated by the cells. Complementarily, CDC20 downregulation prolongs metaphases and reduces the overall levels of chromosomal instability in the cells, allowing them to better tolerate the effects of SAC inhibition, thereby making them more resistant to this class of drugs.

CDC20 overexpression is known to be a negative prognostic marker in multiple cancer types (Xian et al, 2022; Wu et al, 2013; Karra et al, 2014; Kato et al, 2012), possibly due to its association

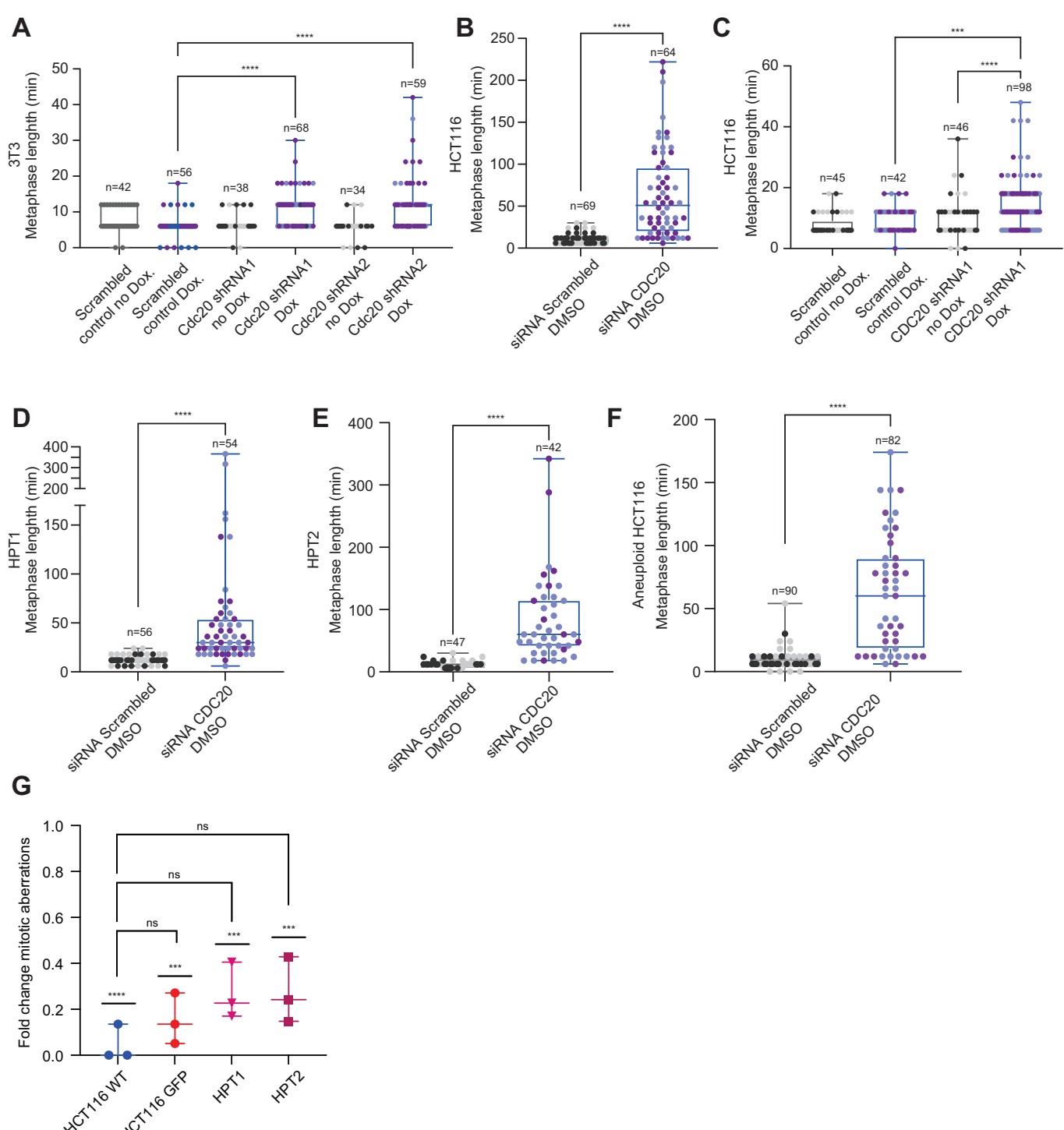

**Figure 5. CDC20 expression levels determine metaphase duration.**

(A–C) Metaphase duration in mouse 3T3 cells (A) or human HCT116 cells (B, C) under control conditions or CDC20 depletion by siRNA (B) or shRNA (A, C). CDC20 depletion in both cell lines resulted in an increased metaphase duration. Box plots show the median (middle line) and interquartile range (IQR; box edges). Whiskers extend to values within 1.5×IQR from the 25th and 75th percentiles. One-way ANOVA or two-sided t test (N, number of biological replicates; N = 3; ***, P value = 0.0002; ****, P value < 0.0001). (D–F) Metaphase duration in the aneuploid HPT1 (D), HPT2 (E) or the third HCT116 aneuploid derivative (F) under control conditions or CDC20 depletion by siRNA. In all three cell lines, CDC20 depletion led to a significant increase in metaphase duration. Two-sided t test (N = 3, ****, P value < 0.0001). (G) Fold change of mitotic aberrations in HCT116-HPT cells that underwent a prolonged (120 min) vs normal (45 min) metaphase. To prolong metaphase, cells were synchronized to the G2/M border with 7.5 nM RO-3306, released for 30 min and then treated with 10 μM MG-132 for 90 additional minutes before fixation. To synchronize cells in normal metaphase, cells were treated only with RO-3306 and released from arrest for 45 min before fixation. The rate of mitotic errors decreased significantly in the cells that underwent prolonged metaphases, as can be seen by the fold change of mitotic aberrations. Two-sided t tests and one-way ANOVA (N = 3; ns, P value = 0.5598 for HCT116 GFP, P value = 0.1099 for HPT1, P value = 0.1016 for HPT2; ***, P value = 0.0002 for HCT116 GFP, P value = 0.0005 for HPT1 and P value = 0.0009 for HPT2; ****, P value < 0.0001). Bars represent the data range. Source data are available online for this figure.

with higher tumor cell division rate (Shang et al, 2018; Chu et al, 2019). This might suggest high CDC20 expression as a favorable prognostic marker of the response to MPS1 inhibitors, and perhaps even more so in highly aneuploid tumors. Our findings may be clinically relevant as several MPS1 inhibitors are currently in clinical trials, with one of them being fast-tracked by the FDA (NCT05251714). However, the associations that we observed remain to be validated clinically using data from patients treated with SAC inhibitors (data to which we currently do not have access). More broadly, our findings suggest that CDC20 may be a potential therapeutic vulnerability for aneuploid cells that can be targeted by CIN-inducing therapies.

# Methods

### Reagents and tools table

| Reagent/resource | Reference or source | Identifier or catalog number |
|---|---|---|
| **Experimental models** | | |
| NIH-3T3 | | CRL-1658 |
| HCT116 | Kuznetsova et al, 2015 | CCL-247 |
| HPT1, HPT2 | Kuznetsova et al, 2015 | |
| RPE1 | Kuznetsova et al, 2015 | CRL-4000 |
| RPT1, RPT3, RPT4 | Kuznetsova et al, 2015 | |
| **Recombinant DNA** | | |
| Lenti-iCas9-neo | A gift from Qin Yan | Addgene plasmid #85400 |
| PMD2 | Addgene | Plasmid #12260 |
| VSVG | Addgene | Plasmid #12259 |
| Tet-pLKO-puro | Addgene | Plasmid #21915 |
| Tet-pLKO-puro-Scrambled | Addgene | Plasmid #47541 |
| **Antibodies** | | |
| CDC20 | Santa Cruz | sc-13162 |
| CDC20 | Abcam | ab26483 |
| β-Actin | Cell Signaling | 4970 or 3700S |
| GAPDH | Cell signalling | D16H11 |
| Vinculin | Cell Signalling | 4650S |
| IRDye 800CW Goat anti-Rabbit IgG (H + L) | Licor | 926-32211 |
| Goat anti-Mouse | Jackson lab | 115-035-003 |
| Anti-mouse IgG, HRP-linked Ab. | Cell Signaling | #7076S |
| IRDye 680CW Goat anti-Mouse IgG (H + L) | Licor | 926-68070 |
| Anti-rabbit IgG, HRP-linked Ab | Cell Signaling | #7074S |

| Reagent/resource | Reference or source | Identifier or catalog number |
|---|---|---|
| Goat anti-Rabbit | Jackson lab | 111-035-003 |
| Rabbit anti-phospho-Histone H3 (Ser10) | Cell signaling | 9706 |
| Anti-phospho-Histone H3 (Ser10) | Sigma-Aldrich | 05-806 |
| Anti-CREST | Antibodies Incorporated | 15-234 |
| Alexa 488-, Alexa Cy3- and Alexa 647-labeled secondary antibodies | Invitrogen | A28175, A-31573, A27040 |
| Anti-mouse Alexa-555, anti-rabbit Alexa-488 | Cell signaling | 4409, 4412 |
| **Oligonucleotides and other sequence-based reagents** | | |
| CDC20 siRNA | Tsang and Cheeseman, 2023 | 5′-CGGAAGACCUGCCGUUACAUU |
| Cdc20 shRNA1 | | GCAGCAGAAACGACTTCGAAA |
| Cdc20 shRNA2 | | GCCGAACTCCTGGCAAATCTA |
| CDC20 shRNA1 | | AGACCAACCCATCAC |
| CDC20 shRNA2 | | CCCATTACAAGGAGCTCAT |
| Cdc20 (Sense) | | TTCGTGTTCGAGAGCGATTTG |
| Cdc20 (Anti-sense) | | ACCTTGGAACTAGATTTGCCAG |
| Actb (Sense) | | CTAGGCACCAGGGTGTGATG |
| Actb (Anti-sense) | | GGCCTCGTCACCCACATAG |
| CDC20 (Sense) | | GACCACTCCTAGCAAACCTGG |
| CDC20 (Anti-sense) | | GGGCGTCTGGCTGTTTTCA |
| Tubulin (Sense) | | CTTCGTCTCCGCCATCAG |
| Tubulin (Anti-sense) | | CGTGTTCCAGGCAGTAGAGC |
| **Chemicals, enzymes, and other reagents** | | |
| puromycin | Invivogen | ant-pr-1 |
| DMSO | Sigma-Aldrich | D2650-100ML |
| reversine | Sigma-Aldrich | R3904 |
| NdeI, SacII, AgeI and EcoRI restriction enzymes | New England Biolabs | R0111S, R0157S, R3552S, R3101S |
| Exonuclease I | New England Biolabs | M0293S |
| polybrene | Sigma-Aldrich | TR-1003-G |
| doxycycline | Sigma-Aldrich | hyclate D9891 |
| control pool siRNA | Dharmacon | D-001206-13 |
| Lipofectamine RNAiMAX | Thermo | 13778075 |
| RO-3306 | Sigma-Aldrich | S7157 |
| MG-132 | Tocris | 1095 |
| Nocodazole | Sigma-Aldrich | 31430-18-9 |
| Lipofectamine RNAiMAX | Thermo | 13778075 |
| chemiluminescence | Millipore | #WBLUR0500 |

| Reagent/resource | Reference or source | Identifier or catalog number |
|---|---|---|
| **Software** | | |
| DESeq2 | Love et al, 2014 | |
| MAGeCK ranking aggregation method (Robust Rank Algorithm (RRA) | van der Noord et al, 2023 | |
| original GSEA method | GSEA-MSigDB website | https://www.gsea-msigdb.org/gsea/msigdb |
| GraphPad Prism 9.1 | | |
| BioRender | | www.biorender.com |
| ImageJ v1.53C | | |
| NineAlliance v18.12 | | |
| **Other** | | |
| Alt-R™ Genome Editing Detection Kit | Integrated DNA Technologies | 1075931 |
| Mouse Two Plasmid Activity-Optimized CRISPR Knockout Library | Addgene | #1000000096 |
| QIAamp DNA Blood Maxi Kit | Qiagen | 51192 |
| Streptavidin T1 Dynabeads | ThermoFisher Scientific | 65601 |
| NucleoSpin Gel and PCR Clean-up kit | Machery-Nagel | 740609.50 |
| NextSeq 500 sequencer | Illumina | 20024906 |
| RNA plus isolation kit | Qiagen | MN 740984.250 |
| LunaScript RT SuperMix Kit | Bioke | M3010X |
| iTaq Universal SYBR Green Supermix | BioRad | 1725124 |
| Odyssey imaging system | LI-COR Biosciences | |
| Leica Histo-Fluo | Leica Biosystems | |

The sgRNA sequences were obtained from the AddGene website ((Wang et al, 2017), https://www.addgene.org/pooled-library/sabatini-crispr-mouse-high-activity-two-plasmid-system/).

## CRISPR-Cas9 screen and validation

### Introduction of Cas9 and sgRNA library into 3T3 cells

NIH-3T3 cells were transfected with Lenti-iCas9-neo (Addgene plasmid #85400, a gift from Qin Yan) (Cao et al, 2016), along with PMD2 and VSVG lentivirus vectors to introduce doxycycline-inducible Cas9. Following transfection, cells were selected with 400 µg/ml neomycin and monoclonal Cas9-expressing cells were isolated in 96-well plates. Cas9 activity was confirmed using the Alt-R™ Genome Editing Detection Kit (Integrated DNA Technologies, 1075931).

To introduce the sgRNA library, 377 million iCas9-expressing 3T3 cells were transduced with the Mouse Two Plasmid Activity-Optimized CRISPR Knockout Library (Wang et al, 2017) (Addgene; 1000000096) using spinfection at an MOI of 0.3 to ensure a final library coverage of 500×. The required number of iCas9 3T3 cells (377 million) was calculated as follows: the required minimal complexity (500×) multiplied by library complexity (188,509 gRNAs) divided by minimal surviving fraction of cells at an MOI of 0.3 (25%).

Cells were then selected with puromycin for 5 days and frozen or maintained at a minimal complexity of 500× (94 million cells after puromycin selection). To identify sgRNAs enriched under Reversine treatment, cells were expanded and seeded at 94 million cells per replicate (2 replicates per condition) with either DMSO (0.1%; Sigma) or 500 nM Reversine (R3904; Sigma-Aldrich) on 15-cm dishes (833903; Sarstedt) and cultured for 5 days with treatment after which the treatments were removed.

Cells were then maintained and passaged for two more weeks when necessary (DMSO-treated cells ~every 3 days, and Reversine-treated cells once or twice at later time points), always ensuring a minimal number of 94 million cells per replicate. The whole screen (2 conditions, 2 replicates per condition) was performed twice with a small variation between screens: in the first iteration, cells were treated with doxycycline to induce iCas9 directly following transduction, while at the second iteration doxycycline was added 5 days after DMSO or Reversine were added to the cultures. At the end of the screen, DNA was isolated from a minimum of 94 million cells for each replicate to determine sgRNA distribution.

Genomic DNA was isolated from samples using the QIAamp DNA Blood Maxi Kit (Qiagen). DNA was fragmented using NdeI and SacII restriction enzymes (New England Biolabs) and hybridized overnight with biotinylated capture oligos for selective DNA targeting. The hybridized DNA was captured using Streptavidin T1 Dynabeads (ThermoFisher) and non-hybridized oligos were removed by Exonuclease I digestion. sgRNAs were amplified by two rounds of PCR. The first round used unique barcoded forward primers, allowing for sample identification in pooled sequencing, while the second round attached necessary adapter sequences for sequencing compatibility. PCR products were purified using the NucleoSpin Gel and PCR Clean-up kit (Machery-Nagel). Sequencing was performed on a NextSeq 500 sequencer (Illumina).

sgRNA enrichment was determined using DESeq2, followed by the MAGeCK ranking aggregation method (Robust Rank Algorithm (RRA) (Love et al, 2014; Li et al, 2014), with multiple testing corrections by the Benjamini–Hochberg method. Our protocol was developed by adapting and refining methodologies from the study by Vera E. van der Noord et al (van der Noord et al, 2023).

### Computational analysis of sequencing data

The raw sequencing data (fastq files) were first sorted into separate files using sample-specific barcodes (the first six bases of the read sequences contained the barcodes; Perl script). Another Perl script was subsequently used to obtain the sgRNA counts from the individual (sample-specific) fastq files. This script searches for a key sequence (CGAAACACC), which precedes the sgRNA sequences. This key sequence was detected within a certain window (bases 16-44). When found, it extracts the sgRNA sequences which are the 20 bases just after the key sequence or the 20 bases after a G base that sometimes follows the key sequence. When one of the two sequences matched with the list of known sgRNA sequences, the

count of this sgRNA was raised by one. The sgRNA sequences were obtained from the AddGene website ((Wang et al, 2017), https://www.addgene.org/pooled-library/sabatini-crispr-mouse-high-activity-two-plasmid-system/). Next, a differential analysis was performed on the sgRNA level (counts) between the two conditions (Reversine and DMSO) with DESeq2 (Love et al, 2014). A paired design was implemented to account for differences between the samples while estimating the effect due to the condition (design = ~ replicate + condition). The median of ratios method was used for the normalization of the sgRNA counts. This analysis resulted in a log2 fold change value, a P value and a test statistic for each sgRNA. sgRNAs with a zero count in all samples were excluded. The results were subsequently ranked on the test statistic putting either the most enriched or depleted sgRNA on top. MAGeCK's RRA tool (van der Noord et al, 2023) was subsequently used to find genes that were consistently ranked better than expected under the null hypothesis of uncorrelated inputs (Settings: -permutation 100; -p [maximum percentile] = proportion of sgRNAs with positive or negative test statistic [depending on what is tested; positive or negative selection] and P value < = 0.25). MAGeCK's RRA tool uses the Benjamini–Hochberg method for multiple testing corrections (Benjamini and Hochberg, 1995).

## Computational analyses and statistics

For analysis of gene set enrichment within the CRISPR screen ranking results, enrichment was measured using the original GSEA method (Subramanian et al, 2005) (based on the estimated log-fold-change), which estimates the concentration of each gene set in the list of up-and down-regulated genes. We used the GSEA implementation in the GSEA-MSigDB website (https://www.gsea-msigdb.org/gsea/msigdb) to identify pathways enriched in the top 10% ranking genes of both CRISPR screens. The collections of gene sets used were the "KEGG", "Hallmark" and "GOBP" gene set collections from MSigDB v.2023.2.Mm (Liberzon et al, 2011).

For analysis of the top genes enriched in both CRISPR screens, the top 25% of genes of each screen were intersected and their significance was compared.

mRNA and protein expression datasets were obtained from DepMap release 22Q2 (Tsherniak et al, 2017). SAC genetic dependency data were obtained from the Achilles genome-wide RNAi screen (release 22Q2), and drug sensitivity data were obtained from the PRISM repurposing primary screen release (23Q2), both available on the cancer DepMap. For single-gene mRNA expression analyses, mRNA expression Z-score were calculated. For multiple-gene mRNA expression analyses, a gene set enrichment analysis (ssGSEA) score was assigned for each signature. Aneuploidy scores (AS) for all cell lines were obtained from Cohen-Sharir et al 2021. General analyses were performed on the ~1700 cell lines documented in the DepMap. Ploidy-centered analyses were performed on a subset of ~1000 cells for which an aneuploidy score was previously calculated.

The DepMap cancer cell lines were split into two groups of near-diploid and highly aneuploid cell lines, correlating to the quartiles with the bottom and top aneuploidy scores. Two-sided t tests were used to compare gene expression and drug sensitivity between the groups. The expression of CDC20 was removed from the drug sensitivity as a linear covariate using the partialize method in R (https://www.rdocumentation.org/packages/purrr/versions/0.2.4/

topics/partial). The cells were also split into groups of the top and bottom CDC20 expression quartiles, and two-sided t-tests were used to compare drug sensitivity between the groups.

Statistical analysis for viability/proliferation and comparison of fluorescence intensity levels between multiple groups were performed either by two-sided t tests (when two groups were compared) or by one-way ANOVA (when more than two groups were compared). All statistical analyses were performed in GraphPad Prism 9.1.

## CDC20–drug response association analysis

For gene–drug response association analysis, the expression of each of ~19,000 genes (DepMap release 22Q2) was correlated with the sensitivity to each of ~6500 drugs (PRISM primary repurposing screen, depmap release 23Q2) across ~1400 cell lines. For each gene, the sensitivity of the cell lines in its top quartile of expression was compared to that of its bottom quartile of expression using a Student's t test. The genes were ranked according to the resultant P values and the distribution of the percentile of the CDC20 ranking was plotted. The average ranking for MPS1 inhibitors was compared to that of all drugs.

## Tissue culture

3T3-NIH and 293FT cell lines were procured from the American Type Culture Collection (ATCC). HCT116 and RPE1 cells, and their aneuploid derivatives HPT1, HPT2, RPT1, RPT3, RPT4 included in this study were derived as described in Kuznetsova et al (Kuznetsova et al, 2015). All cells were cultured in Dulbecco's Modified Eagle Medium (DMEM; Gibco) supplemented with 10% fetal bovine serum (FBS; ThermoFisher Scientific or Sigma-Aldrich), 100 U/mL penicillin and 100 U/mL Streptomycin (P/S; Gibco). Cells were grown at 37 °C and 5% $CO_2$. For passaging and subculturing, cells were detached using either 0.25% Trypsin-EDTA (Life Technologies) or Tryple Express (Gibco).

## Lentiviral transduction

To produce lentiviruses, 293FT cells were transfected with 3 μg of the selected vector, complemented with essential packaging plasmids: 3 μg of pSPAX2 and 1 μg of pMD2.G. Notably, pSPAX2 (Addgene plasmid #12260) and pMD2.G (Addgene plasmid #12259), both gifts from Didier Trono. Forty-eight hours after transfection, the medium from the 293FT cells was harvested, filtered through a 0.45-μm filter (VWR Science), and then directly added to the intended cells in the presence of 8–10 μg/ml polybrene (TR-1003-G, Sigma-Aldrich).

## shRNA mediated knockdown

All shRNA sequences used in this work can be found in Table EV2. shRNAs were cloned into the Tet-pLKO-puro vector (Addgene plasmid #21915) (Wiederschain et al, 2009) using AgeI and EcoRI restriction enzymes (NEB) and verified by sequencing. Following lentiviral transduction, target cells were selected with puromycin (3–7 days) to isolate successfully transduced cells. Inducible shRNA expression was activated using 1 μg/mL doxycycline (hyclate D9891, Sigma-Aldrich) in the culture medium. Gene knockdown efficacy was confirmed by quantitative RT-PCR post three-day induction, and gene

knockout effectiveness was assessed via western blotting. The cellular impact of gene silencing was evaluated using various assays, including a five-day crystal violet staining protocol.

## CDC20 overexpression

The plasmids PMJ 031, PMJ 039, and PMJ 040 were generously provided by Iain Cheeseman's lab (Tsang and Cheeseman, 2023). PMJ 031 served as the empty vector control, while PMJ 039 and PMJ 040 contained the main isoform (isoform1) and the secondary enriched isoform (isoform2, truncated isoform starting at Met43) of CDC20, respectively. Isoform3, which is upregulated in the HEK293 overexpression attempt, is a truncated CDC20 isoform starting at Met88. All plasmids were constructed on a pBABE retroviral backbone. To generate retroviruses, PMJ 031, PMJ 039, and PMJ 040 constructs were used as previously described (Tsang and Cheeseman, 2023).

For the rtTA system, lentiviruses were produced by transfecting 293FT cells with the pLVX Tet-On Advanced vector, complemented by essential packaging plasmids and harvest the supernatant as previously mentioned. The viral supernatant was collected and added directly to target cells (HCT116, HPT1, HPT2) in the presence of 8–10 μg/ml polybrene (Sigma-Aldrich, TR-1003-G). After five days of transduction, neomycin selection (Sigma, N1142) was applied for approximately two weeks.

For inducible CDC20 construct generation, empty RES-eGFP or various CDC20-IRES2-eGFP fragments were cloned into the pLVX-tight-Puro vector using AgeI and EcoRI restriction enzymes (NEB). Constructs were verified by sequencing. Following transduction, target cells were selected with puromycin for 3–7 days. Induction of CDC20 gene expression was initiated with 1 μg/mL doxycycline (Sigma-Aldrich, D9891) in the culture medium. The efficacy of gene overexpression was confirmed by western blot analysis at specified time points post-induction.

## siRNA mediated knockdown

Custom siRNAs targeting CDC20 (5′-CGGAAGACCUGCC-GUUACAUU-3′), and a non-targeting control pool (D-001206-13) were obtained from Dharmacon. siRNAs were applied at a final concentration of 50 nM for HPT1 and HPT2 cell lines, and 25 nM for HCT116 cells, unless otherwise specified in the figure legend. A total of 2.5 μl Lipofectamine RNAiMAX (Thermo,13778075) was used per ml of the final transfection medium. For time-lapse microscopy analyses, assays were performed 20 h post-transfection, followed by CIN induction. For EdU assays, western blot and qPCR analyses, experiments were conducted 24 h post-transfection.

## Induction of chromosomal instability

HCT116 cells were treated with 125 nM Reversine for 24 h, and cultured for an additional 36 following Reversine wash-off. These CIN-induced cells were subsequently used for various experiments. For time-lapse microscopy assays, cells were seeded and subjected to siRNA treatment. Twenty hours post-siRNA treatment, Reversine or DMSO was added for imaging. For EdU assays, cells were seeded, treated with siRNA, and 20 h later, Reversine was added. Cells were then incubated for 2 h with EdU and harvested for analysis. For western blot and qPCR analyses, cells were treated with siRNA and treated with Reversine 20 h later. 24 h later, the cells were then harvested for analysis.

## Time-lapse imaging

Chromosomal abnormalities were quantified using time-lapse imaging. For this, cell lines were transduced with lentiviral H2B-mCherry constructs. One day prior to imaging, $3.2 \times 10^5$ 3T3-NIH cells, $6.8 \times 10^5$ HCT116 cells, $4 \times 10^5$ HCT116 CIN-induction cells and $3.2 \times 10^5$ cells for both HPT1 and HPT2 were seeded into imaging disks (Greiner Bio-One, catalog #627870). Imaging was performed on a DeltaVision Elite microscope (GE Healthcare), fitted with a CoolSNAP HQ2 camera and a ×40, 0.6 NA immersion objective lens (Olympus) for at least 20 h with images captured every 6 min. Each imaging stack consisted of 30–40 Z-stacks, at 0.5 μm apart. Images were analyzed using ICY software (Institut Pasteur). Only mitotic cells were included in the analyses. For live-cell imaging, the cells were pretreated with the drug one hour before initiating the imaging sessions.

## Scoring system for chromosomal instability resulting from missegregation

To quantify CIN phenotypes, we phenotypes were scored according to previously defined standards (Thu et al, 2018; Crozier et al, 2022; Huis in 't Veld et al, 2019). A minority of chromosomal events, representing diverse chromosomal aberrations and anomalies that did not align with standard categories were classified as "other". Each of these "other" events were scored from 1 to 5, based on its severity and complexity, relative to the predefined categories. However, due to their atypical nature, 'other' events were excluded from the main statistical analysis. The missegregation events in Figs. 4C–L and EV5C,D are colored by score, with specific events within the same scoring category marked with different patterns.

0 points (green): Correct chromosomal Segregation.

1 point (blue): DNA Bridge Formation, Micronucleus Formation, Chromosomal Lagging.

2 points (yellow): DNA Bridge/Lagging with Micronucleus Formation, Metaphase Misalignment.

3 points (orange): Metaphase Misalignment with Micronucleus Formation, Metaphase Misalignment with Chromosomal Lagging/Bridge Formation.

4 points (purple): Metaphase Skipping, Cytokinesis Failure, Metaphase misalignment with chromosomal lagging/bridge formation and emergence of micronuclei.

5 points (red): Metaphase Skipping with DNA Bridging, Metaphase Skipping with Micronucleus Formation.

Other (Variable 1–5 points, gray): Chromosomal aberrations or anomalies that do not fit into the predefined categories.

For each sample, aberrations of categories 0–5 were grouped by score, and the differences in their distribution across samples were calculated using one-sided Kruskal–Wallis tests performed in GraphPad Prism 9.1. Note that we define two related categories for errors in metaphase: metaphase misalignment and metaphase skipping. The former refers to a chromosome that is displaced outside of the metaphase plate while other chromosomes are in the metaphase plate, whereas the latter refers to cells that fail to form a metaphase plate altogether before segregating their chromosomes in (an aberrant) anaphase.

## Quantification of proliferation using EdU incorporation

For EdU assays, cells were cultured on coverslips in 24-well plates and treated with 125 nM (or 250 nM) Reversine or DMSO for 48 h. To determine the fraction of cells in S-phase, cells were pulse-labeled with 10 μM EdU for 2 h. Cells were fixed in 4% formaldehyde, and EdU was detected using Click chemistry and (2 mM Cu(II)SO$_4$, 4 μM sulfo-Cy3-azide, and 20 mg/ml sodium ascorbate in PBS). EdU incorporation was visualized by fluorescence microscopy (Olympus IX51 or Olympus BX43), and images were processed and quantified using the Fiji software (ImageJ 1.53C).

## Western blotting

For Western blotting, cells were lysed in RIPA (50 mM Tris-HCl (pH 8.0), 1% NP-40, 0.5% Sodium Deoxycholate, 0.1% SDS, 150 mM NaCl (5 M), dilute with ddH$_2$O), supplemented with a Roche protease inhibitor cocktail for 30 min. In total, 20–30 μg of protein was loaded on 7.5–10% polyacrylamide gels and transferred to PVDF membranes. Membranes were blocked with Odyssey blocking buffer or 5% milk in TBST. Primary antibodies used were CDC20 (Abcam, ab26483, 1:1000 or Santa Cruz, sc-13162, 1:500), β-Actin (Cell Signaling, 4970 or 3700S, 1:2000), GAPDH (Cell signalling D16H11, 1:1000), and Vinculin (Cell Signalling, 4650S, 1:1000) with incubation overnight at 4 °C. Secondary antibodies (IRDye 800CW Goat anti-Rabbit IgG (H + L) (Licor; 1:15,000) or Goat anti-Rabbit (Jackson lab, 111-035-003, 1:10,000) and IRDye 680CW Goat anti-Mouse IgG (H + L) (Licor; 1:15,000) or Goat anti-Mouse (Jackson lab, 115-035-003, 1:10,000) were incubated for 1 h at room temperature. Blots were visualized on an Odyssey imaging system (LI-COR Biosciences), in combination with Image Studio Lite software (LI-COR Biosciences) or using chemiluminescence (Millipore #WBLUR0500) on a UVITEC machine. The quantitative analysis of western blot bands was performed by ImageJ v1.53 or by NineAlliance v18.12. For bulk CDC20 quantification by western blotting HC116-HPT cells were seeded in 6-cm plates at 80% confluence and treated with 330 nM Nocodazole (Sigma-Aldrich) for 20 h before collection.

For CDC20 quantification after aneuploidy induction, cells were plated in a 10-cm plate and, 24 h later, treated with 250 nM Reversine or DMSO for 24 h. After 24 h of drug washout, cells were treated with 250 nM nocodazole for 20 h, and mitotic cells were harvested by shake-off. Western blot bands were quantified using FIJI software. The fold change of the Reversine-treated samples relative to DMSO was calculated by normalizing CDC20 to pH3Ser10, which was previously normalized to vinculin.

## Cell cycle flow cytometry analysis

HCT116-HPT cells were seeded in 6-cm plates at 80% confluence, treated with 330 nM Nocodazole (Sigma-Aldrich) for 20 h and collected by using 0.25% Trypsin-EDTA (Life Technologies). Cells were then washed with PBS and fixed in 70% ethanol for 30 min. Cells were stained with 50 μg/mL PI (BioLegend) for 10 min, then analyzed in CytoFlex (Beckman and Coulter).

## Real-time quantitative PCR

For qPCR analysis, RNA was extracted from cell pellets using the RNA plus isolation kit (Qiagen, MN 740984.250). For cDNA synthesis, 1.5 μg of RNA was used with the LunaScript RT SuperMix Kit (Bioke, M3010X) in a 20 μl reaction. The synthesized cDNA was then quantified by quantitative PCR (qPCR) using iTaq Universal SYBR Green supermix (BioRad, 1725124) on a Light-Cycler® 480 Instrument. Relative RNA levels were calculated in Excel (Microsoft) and plotted using Prism software (GraphPad).

## Immunofluorescence microscopy

For immunofluorescence imaging, HCT116-HPT cells were grown on glass coverslips in 24-well plates at a density of $2 \times 10^5$. Cells were synchronized at the G2/M transition with 9 nM HCT-HPT of the CDK1 inhibitor RO-3306 (Sigma-Aldrich) for 20–21 h, released from arrest and incubated for 45–55 min at 37 °C. Cells were visually followed until metaphase by H2B-GFP staining in an Axio Imager Z microscope (Carl Zeiss) and fixed in 4% paraformaldehyde for 15 min. Fixed cells were incubated for 15 min with fresh 0.1 M Glycine to prevent quenching, then for 5 min with a 0.5% Triton X-100 solution to permeabilize cells. Cells were blocked for 30 min (BSA, Glycine, NaCl and 0.1% Triton X-100) and stained with primary antibodies against CDC20 (1:50, Santa Cruz) and the mitosis marker H3ser10p (1:400, Cell Signaling) in blocking buffer. Used secondary antibodies were anti-mouse Alexa-555 antibody (1:400, Cell Signaling) and an anti-rabbit Alexa-488 (1:400, Cell Signaling). Both primary and secondary antibodies were incubated for 1 h in a humidified environment at room temperature. Images were acquired using CellSens Imaging Software (Olympus) at a 40x resolution. Cells at metaphase were identified by H3ser10p staining and CDC20 intensity per area was quantified using ImageJ v1.53 in single cells. CDC20 levels were later compared in bulk between the isogenic cell lines in One-way ANOVA tests performed in GraphPad Prism 9.1.

## Chromosome alignment analysis

For chromosome alignment analysis, HCT116-HPT cells were seeded onto coverslips coated with 5 μg/ml Fibronectin (Sigma-Aldrich) at 60% confluence. The cells were synchronized with 7.5 μM RO-3306 (Sigma-Aldrich) for 16 h at 37 °C. After synchronization, cells were washed three times with 1× PBS and fixed 45 min later. For chromosome alignment analysis in the presence of MG132, cells were released from RO-3306 and 30 min later treated with 10uM MG-132 (Tocris) for 90 min at 37 °C and then fixed. Fixation was performed using 4% paraformaldehyde (in PBS) for 15 min at room temperature, followed by blocking in 5% BSA in PBS for 30 min. Then, cells were incubated with the following antibodies for 90 min at room temperature: anti-CDC20 (sc-13162) 1:250, anti- phospho-Histone H3 (Ser10) (Sigma-Aldrich) 1:500, anti-centromeric antibody (Antibodies Incorporated) 1:100. Alexa 488-, Alexa Cy3- and Alexa 647-labeled secondary antibodies (Invitrogen) were used 1:400 for 45 min at room temperature. DAPI 1:5000 was used to stain DNA. Coverslips were mounted using Mowiol. Cells were acquired using Leica Histo-Fluo or a spinning disk microscope with a magnification objective of 100x. A focal plane was used for the analysis. FIJI software was used for image processing. CREST was used as a centromere marker to evaluate the proper alignment of chromosomes to the metaphase plate. Chromosomes with CREST signal not present in the metaphase plate were scored as mis-aligned (Ippolito et al, 2021, 2024).

## Data availability

All datasets generated and analyzed in the CRISPR screen are available in the European Nucleotide Archive (ENA) under accession ID PRJEB71335. This includes detailed sample data for the CRISPR screen in NIH-3T3 cells, using a sgRNA library for sgRNA enrichment analysis. The availability of these datasets ensures transparency and facilitates replication of our findings, adhering to the open science principles of the University Groningen/University Medical Center Groningen. The computer code used for analysis is this work is available at https://github.com/bendavidlab/cdc20.

The source data of this paper are collected in the following database record: biostudies:S-SCDT-10_1038-S44319-024-00363-8.

## Peer review information

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

## Acknowledgements

The authors are grateful to Roderick Beijersbergen and Cor Lieftink (Netherlands Cancer Institute, Amsterdam, the Netherlands) for providing scripts and advice for the analysis of the CRISPR screen data; and to Zuzana Storchova (RPTU Kaiserslautern, Germany) for kindly providing the HPT1, HPT2, RPT1, RPT3, and RPT4 cell lines. Siqi Zheng and Lin Zhou were supported by personal fellowships from the Chinese Scholar Council (CSC). This work was further supported by two Dutch Cancer Society grants to Foijer (2015-RUG-7822 and 2022-4EXPL-14805; ScreeninC). Work in the Ben-David Lab was supported by the DoD CDMRP Career Development Award (grant #CA191148 to UB-D), the European Research Council Starting Grant (grant #945674 to UB-D), the Israel Cancer Research Fund Project Award (UB-D), the Azrieli Foundation Faculty Fellowship (UB-D), the Israel Science Foundation (grant #1805/21 to UB-D), the BSF Project Grant (grant #2019228 to UB-D), and the Israel Cancer Association (grant #20230018 to UB-D). Work in the Santaguida lab was supported by the Italian Association for Cancer Research (AIRC-MFAG 2018—ID. 21665 and Bridge Grant 2023—ID. 29228, projects to SS), Ricerca Finalizzata (GR-2018-12367077 to SS), Fondazione Cariplo (SS), the Rita-Levi Montalcini program from MIUR (to SS), and the Italian Ministry of Health with Ricerca Corrente and 5×1000 funds (SS). Marica R. Ippolito was supported by an AIRC Fellowship (ID 26738-2021). A subset of the figures and text were included in the PhD thesis by Dr. Siqi Zheng (Zheng, 2024; https://doi.org/10.33612/diss.993723132) and in the MSc thesis by Linoy Raz (Raz, 2024). Schematic illustrations were created using https://www.biorender.com.

## Author contributions

**Siqi Zheng**: Data curation; Formal analysis; Validation; Investigation; Visualization; Methodology; Writing—original draft. **Linoy Raz**: Data curation; Formal analysis; Validation; Investigation; Visualization; Methodology; Writing—original draft. **Lin Zhou**: Data curation; Formal analysis; Validation; Investigation; Visualization; Methodology; Writing—review and editing. **Yael Cohen-Sharir**: Formal analysis; Validation; Investigation; Visualization; Methodology; Writing—review and editing. **Ruifang Tian**: Formal analysis; Validation; Investigation; Visualization; Methodology; Writing—review and editing. **Marica Rosaria Ippolito**: Formal analysis; Validation; Investigation; Visualization; Methodology; Writing—review and editing. **Sara Gianotti**: Formal analysis; Validation; Investigation; Visualization; Methodology; Writing—review and editing. **Ron Saad**: Data curation; Software; Formal analysis; Validation; Investigation; Visualization; Methodology; Writing—review and editing. **Rene Wardenaar**: Data curation; Software; Formal analysis; Validation; Investigation; Visualization; Methodology; Writing—review and editing. **Mathilde Broekhuis**: Formal analysis; Validation; Investigation; Visualization;

Methodology; Writing—review and editing. **Maria Suarez Peredo Rodriguez**: Formal analysis; Validation; Investigation; Visualization; Methodology; Writing —review and editing. **Soraya Wobben**: Formal analysis; Validation; Investigation; Visualization; Methodology; Writing—review and editing. **Anouk van den Brink**: Formal analysis; Validation; Investigation; Visualization; Methodology; Writing—review and editing. **Petra Bakker**: Formal analysis; Validation; Investigation; Visualization; Methodology; Writing—review and editing. **Stefano Santaguida**: Formal analysis; Supervision; Funding acquisition; Investigation; Methodology; Writing—review and editing. **Floris Foijer**: Conceptualization; Formal analysis; Supervision; Funding acquisition; Validation; Investigation; Visualization; Methodology; Writing—original draft; Project administration. **Uri Ben-David**: Conceptualization; Formal analysis; Supervision; Funding acquisition; Validation; Investigation; Visualization; Methodology; Writing—original draft; Project administration.

Source data underlying figure panels in this paper may have individual authorship assigned. Where available, figure panel/source data authorship is listed in the following database record: biostudies:S-SCDT-10_1038-S44319-024-00363-8.

## Disclosure and competing interests statement

UB-D is a consultant for Accent Therapeutics. FF is the scientific director (Chief Scientific Officer) for iPsomics, which is unrelated to this work. The remaining authors declare no competing interests.

# Expanded View Figures

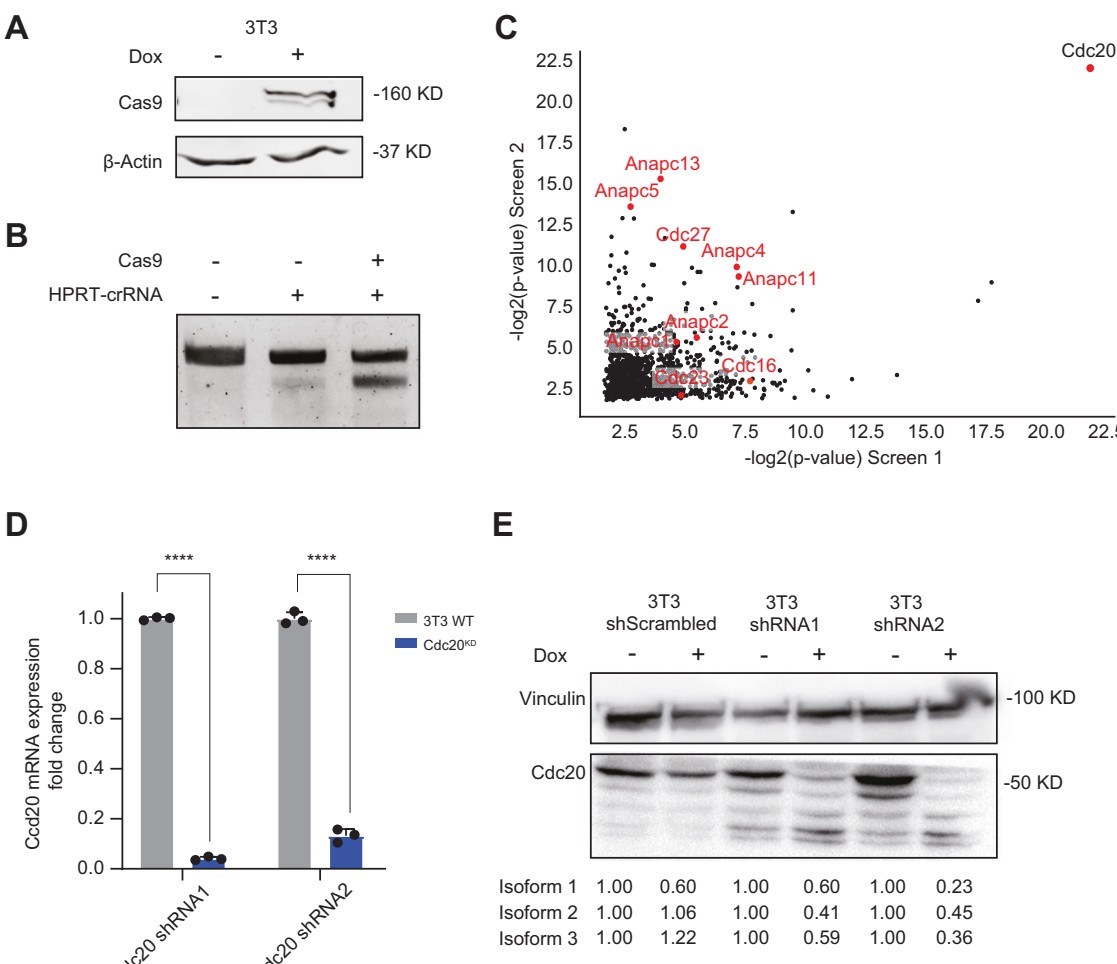

**Figure EV1. CDC20 is strongly associated with resistance to SAC inhibition.**

(A) Western blot validation for Cas9 expression in 3T3 cells used for the CRISPR screen. (B) Cleaved and uncleaved PCR product in a T7 assay as a readout of Cas9 activity in 3T3 cells used for the CRISPR screen. (C) Correlation between the top-ranked 25% of genes in both CRISPR screens based on their statistical significance with all APC/C-related genes highlighted, showing that Cdc20 is by far the most significant outlier of all APC/C extended complex members. *P* values were calculated using the RRA method (see "Methods"). (D, E) qPCR (D) or western blot (E) validation of Cdc20 knockdown by shRNA. Paired *t* test (*N*, number of biological replicates; *N* = 3; ****, *P* value < 0.0001). Error bars represent the standard deviation (SD) of the mean. Source data are available online for this figure.

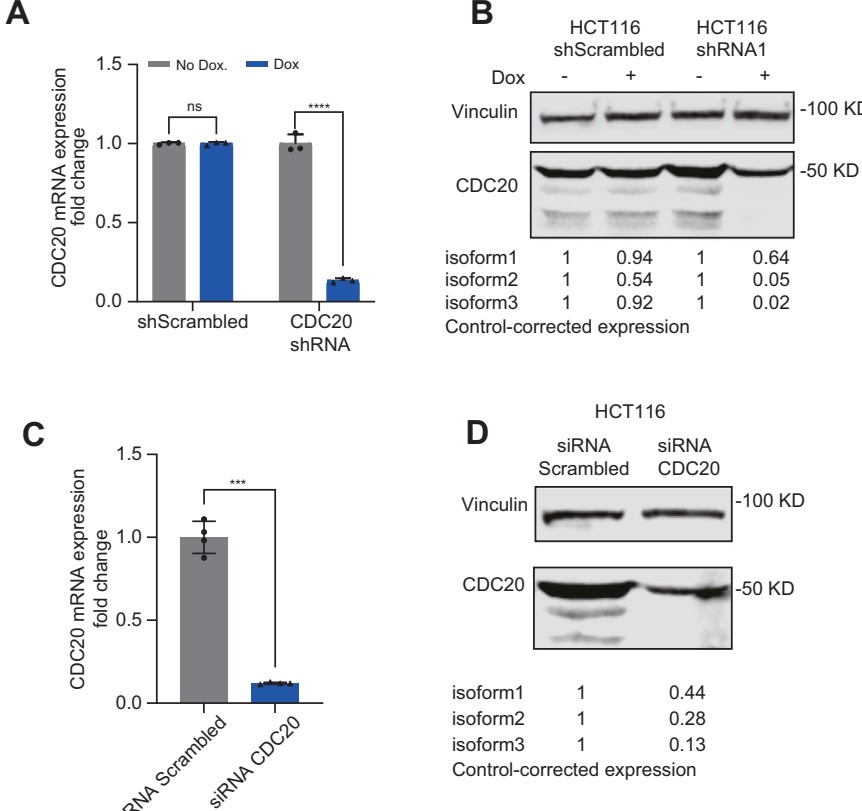

**Figure EV2. CDC20 expression predicts sensitivity to genetic and chemical SAC perturbation.**

(A, B) qPCR (A) and western blot (B) validation of CDC20 knockdown by shRNA in HCT116 cells. Two-sided paired *t* test (*N*, number of biological replicates; *N* = 3; ns, *P* value = 0.9999; ****, *P* value < 0.0001). Error bars represent the standard deviation (SD) of the mean. (C, D) qPCR (C) and western blot (D) validation of CDC20 knockdown by siRNA in HCT116 cells. Two-sided paired *t* test (*N* = 4; ***, *P* value = 0.001). A representative image is shown. Quantification values of the individual bands show the average value of all replicates. Source data are available online for this figure.

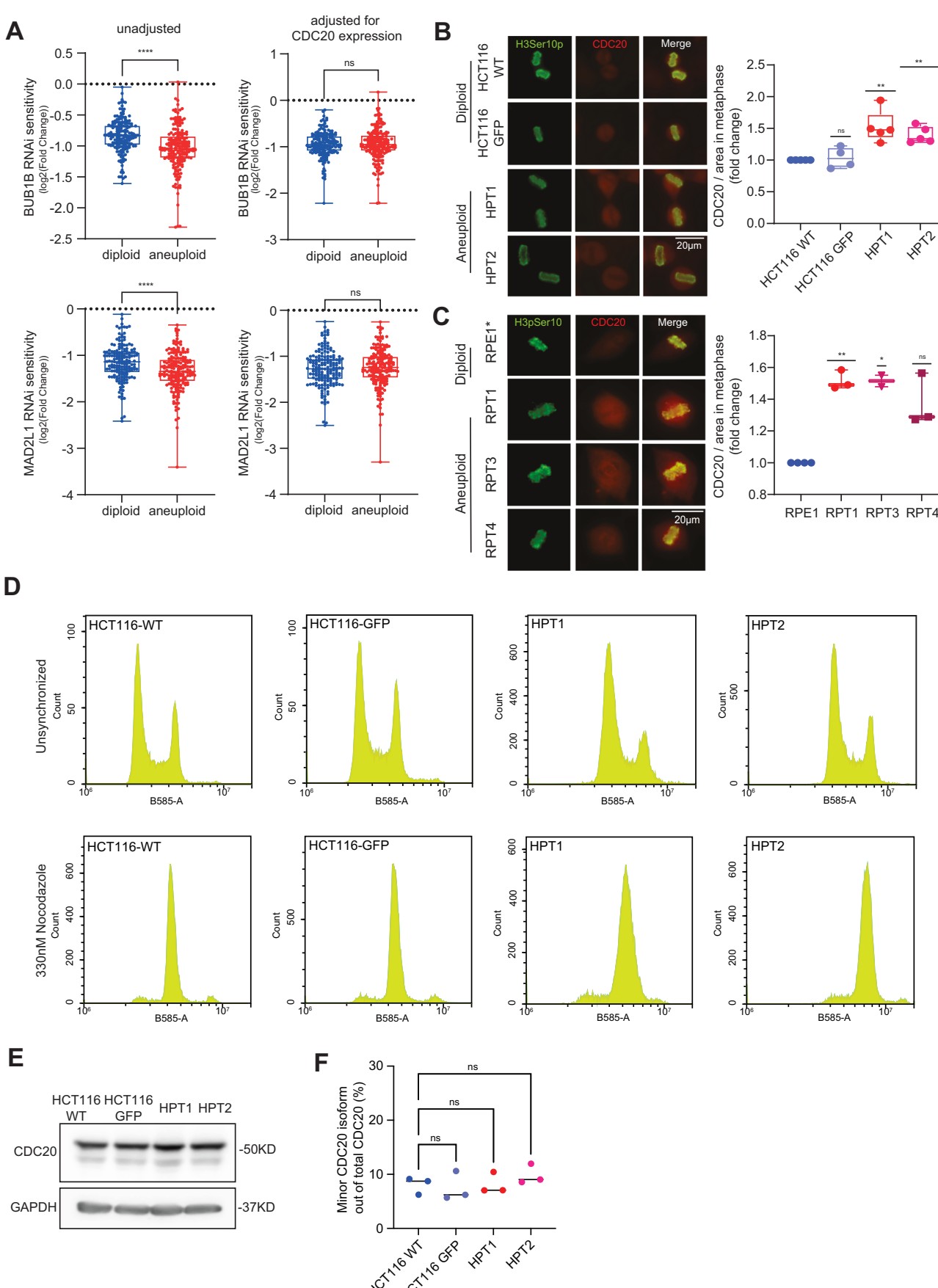

**◀ Figure EV3. Increased CDC20 expression is associated with the preferential response of aneuploid cells to SAC inhibition.**

(A) Left - correlation between ploidy and the sensitivity to genetic perturbation of the core SAC components BUB1B (top) and MAD2L1 (bottom). Right – same correlation with CDC20 expression removed as a linear covariate (see "Methods"). When removing the effect of CDC20 expression the trend reverses or becomes insignificant. Box plots show the median (middle line) and interquartile range (IQR; box edges). Whiskers extend to values within 1.5×IQR from the 25th and 75th percentiles. Two-sided *t* test (ns, *P* value = 0.5302 for MAD2L1 RNAi and 0.8019 for BUB1B RNAi; ****, *P* value < 0.0001). (B, C) Representative images (left) and single-cell quantification (right) of CDC20 at metaphase in cells of the HCT116-HPT system (B) or the RPE-RPT system (C) after synchronization with 9 nM or 4.5 nM RO-3306 (respectively) for 20 h. Highly aneuploid cells express higher levels of CDC20 than their diploid counterparts. In (B), box plots show the median (middle line) and interquartile range (IQR; box edges). Whiskers extend to values within 1.5×IQR from the 25th and 75th percentiles. In (C), bars represent the data range. One-sample *t* test (*N*, number of biological replicates; *N* = 5 or *N* = 4 respectively; ns, *P* value = 0.7017 (B) or *P* value 0.0583 (C); *, *P* value = 0.0447, **, *P* value = 0.0093 or *P* value = 0.0023 (B, left to right) and *P* value = 0.0046 (C)). (D) Flow cytometry analysis showing the cell cycle distribution of HCT116-HPT cells in an unsynchronized state (top) or after synchronization with 330 nM Nocodazole for 20 h (bottom), the same conditions that were used for the bulk CDC20 quantification in Fig. 3D. All four cell lines are synchronized to a similar extent, allowing for a bulk comparison of a cell cycle protein expression. (E) Representative western blot image of bulk CDC20 expression in the HCT116-HPT cell line set following synchronization with 330 nM Nocodazole. Aneuploid cells express higher levels of CDC20 than their diploid counterparts. (F) Percent of minor CDC20 isoform out of total CDC20, as observed in bulk quantification. There is no significant difference in the fraction of minor isoform between the diploid and aneuploid cell lines. One-way ANOVA (*N*, number of biological repeats; *N* = 3; ns, *P* value = 0.9781 or *P* value = 0.9996 or *P* value = 0.5945 (from left to right)). Source data are available online for this figure.

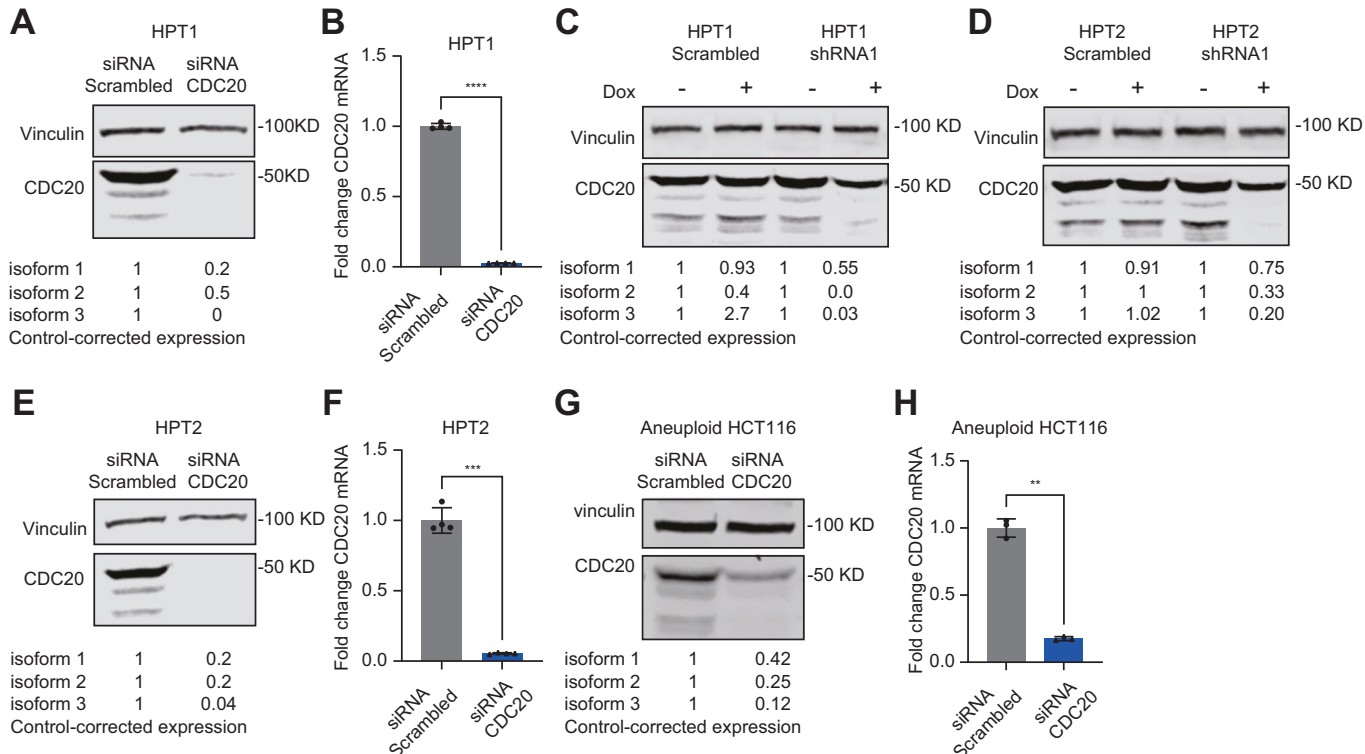

**Figure EV4.  Validation of CDC20 knockdown in aneuploid cell lines.**

(A, B) Western blot (A) and qPCR (B) validation of CDC20 knockdown by siRNA in HPT1 cells. Two-sided *t* test (*N*, number of biological replicates; *N* = 4; ****, *P* value < 0.0001). Error bars represent the standard deviation (SD) of the mean. A representative image is shown; Quantification values of the individual bands show the average value of all replicates. (C, D) Western blot validation of CDC20 depletion by shRNA in HPT1 (C) and HPT2 (D). (E, F) Western blot (E) and qPCR (F) validation of CDC20 knockdown by siRNA in HPT2 cells. Two-sided *t* test (*N*, number of biological repeats; *N* = 4; ***, *P* value = 0.0002). (G, H) Western blot (G) and qPCR (H) validation of CDC20 knockdown by siRNA in aneuploid HCT116 cells. Two-sided *t* test (*N* = 3; **, *P* value = 0.0034). Source data are available online for this figure.

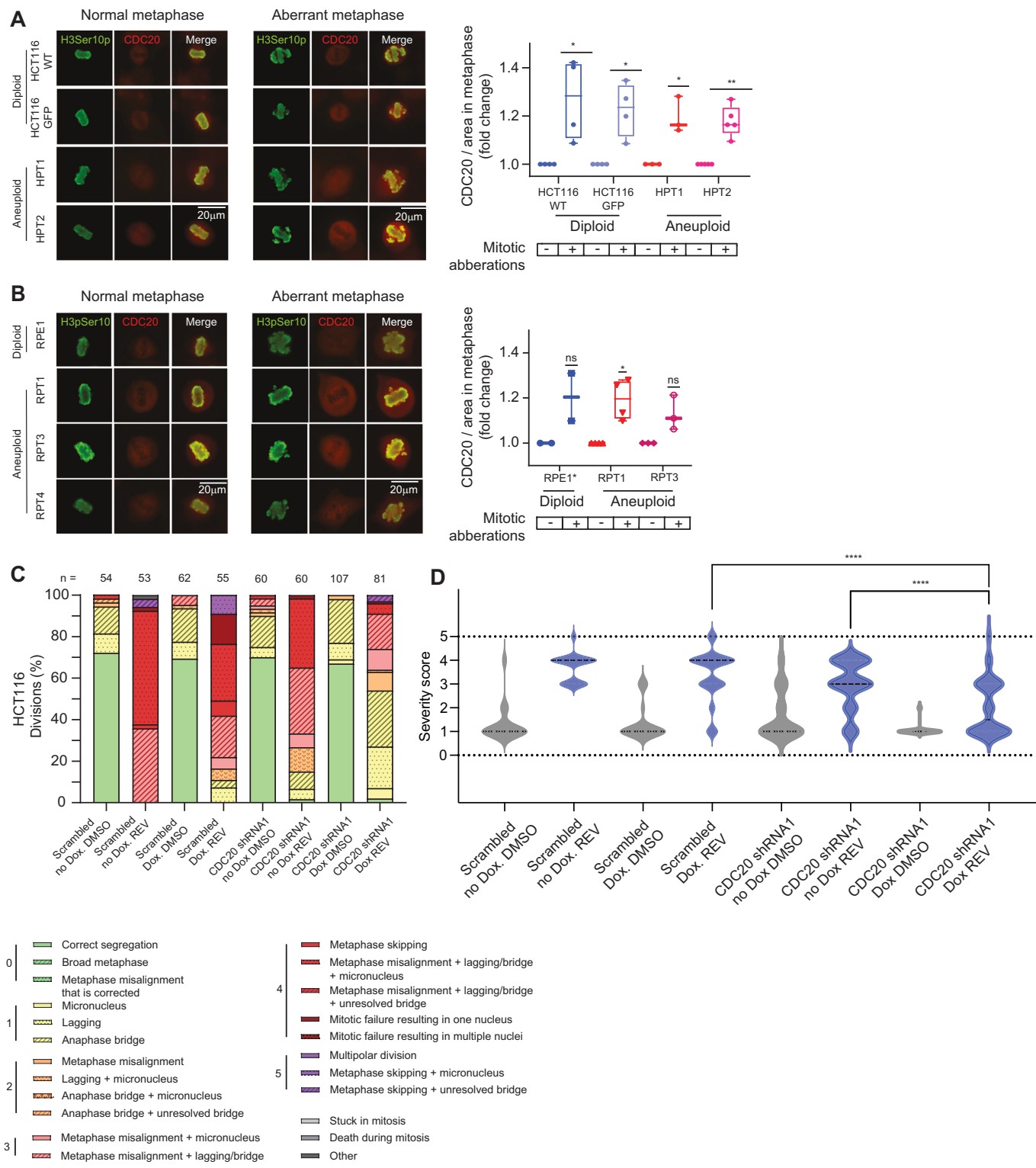

◀ **Figure EV5. CDC20 expression levels determine the prevalence of mitotic errors and metaphase duration.**

(A, B) Representative IF images (left) and quantification (right) of CDC20 in HCT116-HPT cells (A) or RPE-RPT cells (B) undergoing normal or aberrant mitoses after synchronization with RO-3306. Cells with mitotic aberrations express significantly higher levels of CDC20 during metaphase than cells undergoing normal division, regardless of ploidy background. In (A), Box plots show the median (middle line) and interquartile range (IQR; box edges). Whiskers extend to values within 1.5×IQR from the 25th and 75th percentiles. In (B), bars represent the data range. One-sample $t$ test ($N$, number of biological replicates; $N = 5$ (A) and $N = 4$ (B); ns, $P$ value $= 0.3037$ or $0.1028$ (B, from left to right); *, $P$ value $= 0.0498$ or $P$ value $= 0.0271$ or $P$ value $= 0.0465$ (A, left to right) and $P$ value $= 0.0233$ (B); **, $P$ value $= 0.0033$). (C, D) Distribution (C) and severity quantification (D) of mitotic abnormalities in HCT116 cells treated with 125 nM Reversine, under control conditions or CDC20 depletion by shRNA. Mitotic aberrations were identified by live-cell imaging and scored on a severity scale of 0–5, then grouped and colored by score. The changes in severity distribution across samples were assessed using one-sided Kruskal–Wallis tests, as elaborated in "Methods" section. Cells treated with Reversine exhibit reduced mitotic aberrations after CDC20 depletion. Sample size ($N$) in (D) corresponds to the sample size in (C). One-sided Kruskal–Wallis test (****, $P$ value $< 0.0001$). Source data are available online for this figure.

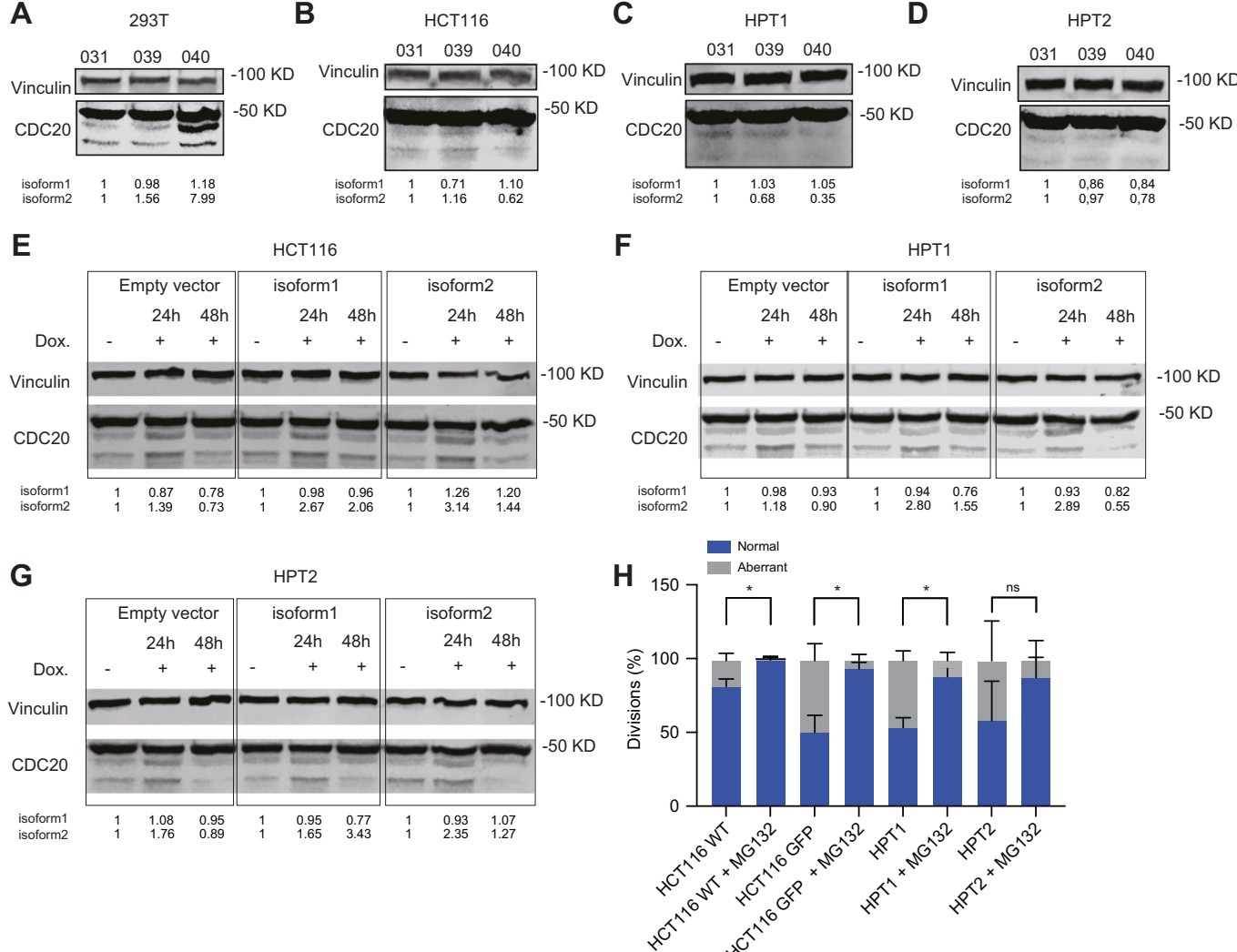

**Figure EV6. CDC20 overexpression attempts in HCT116-HPT cells.**

(A) Western blot quantification of CDC20 overexpression with plasmids from Tsang et al (Tsang and Cheeseman, 2023), in 293T cells. CDC20 is overexpressed in these cells. (B–D) Western blot quantification of CDC20 overexpression with plasmids from Tsang et al 2023, in HCT116 (B), HPT1 (C) and HPT2 (D) cells. No CDC20 overexpression can be detected in these cells. (E–G) Western blot quantification of CDC20 overexpression with lentiviral rTTA inducible system, in HCT116 (E), HPT1 (F) and HPT2 (G) cells. No CDC20 overexpression can be detected in this system either. (H) Rate of normal and abnormal cell divisions in the HCT-HPT system during normal (45 min) and prolonged (120 min) metaphases (see legend for Fig. 5G, "Methods"). The rate of mitotic aberrations is significantly decreased in cells undergoing prolonged metaphases. Two-sided paired *t* test (*N*, number of biological replicates; *N* = 3; ns, *P* value = 0.0929; *, *P* value = 0.0251 or *P* value = 0.0385 or *P* value = 0.0210 (from left to right)). Source data are available online for this figure.

