## [Peer Review File · EMBO Reports]

High CDC20 levels increase sensitivity of cancer cells to MPS1 inhibitors

Siqi Zheng, Linoy Raz, Lin Zhou, Yael Cohen-Sharir, Ruifang Tian, Marica Rosaria Ippolito, Sara Gianotti, Ron Saad, Rene Wardenaar, Mathilde Broekhuis, Maria Suarez Peredo Rodriguez, Soraya Wobben, Anouk van den Brink, Petra Bakker, Stefano Santaguida, Floris Foijer, and Uri-Ben David

Corresponding authors: Uri Ben-David (ubendavid@tauex.tau.ac.il) , Floris Foijer (f.foijer@umcg.nl)

Review Timeline:

Transfer Date:	4th Oct 24
Editorial Decision:	31st Oct 24
Revision Received:	13th Nov 24
Accepted:	2nd Dec 24

Editor: Bernd Pulverer

Transaction Report: This manuscript was transferred to EMBO reports following peer review at The EMBO Journal.

Referee #1:

The spindle assembly checkpoint (SAC) is an intriguing target for anti-cancer therapeutics as loss of SAC function can lead to the accumulation of intolerable levels of aneuploidy in dividing cells. However, there remains some controversy about whether there is a sufficiently therapeutic window (doi 10.1158/1078-0432.CCR-20-4185) as SAC-targeting drugs also impact all healthy proliferating tissues. Notably, Aurora B inhibitors suppress SAC function but have failed to progress in the clinic due to severe myelosuppression. Nevertheless, not all cells accumulate mitotic errors at the same rate in response to SAC inhibitors, and the success of this therapeutic approach will rely on the development of clear biomarkers for highly susceptible cancers.

Previous work has identified weak anaphase promoting complex/cyclosome (APC/C) function as a resistance mechanism to SAC inhibitors. Cells with reduced APC/C activity take longer to complete mitosis and have more opportunity for error correction before mitotic exit. In this manuscript, the authors use a combination of genetic and bioinformatic approaches to propose that CDC20, the co-activator of the APC/C complex, rather than the APC/C itself, controls the response to SAC inhibition. As such, CDC20 expression levels are proposed as a biomarker for SAC inhibitor-sensitive cancers.

The link between CDC20 expression and SAC inhibitor sensitivity is predictable and well-established by the authors in gene perturbation assays. However, the synthetic systems employed in the manuscript far over-represent the range of CDC20 expression in cancers. This is revealed in the data from the pan-cancer DepMap dataset which show only a very weak correlative relationship between CDC20 expression and sensitivity to SAC inhibition. The data shown in the paper reveal that CDC20 expression alone has little to no predictive power to identify cancers that are more sensitive to SAC perturbation. Therefore, while the authors have performed a clean and thoughtful analysis, they are ultimately unable to establish CDC20 as a bona fide clinically useful biomarker for SAC inhibitor sensitivity.

Specific Comments:

- The DepMap analysis in Figure 2 is the crux of the paper as it investigates the impact of CDC20 expression across a large panel of cancer cell lines. The data attempting to discriminate cancers sensitivity to SAC inhibition by CDC20 expression in this analysis is not sufficient to support the central claim of the paper. In these experiments, the correlation between CDC20 expression and sensitivity to MAD2L1/BUB1B disruption is very weak ($R^2 < 0.2$). This is not improved when comparing CDC20 expression versus drug sensitivity for two different MSP1 inhibitors MP1-0479605 ($R^2 = 0.15$) or AZ-3016 ($R^2 = 0.07$). Even segregating sensitivity using the top and bottom quartiles of CDC20 expression as performed in Figure 2C only yields a modest effect that could not be used to stratify sensitive versus insensitive tumors. Ultimately, the authors show that, as anticipated, strong genetic perturbations of CDC20 affect sensitivity, presumably through APC/C activation, but that this effect is not strong enough to be a clinically relevant biomarker. There are likely to be many factors that contribute to the sensitivity to SAC inhibition and CDC20 expression is one minor contributor.

- Polyclonal CDC20 "knockout" cells are not well defined. CDC20 is a common essential gene and therefore either CDC20 is not completely eliminated or there is some unexplored biology at play. These cells would benefit from further characterization such as clonal isolation to define the level of CDC20 remaining. In addition, recent work from the Cheeseman lab identified several translational isoforms of CDC20, but surprisingly there is no discussion of this work or how the sgRNAs used are expected to impact expression of some or all of these isoforms.
- The claim that APC/C activity independently from CDC20 expression does not affect SAC inhibitor sensitivity is not well supported. APC/C subunits came out of one of the initial screens refuting this claim and only APC4 is probed in the manuscript. Furthermore, depletion of APC4 was less effective than CDC20 in some critical experiments (Fig S3A-B) confounding these results. Finally, bioinformatically approximating APC/C complex activity through an averaged APC/C expression score has significant drawbacks. Notably, this approach masks the effect of concentration-limiting APC/C subunits during complex formation.
- It was not mentioned in the manuscript that the aneuploid derivatives of RPE1 and HPT cells are also whole genome doubled. This may confound the analysis performed in these cells.

Referee #2:

In this manuscript, Zheng et al identify Cdc20 as the top hit in two independent CRISPR-Cas9 screens to identify genes whose depletion confers resistance to inhibition of the spindle assembly checkpoint (SAC) kinase Mps1. Depletion of Cdc20 conferred resistance to Mps1 inhibition in 3T3, HCT116 and HCT116 post-tetraploid (HPT) cells. Previous screens to identify causes of resistance to Mps1 inhibition (Sansregret et al Cancer Discovery 2017, Thu et al PNAS 2018) identified APC/C subunits as contributing to response to Mps1 inhibition. However, Zheng et al report that targeting of the APC/C subunit ANAPC4 does not confer resistance to Mps1 inhibition in 3T3, HCT116 or HPT cells. Moreover, though expression of APC/C subunits correlates with sensitivity to SAC inhibition in the cancer Dependency Map, this is largely driven by expression of Cdc20, which is the APC/C cofactor that drives APC/C activity during mitosis. Targeting Cdc20 extends mitotic duration and decreases chromosome segregation errors after Mps1 inhibition, offering a mechanistic explanation for causing resistance, while targeting ANAPC4 largely does not. The authors conclude that increased Cdc20 expression is a promising biomarker for Mps1 inhibitors (and any other SAC inhibitors), while APC/C expression is not.

Predictive biomarkers of response substantially improve outcomes and are therefore highly sought after for both existing and emerging therapies. A biomarker for Mps1 inhibitors would be beneficial to include in clinical trials and would substantially aid in bringing these drugs to market. The broad focus on SAC inhibitors in general in the title and summary is somewhat odd, since the manuscript focuses exclusively on Mps1 inhibitors, and these are the only SAC inhibitors in or approaching the clinic to my knowledge. Nevertheless, the identification of Cdc20 mRNA expression as a simple potential biomarker of response to these drugs is a significant finding of broad interest that is well

supported by the data presented here. The companion argument that APC/C expression does not dictate response is less compelling and will require additional justification prior to publication.

Major concerns

1. The argument that APC/C expression does not contribute to Mps1 inhibitor resistance relies on depletion of single subunit of APC/C, ANAPC4. Though the manuscript refers to ANAPC4 knockout throughout, the available western blots (Fig S2B,D, S3B) clearly show that ANAPC4 has not been knocked out but is rather knocked down. Mechanistically (as described very nicely in the discussion), Mps1 inhibition is lethal because the mitotic checkpoint is unable to delay mitosis sufficiently to prevent large amounts of chromosome missegregation. Slowing mitosis allows extra time for chromosomes to make proper attachments to spindle microtubules, reducing chromosome missegregation and cell death. The authors show convincingly in Fig 4 that Cdc20 depletion slows mitosis and decreases chromosome missegregation in the Mps1 inhibitor reversine. However, mechanistically, decreasing expression of APC/C subunits should also slow mitosis and it has previously been demonstrated that sufficient ANAPC4 depletion slows mitosis (Thu et al PNAS 2018 Fig 3G). These experiments appeared to involve a more robust knockdown than achieved here (Thu et al Fig 3C, S3C, S4A). The simplest explanation for the lack of an effect here is that the knockdown of ANAPC4 is insufficient to confer the phenotype. Knockdown of at least a second component of the APC/C would also strengthen the argument.

2. Throughout the manuscript, Cdc20 is described as being knocked out. Though it is clear that a portion of Cdc20 remains in Fig S1E, S2G and S3A, the quantitation in Fig S1F indicates generation of 3T3 cell lines truly null for Cdc20. However, Cdc20 is well established as an essential gene in mice (Li et al, MCB 2017) and in cell lines (1100/1100 <https://depmap.org/portal/gene/CDC20?tab=overview>). Even deletion of a conditional allele of Cdc20 in 4 week-old mice is lethal (Manchado et al, Cancer Cell 2010). Thus, while depletion of Cdc20 causing resistance to Mps1i makes clear mechanistic sense, it is unclear how complete knockout of the essential Cdc20 gene can cause resistance (Fig S1F, 1C-D). Were monoclonal cell lines lacking Cdc20 generated? Or were these experiments done in a polyclonal population? What alleles are being expressed? Is there an internal deletion that removes the epitope for the antibody used but confers a lower level of Cdc20 function? What is the time point at which relative viability was measured after apparent Cdc20 knockout (Fig 1D)? The manuscript would benefit from more precise terminology to describe knockdown/knockout of Cdc20 and ANAPC4.

Since ANAPC4 is also an essential gene, the level of knockdown achieved could be expected to affect proliferation, but only proliferation in the context of reversine is shown (Fig 2E-F). What is the effect of knockdown of CDC20 and ANAPC4 on proliferation in the absence of drug?

3. Additional detail is needed to evaluate the experiments describing increased Cdc20 expression in HPT and RPT versus diploid cells. The images (Fig 4A-B, S4A-B) are too small to readily interpret even when enlarged as much as possible. The Cdc20 signal in red is quite difficult to see. Black and white images for the single channels would be preferable. What is tagged with GFP in the HCT116 GFP cells in Fig 4A? What does the * in RPE1* indicate in Fig S4A? How is the Cdc20 signal

normalized to control for the different cell sizes in HPT and diploid cells? The quantification suggests Cdc20 signal is normalized to area, but area of what?

4. It is really difficult to conclude cells are in an erroneous metaphase in fixed analysis, since the cells may instead be in prometaphase. Yet, this analysis is used to conclude that Cdc20 is higher in cells with an erroneous metaphase. This conclusion would be strengthened by including quantification for cells in early, mid and late prometaphase.

5. In Figure 4C, what is meant by "mitotic failure"? Does that mean the cell failed cytokinesis? Please replace with more standard and/or more descriptive terminology.

6. In Figure 4D, the effects of sgRNA1 on mitosis in HCT116 cells are dramatic while the effects of sgRNA2 are quite subtle. However, in Figure 2E, sgRNA1 has an insignificant effect on EdU incorporation while sgRNA2 has a more substantial effect. If only sgRNA1 affects the mitotic phenotype, why is only sgRNA2 having an impact on resistance?

7. Several clarifications should be made in the Introduction. 1) The SAC is activated in response to unattached kinetochores not abnormal mitotic spindles, as indicated on line 58 (and 383 in the Discussion). 2) Though MCC was previously proposed to prevent progression into anaphase by sequestering CDC20 (lines 61-63), more recent evidence has shown that the MCC can inhibit a second molecule of Cdc20 that has already bound and activated APC/C (Izawa and Pines Nature 2015, Yamaguchi et al Mol Cell 2016). Thus, the sequestration model is appealing simple, but doesn't appear to fully describe the available data. 3) Lines 72-77 seem to suggest that a failure to divide normally is equivalent to mitotic slippage. SAC inhibitors cause cells to transit prometaphase and metaphase more quickly, often so quickly that they don't form a discernable metaphase plate, before segregating their chromosomes in anaphase and dividing into two daughter cells. This is quite different than mitotic slippage, which occurs after long term arrest in mitosis when cells simply decondense their DNA without anaphase, telophase or cytokinesis and form a single tetraploid daughter cell. Please rephrase to clarify this.

Minor comments

8. Figure 4C indicates that ANAPC4 sgRNA2 has a dramatic negative effect on mitosis in 3T3 cells in the absence of reversine. But ANAPC4 sgRNA1 doesn't and neither ANAPC4 sgRNA has dramatic effects in either HCT116 or HPT1 cells. If this is the case, please comment on why ANAPC4 sgRNA2 would have this isolated deleterious effect in 3T3 cells.

9. Throughout the manuscript, it isn't clear if the quantification shown under the western blots is quantification of the single blot shown or (preferably) the average of multiple blots. Please clarify this in the figure legends.

10. The subtitle "Cdc20 is strongly associated with the response to SAC inhibition" would be clearer if it was rephrased to "Cdc20 loss is associated with resistance to SAC inhibition".

11. In the legend for Supplemental Figure 1, the description of panels F and G is reversed.

Referee #3:

General summary and opinion

In the study by Zheng et al, the authors demonstrate that CDC20 levels are a major determinant of sensitivity to the MPS1 inhibitor reversine. They link these effects of CDC20 expression to their prior data demonstrating aneuploid cancer lines are sensitive to MPS1 inhibitors by showing that 1) aneuploid cell lines tend to have increased CDC20 levels, and 2) reducing these levels can cause resistance in these lines, as it does in all other lines tested. These data itself are not surprising or novel given that APC/C loss is well-known to drive MPS1 inhibitor resistance by enhancing SAC strength (Sansregret et al, 2017, Thu et al, 2018, Wild et al, 2016). However, Zheng et al now claim that it is CDC20, and not APC/C subunits themselves, that are the main driver of resistance, which would be an important distinction.

I find this interpretation misleading and I do not find this main conclusion supported by the data itself. I offer some alternative interpretations in the comments below as well as some additional experiments that would help validate some of these claims. I also have a hard time rationalising why APC depletion would not do the same thing as CDC20 depletion in this respect (i.e. improve SAC strength by reducing APC activity). So, I also have a hard time conceptually understanding the overall conclusion and the authors make no attempts to explain this surprising finding. Therefore, in conclusion, I feel the explanations offered below are a more likely explanation of the lack of effect of APC subunit depletion (note only one APC subunit was targeted using approaches that have questionable knockdown efficiency). In my opinion, this seriously affects the novelty of this work, since the only main advance over previous manuscripts would then be the fact that CDC20 itself may in fact be a better/more reliable prognostic marker of MPS1 inhibitor response. That may well turn out to be the case, and this would be an important result with clinical implications, but in my opinion, this does not carry the expected mechanistic advance to warrant publication in EMBO J.

Specific Major Concerns

1) A major conclusion of this study is that it is knockdown of CDC20, and not APC core subunits, that drives resistance to MPS1 inhibitors. The first statement on this (line 137) refers to a lack of effect of APC4 knockout/depletion (S2A-E). However, the CRISPR screen pulled out a variety of APC subunits as driver of resistance (S1C), including APC4. So, it would appear to me that knockout of these subunits does drive resistance, but perhaps not as efficiently as CDC20. That is a very different conclusion to what is currently stated in the manuscript. The lack of effect on re-screening APC4 could be explained by poor shRNA mediated depletion/knockout, and in fact, it appears that this is only approximately 25-50% efficient (S2A and S2D). So, the notion that CDC20 is the key resistance target, and not APC as previously described, is misleading in my opinion.

2) Figure 2 is then devoted to a comparison of APC and CDC20 as predictors of MPS1 inhibitor sensitivity, either from DepMap data or from experiments in HCT116 cells to knockout CDC20 or APC4. This corroborates that CDC20 appears to be the best predictor of response, but again knockout efficiency is likely to be crucial, as evidenced by the fact that one of the gRNAs for CDC20 does not drive resistance. I would certainly not conclude from these experiments that "it is CDC20 expression, rather than expression of the APC/C core components, that mediate the cellular response to SAC inhibition." (line 172). Especially in light of the earlier figure demonstrating very good resistance from a variety of APC components (S1C), and in light of a variety of previous studies, cited in the introduction, that reach similar conclusions.

3) Regarding line 212 "mRNA and protein expression levels of CDC20 and the APC/C subunit ANAPC4 were indeed strongly correlated in the near-diploid cell lines, but this correlation was completely lost in highly-aneuploid cells (Figure 3B)."

I personally cannot see striking differences in the correlation between APC4 and CDC20 mRNA/protein between diploid and aneuploid cells in Figure 3B. The correlations seem at best to be skewed by a few outlier cells in the diploid lines, and there certainly does not appear to me to be a strong correlation in diploid cells, as currently stated.

4) Figures 3c and d show evidence of CDC20 depletion driving resistance in aneuploid lines, but in this case, especially HPT2 cells, the effects are quite marginal. These effects are predicted anyway from earlier data, so I do not see how they support the conclusion that elevated CDC20 drives sensitivity in aneuploid cells. A better experiment in that regard would be to elevate CDC20 in diploid cells and test if that now produces similar sensitivity to their aneuploid counterparts. Also, can making diploid cells aneuploid (without changing CDC20 levels) drive sensitivity or not? These experiments are feasible with acute short-term MPS1 inhibition.

5) Figure 4 explores the link between aneuploidy and CDC20 by using previously derive "highly aneuploid" lines from HCT116 and RPE1 cells. These lines are not discussed or characterized in this manuscript, unless I missed that, but from reading the reference it looks like they are derived from the tetraploidisation of RPE1 or HCT116 cells. If that is correct, then is it aneuploidy or tetraploidy that is associated with elevated CDC20 levels during mitosis? The sentence beginning on line 253 starts to make an assumption about previous aneuploid cell data given these results in tetraploid clones. To make these conclusions one must compare mitotic CDC20 levels in the earlier aneuploid cell lines instead, not in unrelated tetraploid clones. I think acutely inducing aneuploidy in RPE1 cells would be a useful comparison here as well.

6) The final figure in 4C and D shows how mitotic errors are modulated by perturbing CDC20 levels, but not APC4 levels, which are predictable results given the resistance shown earlier. However, the same criticism can be levelled here as before - is this due to the differential penetrance of APC4 and CDC20 depletion? The lack of a significant metaphase delay after APC4 depletion, in comparison to CDC20 depletion, suggests that it is. Otherwise, why does depletion of an APC subunit not cause a metaphase delay? Plenty of studies have shown that metaphase durations can be extended with APC subunit depletion.

Therefore, I believe the most likely interpretation is that CDC20 and APC loss can drive metaphase delays and MPS1 inhibitor resistance, but perhaps it is easier to reduce CDC20 to critically low levels with the chosen sgRNA/shRNAs sequences. I certainly do not agree with conclusion that: "These results indicate that CDC20 is the key player in the cellular response to SAC inhibition, and that it acts by altering metaphase duration and affecting the overall level of chromosomal instability induced by SAC inactivation."

It could be that CDC20 overexpression is a key driver of sensitivity in aneuploid cells, and this would be an important conclusion, but I feel that would require further validation.

Other minor concerns.

7) Given the crucial role of CDC20 in this manuscript, the introduction should point out the role of different CDC20 molecules in the SAC response; namely that CDC20 is a component of MCC, but that MCC then binds a second CDC20 molecules that is bound to APC to inhibit this preformed APC/CDC20 complex (Izawa and Pines, 2015). This model is important to consider when implicating CDC20 levels in the overall SAC response.

8) It is not clear to me how you can quantify CDC20 levels in cells with normal or aberrant divisions from fixed analysis? The legends are not clear but how does one define aberrant division from a fixed population of prometaphase cells?

9) Line 381 "Effectively, SAC inhibition and CDC20 overexpression result in a similar outcome - they both lead to higher effective concentrations of uninhibited APC/C-CDC20 complexes in the cells, allowing cell division to continue even in the presence of malformed spindles."

I cannot see the evidence that CDC20 overexpression leads to higher concentration of uninhibited APC/C-CDC20 complexes? Such mechanistic studies on CDC20 overexpression alone would be valuable.

10) Line 397: "these findings are highly clinically relevant as several SAC inhibitors are currently in clinical trials, with one of them being fast-tracked by the FDA (NCT05251714)."

This refers to a phase I and phase II trial that is estimated to complete in 2 years. I cannot find the evidence that the inhibitors are being fast-tracked by the FDA?

11) In relation to 1D and S1G, how were these assays performed? Legends do not provide details - relative viability after how long of each treatment? Also, what does CDC20 knockout/knockdown do to mitotic duration in the absence of reversine treatment?

Introduction:

We thank the Reviewers for their helpful comments and suggestions. We believe that the manuscript has significantly improved during the Revision process, and we would like to highlight the following:

- We modified the manuscript and focused it on CDC20 as a key (rather than sole) determinant of the cellular sensitivity SACi.
- We took notice of the Reviewers' comments about the physiological relevance, as well as the technical concerns, of full knockout of CDC20, and replaced all the CRISPR-KO experiments with shRNA- and siRNA-knockdown experiments.
- We have expanded the DepMap analyses in several ways, to strengthen the specific association between CDC20 and response to SAC inhibitors.
- We have repeated all key experiments with a non-tetraploid system of aneuploid cells.
- We have added both high-resolution imaging and Western blots following cell synchronization, to further demonstrate CDC20 elevation during aberrant mitosis.

We would like to note that the associations that we observe remain to be validated clinically using data from patients treated with SAC inhibitors. However, these data do not yet exist in public repositories, as relevant drugs are in clinical trials, and to the best of our knowledge parallel CDC20 measurement hasn't been performed in these trials. As we agreed with the *EMBO Rep* Editor, the demonstration of CDC20 as a *bone fide* clinical biomarker is beyond the scope of the current study, but we hope that our paper will encourage the testing of this hypothesis in future clinical studies.

Referee #1:

The spindle assembly checkpoint (SAC) is an intriguing target for anti-cancer therapeutics as loss of SAC function can lead to the accumulation of intolerable levels of aneuploidy in dividing cells. However, there remains some controversy about whether there is a sufficiently therapeutic window (doi 10.1158/1078-0432.CCR-20-4185) as SAC-targeting drugs also impact all healthy proliferating tissues. Notably, Aurora B inhibitors suppress SAC function but have failed to progress in the clinic due to severe myelosuppression. Nevertheless, not all cells accumulate mitotic errors at the same rate in response to SAC inhibitors, and the success of this therapeutic approach will rely on the development of clear biomarkers for highly susceptible cancers.

Previous work has identified weak anaphase promoting complex/cyclosome (APC/C) function as a resistance mechanism to SAC inhibitors. Cells with reduced APC/C activity take longer to complete mitosis and have more opportunity for error correction before mitotic exit. In this

manuscript, the authors use a combination of genetic and bioinformatic approaches to propose that CDC20, the co-activator of the APC/C complex, rather than the APC/C itself, controls the response to SAC inhibition. As such, CDC20 expression levels are proposed as a biomarker for SAC inhibitor-sensitive cancers.

The link between CDC20 expression and SAC inhibitor sensitivity is predictable and well-established by the authors in gene perturbation assays. However, the synthetic systems employed in the manuscript far over-represent the range of CDC20 expression in cancers. This is revealed in the data from the pan-cancer DepMap dataset which show only a very weak correlative relationship between CDC20 expression and sensitivity to SAC inhibition. The data shown in the paper reveal that CDC20 expression alone has little to no predictive power to identify cancers that are more sensitive to SAC perturbation. Therefore, while the authors have performed a clean and thoughtful analysis, they are ultimately unable to establish CDC20 as a bona fide clinically useful biomarker for SAC inhibitor sensitivity.

While we appreciate the Reviewer's comment on the synthetic nature of our systems, we would like to emphasize here, that in this study we meant to carefully control for CDC20 expression in an isogenic system, which is why we downregulated CDC20 levels and compared the impact of these manipulations with the parental (i.e. isogenic) cells. While the CDC20 levels in these experiments might be lower than in most cancers, it does allow us to isolate the effect of CDC20 in and of itself.

To increase the physiological relevance of our findings, in the revised manuscript we have used CDC20 knockdown (via siRNAs and shRNAs) instead of knockout, as detailed below. The establishment of CDC20 as a *bona fide* clinically useful biomarker for SAC inhibitor sensitivity is beyond the scope of the current manuscript, as it would require large patient cohorts and clinical trials. However, the association between CDC20 expression and sensitivity to SAC inhibition is quite strong, as detailed below. The current study focuses on the identification, validation and characterization of this association, which – although not entirely surprising – we believe to be important for the field. Please see below our response to all of the specific comments.

Specific Comments:

- The DepMap analysis in Figure 2 is the crux of the paper as it investigates the impact of CDC20 expression across a large panel of cancer cell lines. The data attempting to discriminate cancers sensitivity to SAC inhibition by CDC20 expression in this analysis is not sufficient to support the central claim of the paper. In these experiments, the correlation between CDC20 expression and sensitivity to MAD2L1/BUB1B disruption is very weak ($R^2 < 0.2$). This is not improved when comparing CDC20 expression versus drug sensitivity for two different MSP1 inhibitors MP1-0479605 ($R^2 = 0.15$) or AZ-3016 ($R^2 = 0.07$). Even segregating sensitivity using the top and bottom quartiles of CDC20 expression as performed in Figure 2C only yields a modest effect that could not be used to stratify sensitive versus insensitive tumors. Ultimately, the authors show that, as anticipated, strong genetic perturbations of CDC20 affect sensitivity, presumably through APC/C activation, but that this effect is not strong enough to be a clinically

relevant biomarker. There are likely to be many factors that contribute to the sensitivity to SAC, inhibition and CDC20 expression is one minor contributor.

It is true that additional factors that contribute to the sensitivity to SAC inhibition likely exist, but our point is that CDC20 is an important factor. Although the correlation is indeed modest, this correlation explains the previously reported association with APC/C signatures – the CDC20 correlation is as strong, or even stronger, than the correlation with the entire multi-gene APC/C signature, and is much stronger in comparison to the expression of APC/C subunits only – this is shown in Fig. 2A-B:

(A,B) Correlation between mRNA expression of an extended APC/C signature (left), an APC/C signature without CDC20 (middle), or CDC20 alone (right) and sensitivity to genetic disruption of the core SAC components BUB1B **(A)** and MAD2L1 **(B)** in human cancer cell lines from the DepMap. The “APC/C subunit-only” signature contains the 14 core APC/C subunits **(14)**, while the “extended APC/C signature” described by Thu et al **(11)** contains three additional APC/C co-factors, including CDC20. The genes included in each signature are listed in Supplementary Table 2. Shown are Spearman’s correlation rho and p values. Spearman correlation (N=661 for MAD2L1 or BUB1B vs extended APC/C or subunit-only APC/C, N=662 for MAD2L1 or BUB1B vs extended APC/C or subunit-only APC/C; *, p-value<0.05; **, p-value<0.01). RNAi dependency scores were obtained from the Achilles genome-wide RNAi screen, DepMap 22Q2 **(13)**

Further, the modest correlation that we identified isn’t minor at all – the DepMap data is inherently noisy, so that a single biomarker rarely allows a complete separation between “sensitive” and “insensitive” cell lines. In fact, the correlation that we see for CDC20 expression and MPS1 inhibition is comparable to that seen for known drug targets and gene expression or gene-level-dependency of these targets. For example, CCNE1 expression and CDK2i ($R^2=0.068$), ATR dependency and ATR kinase inhibitor ($R^2=0.031$), BRD4 dependency and BRD4i ($R^2=0.022$), or TOP2A and TOP2Ai ($R^2=0.018$):

Therefore, the highly-significant correlation that we observed is biologically meaningful, and CDC20 expression isn't merely a minor contributor to MPS1i response (as we demonstrate later on in the manuscript).

To further strengthen this association analysis, we associated CDC20 expression with the response to ~6,500 additional drugs. For each drug we ranked the association of *all genes* with drug sensitivity, allowing us to compare CDC20 to all other protein coding genes. The association of CDC20 expression with drug sensitivity was significantly stronger for MPS1 inhibitors in comparison to all other drugs, showing the specific importance of CDC20 in the response to SAC inhibition – please see the new Fig. 2E:

(E) Distribution of the correlation between CDC20 expression and sensitivity to ~6700 different drugs taken from the PRISM primary repurposing screen 2023, ordered by gene ranking percentile (see methods). Comparison between drug classes was performed by Student's t-test. The ranking of CDC20 compared to all protein coding genes in the response to MPS1 inhibitors is much higher than its average ranking in the response to all other drugs.

Finally, the Reviewer notes that the modest effect shown in Fig. 2C suggests that CDC20 expression is not sufficient for stratifying “sensitive” versus “insensitive” tumors. We agree, and we do not argue that CDC20 expression levels are sufficient for this purpose, or that CDC20 could immediately become a clinically useful biomarker – we acknowledge that examining this would require much more work and large patient cohorts. We therefore toned down any statements related to the clinical applications of our findings throughout the text.

Finally, the results of the additional experiments that were requested by Reviewer #3, as well as our findings in the CRISPR screen that also identify Cdc20 as the strongest determinant of resistance to Mps1 inhibitors in mouse cells, strengthen the association between CDC20 expression and drug sensitivity, as detailed below.

- Polyclonal CDC20 “knockout” cells are not well defined. CDC20 is a common essential gene and therefore either CDC20 is not completely eliminated or there is some unexplored biology at play. These cells would benefit from further characterization such as clonal isolation to define the level of CDC20 remaining.

We thank the Reviewer for this comment. We agree with the Reviewer that CDC20 is considered an essential gene. Our WB analysis following the CDC20 CRISPR-Cas9 editing indeed suggested the residual expression of CDC20. We therefore decided to repeat all of the

CDC20 knockout experiments with CDC20 shRNA and siRNA knockdown. The results recapitulated those with CRISPR, so we decided to replace the CRISPR knockout data with the knockdown data — please see Fig. 1D, Supplementary Fig. 1D,E, Fig. 2F,G, Supplementary Fig. 2A-D, Fig. 3F-I,K, Supplementary Fig. 3F-M:

Fig. 1D (3T3 cells):

(D) Sensitivity of 3T3 cells to 250nM SAC inhibitor reversine for 4 days following Cdc20 knockdown by shRNA, as observed by EdU incorporation assay. Two-way ANOVA, respectively (N = 3, **, p-value <0.01; ****, p-value < 0.0001).

Supplementary Fig. 1D-E:

Zheng et al, Sup. Figure 1

(D,E) qPCR **(D)** or western blot **(E)** validation of Cdc20 knockdown by shRNA in 3T3 cells. Two-sided t-test and multiple t-tests, respectively (N=3; ****, p-value< 0.0001).

Fig. 2F,G (HCT116):

(D,E) Percent of EdU-incorporating HCT116 cells following SAC inhibition (125nM reversine), with and without CDC20 depletion by shRNA (D) or siRNA (E). CDC20 depletion increased the fraction of proliferating cells following drug treatment. Two-way ANOVA (N = 3; *, p-value < 0.05; ****, p-value < 0.0001).

Supplementary Fig. 2A-D:

(A,B) qPCR (A) and western blot (B) validation of CDC20 knockdown by siRNA in HCT116 cells. Two-sided t-test (N=3, ***, p-value < 0.001). (C,D) qPCR (C) and western blot (D) validation of CDC20 knockdown by shRNA in HCT116 cells. Multiple t-tests (N=3, ****, p-value < 0.001).

Fig. 3F-I, K (aneuploid HPT cells):

(F,G) Percent of EdU incorporating cells in HPT1 cells after CDC20 depletion with shRNA **(F)** or siRNA **(G)**, with and without treatment with 250nM Reversine. Two-way ANOVA (N = 3, ****, p-value <0.0001) **(H,I)** Percent of EdU incorporating cells in HPT2 cells after CDC20 depletion with siRNA **(H)** or shRNA **(I)**. Two-way ANOVA (N = 3, ***, p-value < 0.001; ****, p-value < 0.0001). CDC20 depletion in both aneuploid cell lines significantly reduced sensitivity to SAC inhibition. **(K)** Percent of EdU incorporating cells in aneuploid HCT116 cells after CDC20 depletion with siRNA and treatment with 250nM reversine. Two-way ANOVA (N = 3, *, p-value < 0.05). In these cells too, CDC20 depletion decreases the sensitivity to SAC inhibition.

Supplementary Fig. 4A-H:

Zheng et al, Sup. Figure 4

(A,B) Western blot **(A)** and qPCR **(B)** validation of CDC20 knockdown by siRNA in HPT1 cells. Two-sided t-test (N = 3; ****, p-value < 0.0001). A representative image is shown. N=3; Quantification values of the individual bands show the average value of all replicates. **(C,D)** qPCR **(C)** and western blot **(D)** validation of CDC20 knockdown by siRNA in HPT2 cells. Two-sided t-test (N = 3; ***, p-value < 0.001). **(E,F)** Western blot validation of CDC20 depletion by shRNA in HPT1 **(E)** and HPT2 **(F)**. **(G,H)** qPCR **(G)** and western blot **(H)** validation of CDC20 knockdown by siRNA in aneuploid HCT116 cells. Two-sided t-test (N = 3; **, p-value < 0.01).

In addition, recent work from the Cheeseman lab identified several translational isoforms of CDC20, but surprisingly there is no discussion of this work or how the sgRNAs used are expected to impact expression of some or all of these isoforms.

Indeed, the study by Cheeseman et al. shows that the relative levels of CDC20 isoforms influence the mitosis arrest duration and cellular fitness. To address whether the aneuploid and diploid cells differ not only in the total expression levels of CDC20 but also in the relative proportion of the proteo-isoforms, we quantified the isoform expression using CDC20 antibodies that bind to the protein C-terminus (as the different isoforms are truncated at the N-terminus). The ratio between the isoforms was highly similar in the near-diploid and highly-aneuploid cells. Please see **Supplementary Fig. 3E:**

Supplementary Fig. 3E:

(E) Percent of minor CDC20 isoform out of total CDC20, as observed in bulk quantification. There is no significant difference in the fraction of minor isoform between the diploid and aneuploid cell lines.

All the experiments that we show in the revised version of the manuscript were performed using siRNAs and shRNAs, as mentioned above. We did not observe specific differences across CDC20 isoforms. Please see **Supplementary Fig. 1E**, **Supplementary Fig. 2B,D**, **Supplementary Fig. 4A,C,D,E,G**. The relevant WB images were collated below (under a different numbering) for convenience:

In all images, a representative image of a western blot validation of CDC20 knockdown is shown. For each one N=3 and quantification values of the individual bands show the average value of all replicates. **(A)** Cdc20 knockdown by shRNA in 3T3 cells. **(B,C)** CDC20 knockdown by siRNA **(B)** or shRNA **(C)** in HCT116 cells. **(D,E)** CDC20 knockdown by shRNA in HPT1 cells. **(F,G)** CDC20 knockdown by siRNA **(F)** or shRNA **(G)** in HPT2 cells. **(H)** CDC20 knockdown by siRNA in aneuploid HCT116 cells.

As for the gRNAs that we previously used for our experiments, Cdc20 gRNA1 and gRNA2 targeted the full length of Cdc20 in mice; and CDC20 gRNA1 and gRNA2 targeted the full length of CDC20 in humans. Therefore, the effect of the knockout was due to the reduction in the expression of the protein as a whole, not a specific rare isoform, in line with the quantification of the isoforms that we now show.

- The claim that APC/C activity independently from CDC20 expression does not affect SAC inhibitor sensitivity is not well supported. APC/C subunits came out of one of the initial screens refuting this claim and only APC4 is probed in the manuscript. Furthermore, depletion of APC4 was less effective than CDC20 in some critical experiments (figure S3A-B) confounding these results. Finally, bioinformatically approximating APC/C complex activity through an averaged APC/C expression score has significant drawbacks. Notably, this approach masks the effect of concentration-limiting APC/C subunits during complex formation.

We completely accept this criticism, and we toned down this claim. We do not argue that APC/C itself does not affect the sensitivity to SAC inhibitors at all. In fact, as mentioned by the Reviewer, APC/C subunits came up as hits in our CRISPR screen – please see the volcano plot in **Supplementary Fig. 1C** in which we've now annotated the APC/C genes:

Supplementary Fig. 1C:

(C) Correlation between the top ranked 25% of genes in both CRISPR screens based on their statistical significance with all APC/C-related genes highlighted, showing that Cdc20 is by far the most significant outlier of all APC/C extended complex members.

However, CDC20 came up as the strongest hit in this screen (**Fig. 1C** and **Supplementary Fig. 1C**), and its expression was the main contributor to a previously-reported APC/C gene expression signature (Thu et al, PNAS (2018); **Fig. 2A,B** (see the plot above)) suggesting it is of particular importance. Therefore, we claim that CDC20 is a stronger predictor of SACi response in comparison to the APC/C subunits, on which previous studies focused. We have removed the comparisons to ANAPC4, clarified and refined the message, and focused the revised manuscript on CDC20. Please see throughout the text, Fig. 1-4, and the Results and Discussion sections.

- It was not mentioned in the manuscript that the aneuploid derivatives of RPE1 and HPT cells are also whole genome doubled. This may confound the analysis performed in these cells.

We thank the Reviewer for this comment. We now mention this in the manuscript. Importantly, we have now repeated our experiments with HCT116 cells in which aneuploidy was induced by transient SACi (reversine treatment). Following aneuploidization, these cells overexpress CDC20, which increases their mitotic aberrations. CDC20 knockdown increases metaphase duration and significantly reduces mitotic aberrations in these cells, in full agreement with the results from the HCT/HPT system. Therefore, the results are not specific for WGD+ cells. Please see **Fig. 3J,K, Fig. 4K,L, Fig. 5G**, and the Discussion.

Fig. 3J:

(J) Bulk quantification of CDC20 in wild-type HCT116 and after following aneuploidy induction using 250nM Reversine. Two-sided (T-test N= 5; **, p-value < 0.01).

Fig. 3K:

K

(K) Percent of EdU incorporating cells in aneuploid HCT116 cells after CDC20 depletion with siRNA and treatment with 250nM Reversine. Two-way ANOVA (N = 3, *, p-value < 0.05). In these cells too, CDC20 depletion decreases the sensitivity to SAC inhibition.

Fig. 4K,L:

(K,L) Aneuploid HCT116 cells treated with 125nM Reversine also exhibit significantly decreased mitotic aberrations after CDC20 depletion.

Fig. 5F:

(F) Metaphase duration in aneuploid HCT116 cells under control conditions or CDC20 depletion by siRNA. In all three cell lines CDC20 depletion led to a significant increase in metaphase duration. Two-sided t-test (N =3, ****, p-value < 0.0001).

Referee #2:

In this manuscript, Zheng et al identify Cdc20 as the top hit in two independent CRISPR-Cas9 screens to identify genes whose depletion confers resistance to inhibition of the spindle assembly checkpoint (SAC) kinase Mps1. Depletion of Cdc20 conferred resistance to Mps1 inhibition in 3T3, HCT116 and HCT116 post-tetraploid (HPT) cells. Previous screens to identify causes of resistance to Mps1 inhibition (Sansregret et al Cancer Discovery 2017, Thu et al PNAS 2018) identified APC/C subunits as contributing to response to Mps1 inhibition. However, Zheng et al report that targeting of the APC/C subunit ANAPC4 does not confer resistance to Mps1 inhibition in 3T3, HCT116 or HPT cells. Moreover, though expression of APC/C subunits correlates with sensitivity to SAC inhibition in the cancer Dependency Map, this is largely driven by expression of Cdc20, which is the APC/C cofactor that drives APC/C activity during mitosis. Targeting Cdc20 extends mitotic duration and decreases chromosome segregation errors after Mps1 inhibition, offering a mechanistic explanation for causing resistance, while targeting ANAPC4 largely does not. The authors conclude that increased Cdc20 expression is a promising biomarker for Mps1 inhibitors (and any other SAC inhibitors), while APC/C expression is not.

Predictive biomarkers of response substantially improve outcomes and are therefore highly sought after for both existing and emerging therapies. A biomarker for Mps1 inhibitors would be beneficial to include in clinical trials and would substantially aid in bringing these drugs to market. The broad focus on SAC inhibitors in general in the title and summary is somewhat odd, since the manuscript focuses exclusively on Mps1 inhibitors, and these are the only SAC inhibitors in or approaching the clinic to my knowledge. Nevertheless, the identification of Cdc20 mRNA expression as a simple potential biomarker of response to these drugs is a significant finding of broad interest that is well supported by the data presented here. The companion argument that APC/C expression does not dictate response is less compelling and will require additional justification prior to publication.

We thank the Reviewer for finding our CDC20 results compelling.

We changed the title of the paper to highlight that we focus on MPS1 inhibitors, which are indeed the only SAC inhibitors approaching the clinic to the best of our knowledge.

We revised and refined the argument related to the APC/C subunits, as detailed below.

Major concerns

1. The argument that APC/C expression does not contribute to Mps1 inhibitor resistance relies on depletion of a single subunit of APC/C, ANAPC4. Though the manuscript refers to ANAPC4 knockout throughout, the available western blots (figure S2B,D, S3B) clearly show that ANAPC4 has not been knocked out but is rather knocked down. Mechanistically (as described very nicely in the discussion), Mps1 inhibition is lethal because the mitotic checkpoint is unable to delay mitosis sufficiently to prevent large amounts of chromosome missegregation. Slowing mitosis allows extra time for chromosomes to make proper attachments to spindle microtubules,

reducing chromosome missegregation and cell death. The authors show convincingly in figure 4 that Cdc20 depletion slows mitosis and decreases chromosome missegregation in the Mps1 inhibitor reversine. However, mechanistically, decreasing expression of APC/C subunits should also slow mitosis and it has previously been demonstrated that sufficient ANAPC4 depletion slows mitosis (Thu et al PNAS 2018 figure 3G). These experiments appeared to involve a more robust knockdown than achieved here (Thu et al figure 3C, S3C, S4A). The simplest explanation for the lack of an effect here is that the knockdown of ANAPC4 is insufficient to confer the phenotype. Knockdown of at least a second component of the APC/C would also strengthen the argument.

We completely accept this criticism, and in view of the Reviewers' concerns and our discussion with the Editor we decided to remove the data pertaining to ANAPC4 and focus our manuscript on CDC20. We do not mean to argue that APC/C itself does not affect the sensitivity to SAC inhibitors. In fact, APC/C subunits came up as hits in our CRISPR screen – please see the plot in **Supplementary Fig. 1C** in which we have now annotated the APC/C genes (as also described above in our response to Reviewer 1). However, CDC20 came up as the strongest hit in this screen (**Fig. 1C and Supplementary Fig. 1C**), and its expression was the main contributor to a previously reported APC/C gene expression signature (Thu et al PNAS 2018, **Fig. 2A,B**) suggesting it is of particular importance. Therefore, in our revised manuscript, we claim that CDC20 is a stronger predictor of SACi response in comparison to the APC/C subunits, on which previous studies focused. We have removed the comparisons to ANAPC4, clarified and refined the message, and focused the revised manuscript on CDC20. Please see throughout the text, Fig. 1-4, and the Results and Discussion sections.

Supplementary Fig. 1C:

(C) Correlation between the top ranked 25% of genes in both CRISPR screens based on their statistical significance with all APC/C-related genes highlighted, showing that Cdc20 is by far the most significant outlier of all APC/C extended complex members.

Fig. 2A,B:

(A,B) Correlation between mRNA expression of an extended APC/C signature (left), an APC/C signature without CDC20 (middle), or CDC20 alone (right) and sensitivity to genetic disruption of the core SAC components BUB1B **(A)** and MAD2L1 **(B)** in human cancer cell lines from the DepMap. The “APC/C subunit-only” signature contains the 14 core APC/C subunits **(14)**, while the “extended APC/C signature” described by Thu et al **(11)** contains three additional APC/C co-factors, including CDC20. The genes included in each signature are listed in Supplementary Table 2. Shown are Spearman’s correlation rho and p values. Spearman correlation (N=661 for MAD2L1 or BUB1B vs extended APC/C or subunit-only APC/C, N=662 for MAD2L1 or BUB1B vs extended APC/C or subunit-only APC/C; *, p-value<0.05; **, p-value<0.01). RNAi dependency scores were obtained from the Achilles genome-wide RNAi screen, DepMap 22Q2 **(13)**

2. Throughout the manuscript, Cdc20 is described as being knocked out. Though it is clear that a portion of Cdc20 remains in figure S1E, S2G and S3A, the quantitation in figure S1F indicates generation of 3T3 cell lines truly null for Cdc20. However, Cdc20 is well established as an essential gene in mice (Li et al, MCB 2017) and in cell lines (1100/1100 <https://depmap.org/portal/gene/CDC20?tab=overview>). Even deletion of a conditional allele of Cdc20 in 4 week-old mice is lethal (Manchado et al, Cancer Cell 2010). Thus, while depletion of Cdc20 causing resistance to Mps1i makes clear mechanistic sense, it is unclear how complete knockout of the essential Cdc20 gene can cause resistance (figure S1F, 1C-D). Were monoclonal cell lines lacking Cdc20 generated? Or were these experiments done in a polyclonal population? What alleles are being expressed? Is there an internal deletion that removes the epitope for the antibody used but confers a lower level of Cdc20 function? What is the time point at which relative viability was measured after apparent Cdc20 knockout (figure 1D)? The manuscript would benefit from more precise terminology to describe knockdown/knockout of Cdc20 and ANAPC4.

We thank the Reviewer for this comment. We agree with the Reviewer that CDC20 is considered an essential gene. Our WB analysis following the CDC20 CRISPR-Cas9 editing indeed suggested the residual expression of CDC20. We therefore decided to repeat all of the CDC20 knockout experiments with CDC20 shRNA and siRNA knockdown. The results recapitulated those with CRISPR, so we decided to replace the CRISPR knockout data with the siRNA/shRNA knockdown data — please see **Fig. 1D**, **Supplementary Fig. 1D,E**, **Fig. 2F,G**, **Supplementary Fig. 2A-D**, **Fig. 3F-I,K**, **Supplementary Fig. 3F-M**:

Fig. 1D (3T3 cells):

(D,F) Sensitivity of 3T3 cells to 250nM SAC inhibitor reversine for 4 days following Cdc20 knockdown by shRNA **(D)**, or siRNA **(F)**, as observed by EdU incorporation assay. Two-way ANOVA, respectively (N = 3, **, p-value < 0.01; ****, p-value < 0.0001).

Supplementary Fig. 1D,E:

(D,E) qPCR **(D)** or western blot **(E)** validation of Cdc20 knockdown by shRNA in 3T3 cells. Multiple t-tests, (N=3; ****, p-value < 0.0001).

Fig. 2F,G (HCT116):

(F,G) Percent of EdU-incorporating HCT116 cells following SAC inhibition (125nM reversine), with and without CDC20 depletion by shRNA (F) or siRNA (G). CDC20 depletion increased the fraction of proliferating cells following drug treatment. Two-way ANOVA (N = 3; *, p-value < 0.05; ****, p-value < 0.0001).

Supplementary Fig. 2A-D:

(A,B) qPCR (A) and western blot (B) validation of CDC20 knockdown by shRNA in HCT116 cells. Multiple t-tests (N=3, ****, p-value < 0.001). (C,D) qPCR (C) and western blot (D) validation of CDC20 knockdown by siRNA in

HCT116 cells. Two-sided t-test (N=3, ***, p-value < 0.001). A representative image is shown. N=3; Quantification values of the individual bands show the average value of all replicates.

Fig. 3F-I, K (aneuploid HPT cells):

(F,G) Percent of EdU incorporating cells in HPT1 cells after CDC20 depletion with shRNA (F) or siRNA (G), with and without treatment with 250nM Reversine. Two-way ANOVA (N = 3, ****, p-value <0.0001) (H,I) Percent of EdU incorporating cells in HPT2 cells after CDC20 depletion with shRNA (H) or siRNA (I). Two-way ANOVA (N = 3, ***, p-value < 0.001; ****, p-value < 0.0001). CDC20 depletion in both aneuploid cell lines significantly reduced sensitivity to SAC inhibition. (K) Percent of EdU incorporating cells in aneuploid HCT116 cells after CDC20 depletion with siRNA and treatment with 250nM Reversine. Two-way ANOVA (N = 3, *, p-value < 0.05). In these cells too, CDC20 depletion decreases the sensitivity to SAC inhibition.

Supplementary Fig. 4A-H:

(A,B) Western blot **(A)** and qPCR **(B)** validation of CDC20 knockdown by siRNA in HPT1 cells. Two-sided t-test ($N = 3$; ****, p -value < 0.0001). A representative image is shown. $N=3$; Quantification values of the individual bands show the average value of all replicates. **(C,D)** qPCR **(C)** and western blot **(D)** validation of CDC20 knockdown by siRNA in HPT2 cells. Two-sided t-test ($N = 3$; ***, p -value < 0.001). **(E,F)** Western blot validation of CDC20 depletion by shRNA in HPT1 **(E)** and HPT2 **(F)**. **(G,H)** qPCR **(G)** and western blot **(H)** validation of CDC20 knockdown by siRNA in aneuploid HCT116 cells. Two-sided t-test ($N = 3$; **, p -value < 0.01).

Since ANAPC4 is also an essential gene, the level of knockdown achieved could be expected to affect proliferation, but only proliferation in the context of reversine is shown (figure 2E-F). What is the effect of knockdown of CDC20 and ANAPC4 on proliferation in the absence of drug?

We have removed the ANAPC4 data, and replaced the CDC20 CRISPR knockout data with CDC20 shRNA and siRNA knockdown data. We now show the effects of CDC20 knockdown in the absence of the drug in **Supplementary Fig. 1D,E**, **Supplementary Fig. 2A-D**, **Supplementary Fig. 4A-H** (please see above). As expected, CDC20 knockdown impairs proliferation in the absence of drug.

3. Additional detail is needed to evaluate the experiments describing increased Cdc20 expression in HPT and RPT versus diploid cells. The images (figure 4A-B, S4A-B) are too small to readily interpret even when enlarged as much as possible. The Cdc20 signal in red is quite difficult to see. Black and white images for the single channels would be preferable. What is tagged with GFP in the HCT116 GFP cells in figure 4A? What does the * in RPE1* indicate in

figure S4A? How is the Cdc20 signal normalized to control for the different cell sizes in HPT and diploid cells? The quantification suggests Cdc20 signal is normalized to area, but area of what?

The images have been replaced with 100x images including z-stacking, in black and white. The * in RPE* refers to the fact that these are the specific RPE1 cells from which the isogenic-aneuploid cell lines were derived, to emphasize the fact the cell lines are isogenic (i.e., that the control cells are suitable). The CDC20 signal is quantified by intensity per area, specifically to the area of the nucleus (as it concentrates in the nucleus during mitosis). This is now better explained in the Methods. Please see the revised Fig. 3C and Fig. 4A,B:

Fig. 3C:

(C) Representative image and quantification of CDC20 at metaphase in the isogenic-aneuploid system HCT116-HPT, after synchronization with 7.5uM RO-3306 and 10uM MG-132. Highly aneuploid cells express significantly higher levels of CDC20 than their diploid counterparts. One-sample t-test (N = 3, *, p-value < 0.05; **, p-value < 0.01).

Fig. 4A,B:

(A,B) Representative immunofluorescence images (A) and quantification (B) of CDC20 protein levels during normal and aberrant metaphases, in single HCT116 cells and their highly-aneuploid derivatives after synchronization with 7.5uM RO-3306. Regardless of ploidy background, cells with mitotic aberrations express significantly higher levels of

CDC20 during metaphase than cells undergoing normal division. One-sample t-test (N=3, *, p-value <0.05; **, p-value <0.01)

4. It is really difficult to conclude cells are in an erroneous metaphase in fixed analysis, since the cells may instead be in prometaphase. Yet, this analysis is used to conclude that Cdc20 is higher in cells with an erroneous metaphase. This conclusion would be strengthened by including quantification for cells in early, mid and late prometaphase.

We thank the Reviewer for this suggestion. We strengthened this analysis by the following additional experiments:

(1) We synchronized the cells in G2/M using nocodazole, and performed Western Blots on the bulk cell populations to compare CDC20 protein levels (please see **Fig. 3D** and **Supplementary Fig. 3D**).

(2) We specifically synchronized cells in G2/M using the CDK1 inhibitor RO-3306, followed by release into two different conditions: (A) culture for 45min without further treatment (under this condition most of the cells progress into metaphase, based on visual inspection); or (B) culture for 45min followed by an addition of a proteasome inhibitor for an additional 60min, leading to a full metaphase arrest (as Cyclin B1 cannot be degraded). CDC20 was then quantified in single cells by immunofluorescence, repeating our previous experiments in higher magnification (100x + z-stacking). Please see **Fig. 4A,B**.

All of these experiments support our original conclusion that CDC20 expression levels are higher in erroneous metaphases than in normal metaphases.

Fig. 3D:

(D) Bulk quantification of CDC20 protein expression in the HCT116-HPT system during mitosis after synchronization with 330nM Nocodazole. Here as well, the highly aneuploid cells express higher CDC20 protein levels than their diploid counterparts. One-sample t-test (N = 3, *, p-value < 0.05; **, p-value < 0.01).

Supplementary Fig. 3D:

(D) Representative western blot image of bulk CDC20 expression in the HCT116-HPT cell line set following synchronization with 330nM Nocodazole. Aneuploid cells express higher levels of CDC20 than their diploid counterparts.

Fig. 4A,B:

(A,B) Representative immunofluorescence images (A) and quantification (B) of CDC20 protein levels during normal and aberrant metaphases, in single HCT116 cells and their highly-aneuploid derivatives after synchronization with 7.5uM RO-3306. Regardless of ploidy background, cells with mitotic aberrations express significantly higher levels of CDC20 during metaphase than cells undergoing normal division. One-sample t-test (N=3, *, p-value <0.05; **, p-value <0.01).

5. In Figure 4C, what is meant by "mitotic failure"? Does that mean the cell failed cytokinesis? Please replace with more standard and/or more descriptive terminology.

We used the term "mitotic failure" whenever cells failed to form two daughter cells, resulting either in bi-nucleated cells or in one large (near-tetraploid) nucleus. We agree with the Reviewer that the better term is "cytokinesis failure" and we corrected this in the legend and in the text.

6. In Figure 4D, the effects of sgRNA1 on mitosis in HCT116 cells are dramatic while the effects of sgRNA2 are quite subtle. However, in Figure 2E, sgRNA1 has an insignificant effect on EdU incorporation while sgRNA2 has a more substantial effect. If only sgRNA1 affects the mitotic phenotype, why is only sgRNA2 having an impact on resistance?

We do not fully understand this point as the effect of sgRNA1 on mitosis in the HCT116 cells was stronger than the effect of sgRNA2 (please see previous Fig. 4D,G), and thus in line with the observed effects on resistance. Please see previous Fig. 4D,G.

Previous Fig. 4D,G:

(D) HCT116 cells treated with 125nM reversine exhibit decreased mitotic aberrations after CDC20 knockout (middle) and increased mitotic aberrations after ANAPC4 knockout (right). One-sided Kruskal Wallis (*, p-value < 0.05; ****, p-value < 0.0001). **(G)** Time spent in metaphase for indicated genotypes quantified by live-cell imaging. CDC20 knockout (left) prolongs time in metaphase in HCT116 cells.

In any case, as described above, the CRISPR-KO data were replaced with shRNA- and siRNA-KD data in the revised manuscript. Please also see Supplementary Fig. 1D,E, Supplementary Fig. 2A-D, Supplementary Fig. 4A-H above.

7. Several clarifications should be made in the Introduction. 1) The SAC is activated in response to unattached kinetochores, not abnormal mitotic spindles, as indicated on line 58 (and 383 in the Discussion). 2) Though MCC was previously proposed to prevent progression into anaphase by sequestering CDC20 (lines 61-63), more recent evidence has shown that the MCC can inhibit a second molecule of Cdc20 that has already bound and activated APC/C (Izawa and Pines Nature 2015, Yamaguchi et al Mol Cell 2016). Thus, the sequestration model is appealing simple, but doesn't appear to fully describe the available data. 3) Lines 72-77 seem to suggest that a failure to divide normally is equivalent to mitotic slippage. SAC inhibitors cause cells to transit prometaphase and metaphase more quickly, often so quickly that they don't form a discernable metaphase plate, before segregating their chromosomes in anaphase and dividing into two daughter cells. This is quite different from mitotic slippage, which occurs after long term arrest in mitosis when cells simply decondense their DNA without anaphase, telophase or cytokinesis and form a single tetraploid daughter cell. Please rephrase to clarify this.

We thank the Reviewer for these thoughtful clarifications, and we have incorporated them into the Introduction and the discussion. Please see lines 64-67, 80-83, 452-456.

Minor comments

8. Figure 4C indicates that ANAPC4 sgRNA2 has a dramatic negative effect on mitosis in 3T3 cells in the absence of reversine. But ANAPC4 sgRNA1 doesn't and neither ANAPC4 sgRNA has dramatic effects in either HCT116 or HPT1 cells. If this is the case, please comment on why ANAPC4 sgRNA2 would have this isolated deleterious effect in 3T3 cells.

We believe this was because of different knockdown efficiencies between the human and mouse cells. However, in discussion with the Editor, we decided to remove the ANAPC data and replace the CDC20 sgRNA experiments with shRNA and siRNA experiments, as described above.

9. Throughout the manuscript, it isn't clear if the quantification shown under the western blots is quantification of the single blot shown or (preferably) the average of multiple blots. Please clarify this in the figure legends.

We apologize for the confusion, but the quantification numbers under Western blots are indeed the average values of three independent blots. This is now clarified in the legends.

10. The subtitle "Cdc20 is strongly associated with the response to SAC inhibition" would be clearer if it was rephrased to "Cdc20 loss is associated with resistance to SAC inhibition".

Thank you for this suggestion. We amended this subtitle, as suggested.

11. In the legend for Supplemental Figure 1, the description of panels F and G is reversed.

Many thanks for noting this, we corrected this error.

Referee #3:

General summary and opinion

In the study by Zheng et al, the authors demonstrate that CDC20 levels are a major determinant of sensitivity to the MPS1 inhibitor reversine. They link these effects of CDC20 expression to their prior data demonstrating aneuploid cancer lines are sensitive to MPS1 inhibitors by showing that 1) aneuploid cell lines tend to have increased CDC20 levels, and 2) reducing these levels can cause resistance in these lines, as it does in all other lines tested. These data itself are not surprising or novel given that APC/C loss is well-known to drive MPS1 inhibitor resistance by enhancing SAC strength (Sansregret et al, 2017, Thu et al, 2018, Wild et al, 2016). However, Zheng et al now claim that it is CDC20, and not APC/C subunits themselves, that are the main driver of resistance, which would be an important distinction.

I find this interpretation misleading and I do not find this main conclusion supported by the data itself. I offer some alternative interpretations in the comments below as well as some additional experiments that would help validate some of these claims. I also have a hard time rationalising why APC depletion would not do the same thing as CDC20 depletion in this respect (i.e. improve SAC strength by reducing APC activity). So, I also have a hard time conceptually understanding the overall conclusion and the authors make no attempts to explain this surprising finding. Therefore, in conclusion, I feel the explanations offered below are a more likely explanation of the lack of effect of APC subunit depletion (note only one APC subunit was targeted using approaches that have questionable knockdown efficiency). In my opinion, this seriously affects the novelty of this work, since the only main advance over previous manuscripts would then be the fact that CDC20 itself may in fact be a better/more reliable prognostic marker of MPS1 inhibitor response. That may well turn out to be the case, and this would be an important result with clinical implications, but in my opinion, this does not carry the expected mechanistic advance to warrant publication in *EMBO J*.

We thank the Reviewer for their comments and for their suggestions for additional experiments – please also see our detailed point-by-point response below. We focus our manuscript now on CDC20 only, based on the editorial assessment that establishing the role of CDC20 in regulating the sensitivity to SACi (especially in the context of aneuploid cells) would fit well within the scope and interest of the readers of *EMBO Reports*.

Specific Major Concerns

1) A major conclusion of this study is that it is the knockdown of CDC20, and not APC core subunits, that drives resistance to MPS1 inhibitors. The first statement on this (line 137) refers to a lack of effect of APC4 knockout/depletion (S2A-E). However, the CRISPR screen pulled out a variety of APC subunits as driver of resistance (S1C), including APC4. So, it would appear to me that knockout of these subunits does drive resistance, but perhaps not as efficiently as CDC20. That is a very different conclusion to what is currently stated in the manuscript. The lack of effect on re-screening APC4 could be explained by poor shRNA mediated

depletion/knockout, and in fact, it appears that this is only approximately 25-50% efficient (S2A and S2D). So, the notion that CDC20 is the key resistance target, and not APC as previously described, is misleading in my opinion.

We completely accept this criticism, which was noted by all Reviewers, and in discussion with the Editor we decided to remove the data pertaining to ANAPC4 and focus our manuscript on CDC20. In fact, we had not meant to argue that APC/C itself does not affect the sensitivity to SAC inhibitors as APC/C subunits indeed came up as hits in our CRISPR screen, rightly noted by the Reviewer. We have now toned down our conclusions about the relative roles of CDC20 and APC/C in determining the sensitivity towards MPS1 inhibitors. We explicitly note our finding that other APC subunits are enriched in our CRISPR screen, as shown in **Supplementary Fig. 1C**. However, as CDC20 came up as the strongest hit in our CRISPR screen (**Fig. 1C** and **Supplementary Fig. 1C**), and its expression was the main contributor to a previously reported APC/C gene expression signature (Thu et al PNAS 2018, **Fig. 2A,B**), we do describe CDC20 as a key player in the response to MPS1 inhibitors and conclude that CDC20 is a stronger predictor of SACi response than other APC/C subunits, on which previous studies focused. In discussion with the Editor, we have removed the comparisons to ANAPC4, clarified and refined the message, and focused the revised manuscript on CDC20. Please see **Fig. 1-4** and the Results and Discussion sections.

Supplementary Fig. 1C:

(C) Correlation between the top ranked 25% of genes in both CRISPR screens based on their statistical significance with all APC/C-related genes highlighted, showing that Cdc20 is by far the most significant outlier of all APC/C extended complex members.

Fig. 2A,B:

(A,B) Correlation between mRNA expression of an extended APC/C signature (left), an APC/C signature without CDC20 (middle), or CDC20 alone (right) and sensitivity to genetic disruption of the core SAC components BUB1B **(A)** and MAD2L1 **(B)** in human cancer cell lines from the DepMap. The “APC/C subunit-only” signature contains the 14 core APC/C subunits **(14)**, while the “extended APC/C signature” described by Thu et al **(11)** contains three additional APC/C co-factors, including CDC20. The genes included in each signature are listed in Supplementary Table 2. Shown are Spearman’s correlation rho and p values. Spearman correlation (N=661 for MAD2L1 or BUB1B vs extended APC/C or subunit-only APC/C, N=662 for MAD2L1 or BUB1B vs extended APC/C or subunit-only APC/C; *, p-value<0.05; **, p-value<0.01). RNAi dependency scores were obtained from the Achilles genome-wide RNAi screen, DepMap 22Q2 **(13)**.

2) Figure 2 is then devoted to a comparison of APC and CDC20 as predictors of MPS1 inhibitor sensitivity, either from DepMap data or from experiments in HCT116 cells to knockout CDC20 or APC4. This corroborates that CDC20 appears to be the best predictor of response, but again knockout efficiency is likely to be crucial, as evidenced by the fact that one of the gRNAs for CDC20 does not drive resistance. I would certainly not conclude from these experiments that "it is CDC20 expression, rather than expression of the APC/C core components, that mediate the cellular response to SAC inhibition." (line 172). Especially in light of the earlier figure demonstrating very good resistance from a variety of APC components (S1C), and in light of a variety of previous studies, cited in the introduction, that reach similar conclusions.

The Reviewer brings up an important issue: can our phenotypes be explained by differences in knockout efficiencies? We fully agree that since CDC20 and APC/C subunits are essential genes, validating our findings with CRISPR KO is not ideal, exactly for the reasons the Reviewer brings forward. We therefore repeated and replaced all CRISPR KO data with shRNA and siRNA data. Furthermore, as explained above, we removed all ANAPC4 data and now

focus the manuscript on CDC20. Below, please see the results of the shRNA- and si-RNA experiments, which fully reproduced those from the CRISPR KO.

Fig. 1D (3T3 cells):

(D,F) Sensitivity of 3T3 cells to 250nM SAC inhibitor reversine for 4 days following Cdc20 knockdown by shRNA (D), or siRNA (F), as observed by EdU incorporation assay. Two-way ANOVA, respectively (N = 3, **, p-value <0.01; ***, p-value < 0.0001).

Supplementary Fig. 1D,E:

(D,E) qPCR (D) or western blot (E) validation of Cdc20 knockdown by shRNA in 3T3 cells. Multiple t-tests, (N=3; ****, p-value< 0.0001).

Figure 2F,G (HCT116 cells):

(F,G) Percent of EdU-incorporating HCT116 cells following SAC inhibition (125nM reversine), with and without CDC20 depletion by shRNA **(F)** or siRNA **(G)**. CDC20 depletion increased the fraction of proliferating cells following drug treatment. Two-way ANOVA (N = 3; *, p-value < 0.05; ****, p-value < 0.0001).

Fig. 3F-I, K:

(F,G) Percent of EdU incorporating cells in HPT1 cells after CDC20 depletion with siRNA **(F)** or shRNA **(G)**, with and without treatment with 250nM Reversine. Two-way ANOVA (N = 3, ****, p-value < 0.0001) **(H,I)** Percent of EdU incorporating cells in HPT2 cells after CDC20 depletion with siRNA **(H)** or shRNA **(I)**. Two-way ANOVA (N = 3, ***, p-value < 0.001; ****, p-value < 0.0001). CDC20 depletion in both aneuploid cell lines significantly reduced sensitivity to SAC inhibition. **(K)** Percent of EdU incorporating cells in HPT1 cells after CDC20 depletion with siRNA and treatment with 250nM Reversine. Two-way ANOVA (N = 3, *, p-value < 0.05). In these cells too, CDC20 depletion decreases the sensitivity to SAC inhibition.

Supplementary Fig. 2A-D:

(A,B) qPCR (A) and western blot (B) validation of CDC20 knockdown by shRNA in HCT116 cells. Multiple t-tests (N=3, ****, p-value < 0.001). (C,D) qPCR (C) and western blot (D) validation of CDC20 knockdown by siRNA in HCT116 cells. Two-sided t-test (N=3, ***, p-value < 0.001). A representative image is shown. N=3; Quantification values of the individual bands show the average value of all replicates.

Supplementary Fig. 4A-H (aneuploid HPT cells):

(A,B) Western blot **(A)** and qPCR **(B)** validation of CDC20 knockdown by siRNA in HPT1 cells. Two-sided t-test (N = 3; ****, p-value < 0.0001). A representative image is shown. N=3; Quantification values of the individual bands show the average value of all replicates. **(C,D)** qPCR **(C)** and western blot **(D)** validation of CDC20 knockdown by siRNA in HPT2 cells. Two-sided t-test (N = 3; ***, p-value < 0.001). **(E,F)** Western blot validation of CDC20 depletion by shRNA in HPT1 **(E)** and HPT2 **(F)**. **(G,H)** qPCR **(G)** and western blot **(H)** validation of CDC20 knockdown by siRNA in aneuploid HCT116 cells. Two-sided t-test (N = 3; **, p-value < 0.01).

Since ANAPC4 is also an essential gene, the level of knockdown achieved could be expected to affect proliferation, but only proliferation in the context of reversine is shown (figure 2E-F). What is the effect of knockdown of CDC20 and ANAPC4 on proliferation in the absence of drug?

This is a fair point and we indeed noted proliferation defects in CDC20 and ANAPC4 knockout cells. As discussed under previous points, we have removed the ANAPC4 data, and replaced the CDC20 CRISPR knockout experiments with CDC20 shRNA and siRNA knockdown experiments. We now show the effects of CDC20 knockdown in the absence of the drug in **Supplementary Fig. 1D,E** and **Supplementary Fig. 2A-D, Supplementary Fig. 4A-H** above.

Supplementary Fig. 1D,E:

(D,E) qPCR **(D)** or western blot **(E)** validation of Cdc20 knockdown by shRNA in 3T3 cells. Multiple t-tests, (N=3; ****, p-value < 0.0001).

(D,E) qPCR validation of Cdc20 knockdown by siRNA **(D)** or shRNA **(E)** in 3T3 cells. Two-sided t-test and multiple t-tests, respectively (N=3; ****, p-value < 0.0001). **(F,G)** WB validation of Cdc20 knockdown by siRNA **(F)** or shRNA **(G)** in 3T3 cells.

3) Regarding line 212 "mRNA and protein expression levels of CDC20 and the APC/C subunit ANAPC4 were indeed strongly correlated in the near-diploid cell lines, but this correlation was completely lost in highly-aneuploid cells (Figure 3B)."

I personally cannot see striking differences in the correlation between APC4 and CDC20 mRNA/protein between diploid and aneuploid cells in Figure 3B. The correlations seem at best to be skewed by a few outlier cells in the diploid lines, and there certainly does not appear to me to be a strong correlation in diploid cells, as currently stated.

As mentioned in the response to the previous comments, we decided to remove the APC4 data. We do find, however, that the correlation between CDC20 expression and MPS1 inhibitor response is the culprit of the previously-observed correlation between MPS1 inhibition and APC/C multi-gene signatures, and that this correlation is stronger than that of MPS1 inhibition and any of the APC/C subunits – please see **Fig. 2A-B** (also shown above).

Fig. 2A,B:

(A,B) Correlation between mRNA expression of an extended APC/C signature (left), an APC/C signature without CDC20 (middle), or CDC20 alone (right) and sensitivity to genetic disruption of the core SAC components BUB1B **(A)** and MAD2L1 **(B)** in human cancer cell lines from the DepMap. The “APC/C subunit-only” signature contains the 14 core APC/C subunits **(14)**, while the “extended APC/C signature” described by Thu et al **(11)** contains three additional APC/C co-factors, including CDC20. The genes included in each signature are listed in Supplementary Table 2. Shown are Spearman’s correlation rho and p values. Spearman correlation (N=661 for MAD2L1 or BUB1B vs extended APC/C or subunit-only APC/C, N=662 for MAD2L1 or BUB1B vs extended APC/C or subunit-only APC/C; *, p-value<0.05; **, p-value<0.01). RNAi dependency scores were obtained from the Achilles genome-wide RNAi screen, DepMap 22Q2 **(13)**.

Further, the correlation between CDC20 expression and MPS1 inhibition is comparable to that seen for known drug targets and gene expression or gene-dependency of these targets. For example, CCNE1 expression and CDK2i ($R^2=0.068$), ATR dependency and ATR kinase inhibitor ($R^2=0.031$), BRD4 dependency and BRD4i ($R^2=0.022$), or TOP2A and TOP2Ai ($R^2=0.018$):

Therefore, the highly-significant correlation that we observed is biologically meaningful, demonstrating that CDC20 expression is a major contributor to MPS1i response.

To further strengthen this association analysis, we associated CDC20 expression with the response to ~6,500 additional drugs. For each drug we ranked the association of ~19,000 genes with drug sensitivity, allowing us to compare CDC20 to all other protein coding genes. The association of CDC20 expression with drug sensitivity was significantly stronger for MPS1

inhibitors in comparison to all other drugs, showing the specific importance of CDC20 in the response to SAC inhibition – please see the new Fig. 2E:

(E) Distribution of the correlation between CDC20 expression and sensitivity to ~6700 different drugs taken from the PRISM primary repurposing screen 2023, ordered by gene ranking percentile (see methods). Comparison between drug classes was performed by Student's t-test. The ranking of CDC20 compared to all protein coding genes in the response to MPS1 inhibitors is much higher than its average ranking in the response to all other drugs.

4) Figures 3c and d show evidence of CDC20 depletion driving resistance in aneuploid lines, but in this case, especially HPT2 cells, the effects are quite marginal. These effects are predicted anyway from earlier data, so I do not see how they support the conclusion that elevated CDC20 drives sensitivity in aneuploid cells. A better experiment in that regard would be to elevate CDC20 in diploid cells and test if that now produces similar sensitivity to their aneuploid counterparts. Also, can making diploid cells aneuploid (without changing CDC20 levels) drive sensitivity or not? These experiments are feasible with acute short-term MPS1 inhibition.

We agree that some observations in the previous version of our manuscript had a relatively small effect size, presumably due to partial knockouts of the targeted genes, particularly in cell lines of higher ploidy. We therefore repeated the experiments with CDC20 depletion using siRNA and shRNA, and our results show a significant phenotypic rescue in all cell lines (including HPT2) under reversine treatment. Please also see the new Fig. 3F-I:

Fig. 3F-I:

and HCT116

(F,G) Percent of EdU incorporating cells in HPT1 cells after CDC20 depletion with siRNA (F) or shRNA (G), with and without treatment with 250nM Reversine. Two-way ANOVA (N = 3, ****, p-value <0.0001) (H,I) Percent of EdU incorporating cells in HPT2 cells after CDC20 depletion with siRNA (H) or shRNA (I). Two-way ANOVA (N = 3, ***, p-value < 0.001; ***, p-value < 0.0001). CDC20 depletion in both aneuploid cell lines significantly reduced sensitivity to SAC inhibition.

Furthermore, as per the Reviewer's suggestion, we have tried to overexpress CDC20 using a plasmid previously described in Tsang et al (*Nature*, 2023), in which the authors successfully overexpressed two different CDC20 isoforms by Western blot analysis. However, despite following the same technical protocols, we failed to effectively overexpress the individual isoforms in the HCT116, HPT1, or HPT2 cell lines. This finding suggests that the plasmids may not demonstrate full functionality in these specific cell lines, which could be due to variations in cellular context or plasmid compatibility.

To address this limitation, we cloned cDNAs for the CDC20 isoforms into vectors encoding an inducible overexpression system. Using these plasmids, we successfully attained overexpression of isoform 2 (a truncated CDC20 isoform) in HCT116, HPT1, and HPT2 cell lines for a period of up to 24 hours. However, expression levels rapidly decreased and returned to normal by 48-hours, possibly due to a feedback mechanism that restricts sustained overexpression (see panels E-G in the Figure below). Furthermore, we failed to overexpress isoform 1 altogether, even in 293T cells, suggesting its overexpression is highly toxic in the cell lines that we used. Given that our functional experiments such as time lapse imaging and proliferation assays require CDC20 expression to exceed beyond 24 hours, we unfortunately could not test the role of CDC20 overexpression in our study. We show our attempts to

overexpress CDC20 isoforms in various vectors in the Reviewer's Figure below (not shown in the manuscript):

Figure: Analysis of CDC20 Expression in Different Cell Lines Transduced with Plasmids and Lentiviral Systems.

(A) Representative western blot image of CDC20 expression in 293T cells transduced with plasmids from the study: 031 (empty vector), 039 (isoform 1), and 040 (isoform 2). Cell pellets were harvested after viral transduction, and CDC20 levels were assessed. (B) Western blot of CDC20 expression in HCT116 cells transfected with virus containing plasmids from the study: 031 (empty vector), 039 (isoform 1), and 040 (isoform 2). CDC20 protein expression was quantified post-transfection. (C) Western blot of CDC20 expression in HPT1 cells transfected with plasmids from the study: 031 (empty vector), 039 (isoform 1), and 040 (isoform 2). CDC20 expression levels were analyzed post-transfection. (D) Western blot of CDC20 expression in HPT2 cells following transfection with plasmids from the study: 031, 039, and 040. CDC20 protein levels were measured across replicates. (E) Western blot of CDC20 expression in HCT116 cells using the rTta-lenti inducible system. Cells were transfected with doxycycline-inducible plasmid constructs, and CDC20 expression was quantified after induction. (F) Western blot of CDC20 expression in HPT1 cells using the rTta-lenti inducible system. Following doxycycline induction, CDC20 protein levels were assessed. (G) Western blot of CDC20 expression in HPT2 cells using the rTta-lenti inducible system. CDC20 expression was induced with doxycycline and analyzed post-induction.

As detailed in our response to the next comment, we repeated the experiment requested by the Reviewer – making diploid cells aneuploid with acute short-term MPS1i indeed drove sensitivity to SACi. Please see our response to the next comment.

5) Figure 4 explores the link between aneuploidy and CDC20 by using previously derived "highly aneuploid" lines from HCT116 and RPE1 cells. These lines are not discussed or characterized in this manuscript, unless I missed that, but from reading the reference it looks like they are derived from the tetraploidization of RPE1 or HCT116 cells. If that is correct, then is it aneuploidy or tetraploidy that is associated with elevated CDC20 levels during mitosis? The sentence beginning on line 253 starts to make an assumption about previous aneuploid cell data given these results in tetraploid clones. To make these conclusions one must compare mitotic CDC20 levels in the earlier aneuploid cell lines instead, not in unrelated tetraploid clones. I think acutely inducing aneuploidy in RPE1 cells would be a useful comparison here as well.

We thank the Reviewer for this comment and apologize this was not clear. The Reviewer is correct that these are tetraploidized lines. We now explicitly mention this in the revised manuscript (for the HCT116/HPT and the RPE-RPT cells). Importantly, we conducted the experiments proposed by the Reviewer and induced aneuploidy in HCT116 cells via transient MPS1 inhibition (reversine treatment). Following aneuploidization, these cells indeed displayed increased expression of CDC20, which coincided with an increased number of mitotic aberrations. Conversely, reducing expression of CDC20 by shRNA/siRNA knockdown increased metaphase duration and significantly reduced mitotic aberrations in these cells, in full agreement with the results from the HCT/HPT system. Therefore, our findings are not unique to tetraploid cells but also occur in non-tetraploid aneuploid cells. Please see the new **Fig. 3J,K**, **Fig. 4K,L**, **5F**, and the Discussion section.

Fig. 3J:

(J) Bulk quantification of CDC20 in wild-type HCT116 and after following aneuploidy induction using 250nM Reversine. Two-sided (T-test N= 5; **, p-value < 0.01).

Fig. 3K:

K

(K) Percent of EdU incorporating cells in aneuploid HCT116 cells after CDC20 depletion with siRNA and treatment with 250nM Reversine. Two-way ANOVA (N = 3, *, p-value < 0.05). In these cells too, CDC20 depletion decreases the sensitivity to SAC inhibition

Fig. 4K,L:

(K,L) Aneuploid HCT116 cells treated with 125nM Reversine also exhibit significantly decreased mitotic aberrations after CDC20 depletion.

Fig. 5F:

(G) Metaphase duration in aneuploid HCT116 cells under control conditions or CDC20 depletion by siRNA. In all three cell lines CDC20 depletion led to a significant increase in metaphase duration. Two-sided t-test (N =3, ****, p-value < 0.0001).

6) The final figure in 4C and D shows how mitotic errors are modulated by perturbing CDC20 levels, but not APC4 levels, which are predictable results given the resistance shown earlier. However, the same criticism can be levelled here as before - is this due to the differential penetrance of APC4 and CDC20 depletion? The lack of a significant metaphase delay after APC4 depletion, in comparison to CDC20 depletion, suggests that it is. Otherwise, why does depletion of an APC subunit not cause a metaphase delay? Plenty of studies have shown that metaphase durations can be extended with APC subunit depletion.

We agree that depletion of APC/C subunits is expected to lead to a metaphase delay, and we hypothesize that the fact that we did not observe this for ANAPC4 was due to partial knockout of the gene. However, as explained under previous points, in discussion with the Editor we decided to remove the ANAPC4 results and focus this manuscript on the phenotype of CDC20 only.

Therefore, I believe the most likely interpretation is that CDC20 and APC loss can drive metaphase delays and MPS1 inhibitor resistance, but perhaps it is easier to reduce CDC20 to critically low levels with the chosen sgRNA/shRNAs sequences. I certainly do not agree with conclusion that: "These results indicate that CDC20 is the key player in the cellular response to SAC inhibition, and that it acts by altering metaphase duration and affecting the overall level of chromosomal instability induced by SAC inactivation."

As explained under previous points, we decided to mostly focus on CDC20 only in our revised manuscript and tone down our conclusions about the relative role of CDC20 compared to the APC/C. We do demonstrate that CDC20 expression is an important determinant of resistance to SAC inhibitors, as it came up more strongly than any other APC/C-related gene in our CRISPR

screen, and as it contributed the most to the APC/C gene expression signature previously shown to be associated to resistance. However, this does not contradict a role for the APC/C subunits themselves, which also came up as hits (albeit weaker) in our CRISPR screen. We have removed the comparisons to ANAPC4, clarified and refined the message, and focused the revised manuscript on CDC20. Please see throughout the text, Fig. 1-4, and the Results and Discussion sections.

It could be that CDC20 overexpression is a key driver of sensitivity in aneuploid cells, and this would be an important conclusion, but I feel that would require further validation.

To strengthen the association between CDC20 expression and sensitivity to SAC inhibition, we performed the following experiments:

- (1) As described in our response to Reviewer's comment #4 above, we've tried to overexpress CDC20 in our cells, but unfortunately we could not achieve maintained overexpression.
- (2) We knocked down CDC20 in aneuploid cells and found that it decreased the severity and frequency of mitotic aberrations in aneuploid cells treated with MPS1 inhibition (please see **Fig. 4G-L** above) while prolonging the duration of metaphase (**Figure 5D-F** above). We next tested whether the link between metaphase duration, rate of mitotic aberrations and sensitivity to MPS1 inhibition is indeed causal. For this, we synchronized the cells and quantified mitotic aberrations under two conditions:
 1. A direct release from a G2 arrest into mitosis using the CDK1 inhibitor RO-3306.
 2. A release from a G2 arrest using RO 3306, followed by a delay in metaphase through exposure to the proteasome inhibitor MG-132, which delays CyclinB1 degradation and thus metaphase progression.

We found that prolonging metaphase duration with MG132 indeed drastically reduced the rate of mitotic aberrations (please see **Fig. 5G**), corroborating the link between metaphase duration and the prevalence of mitotic aberrations. This result supports our hypothesis that CDC20 expression levels determine the sensitivity to MPS1i via its regulation of metaphase duration.

(3) We induced aneuploidy in HCT116 cells using reversine, quantified CDC20 levels, and found them to be increased. We then knocked down CDC20 in these cells and observed that this decreased their sensitivity to SAC inhibition (please see **Fig. 3K**), also arguing that CDC20 levels determine sensitivity to MPS1 inhibitors..

(4) Using data from the DepMap cell line database, we correlated the sensitivity to different SAC inhibitors with aneuploidy levels, and observed a significant association between high CDC20 expression and increased sensitivity to SAC inhibition (as we previously reported in Cohen-Sharir et al. *Nature* 2021). We then repeated the analysis while removing the effect of CDC20 mRNA expression as a linear covariate, and the association between aneuploidy and the response to SACi became insignificant (please see **Fig. 3B**). These results support our notion of CDC20 expression being a main determinant of sensitivity to SAC inhibition.

While we were unable to overexpress CDC20 in the cells, these above highlighted experiments and analyses in multiple cell lines point to a consistent correlation – and a causal relationship – between CDC20 expression and sensitivity to SAC inhibition.

Fig. 3B:

(B) Left - correlation between ploidy and the sensitivity to the MPS1 inhibitors MPI-0479605 (top) and AZ3164 (bottom). Right – same correlation with CDC20 removed as a covariate using a linear model. When removing the effect of CDC20 expression the trend becomes insignificant. Two-sided t-test (ns – p-value > 0.05, ***; p-value < 0.001; ****, p-value < 0.0001).

Fig. 3K:

(K) Percent of EdU incorporating cells in aneuploid HCT116 cells after CDC20 depletion with siRNA and treatment with 250nM Reversine. Two-way ANOVA (N = 3, *, p-value < 0.05). In these cells too, CDC20 depletion decreases the sensitivity to SAC inhibition.

Fig. 5G:

(G) Fold change of mitotic aberrations in HCT116-HPT cells that underwent a 105 minute metaphase vs cells that underwent a shorter 45 minute metaphase. To reach a 105 minute metaphase, cells were treated with 7.5nM RO-3306, released for 45 minutes, treated with 10uM MG-132 for 60 additional minutes and then fixed. To reach a 45 minute metaphase, cells were treated only with RO-3306. The rate of mitotic errors decreased significantly in the cells that have experienced longer metaphases, as can be seen by the fold change of mitotic aberrations. One sample t-test (N=3; ns, p-value>0.05).

Other minor concerns:

7) Given the crucial role of CDC20 in this manuscript, the introduction should point out the role of different CDC20 molecules in the SAC response; namely that CDC20 is a component of MCC, but that MCC then binds a second CDC20 molecules that is bound to APC to inhibit this preformed APC/CDC20 complex (Izawa and Pines, 2015). This model is important to consider when implicating CDC20 levels in the overall SAC response.

Many thanks for that suggestion, we have amended the Introduction to include this.

8) It is not clear to me how you can quantify CDC20 levels in cells with normal or aberrant divisions from fixed analysis? The legends are not clear but how does one define aberrant division from a fixed population of prometaphase cells?

We strengthened this analysis by the following additional experiments:

(1) We synchronized the cells to G2/M using nocodazole, and performed Western Blots on the bulk cell populations to measure CDC20 levels – Please see **Fig. 3D** and **Supplementary Fig. 3D**.

(2) We specifically synchronized the cells to metaphase by arresting them at G2/M using the CDK1 inhibitor RO-3306, then releasing them in two different conditions: (A) Divide for 45 min with no further treatment (this is the time when most of the cells arrive at metaphase, visually); or (B) Release from arrest for 45min and then treating them with a proteasome inhibitor for an additional 60min, thereby maximizing the enrichment of cells in metaphase. CDC20 was then quantified on the single cell level, repeating the previous experiments in higher magnification (100x + z-stacking). Please see **Fig. 4A,B**.

All of the results support our original conclusion that CDC20 expression levels are higher in erroneous metaphases than in normal metaphases.

Fig. 3D:

(D) Bulk quantification of CDC20 protein expression in the HCT116-HPT system during mitosis after synchronization with 330nM Nocodazole. Here as well, the highly aneuploid cells express higher CDC20 protein levels than their diploid counterparts. One-sample t-test (N = 3, *, p-value < 0.05; **, p-value < 0.01).

Supplementary Fig. 3D:

D

(D) Representative western blot image of bulk CDC20 expression in the HCT116-HPT cell line set following synchronization with 330nM Nocodazole. Aneuploid cells express higher levels of CDC20 than their diploid counterparts.

Figure 4A,B:

(A,B) Representative immunofluorescence images **(A)** and quantification **(B)** of CDC20 protein levels during normal and aberrant metaphases, in single HCT116 cells and their highly-aneuploid derivatives after synchronization with 7.5uM RO-3306. Regardless of ploidy background, cells with mitotic aberrations express significantly higher levels of CDC20 during metaphase than cells undergoing normal division. One-sample t-test (N=3, *, p-value <0.05; **, p-value <0.01).

9) Line 381 "Effectively, SAC inhibition and CDC20 overexpression result in a similar outcome - they both lead to higher effective concentrations of uninhibited APC/C-CDC20 complexes in the cells, allowing cell division to continue even in the presence of malformed spindles."

I cannot see the evidence that CDC20 overexpression leads to higher concentration of uninhibited APC/C-CDC20 complexes? Such mechanistic studies on CDC20 overexpression alone would be valuable.

We agree that these studies would be valuable, but they are beyond the scope of the current study, especially as we now focus our manuscript mainly on CDC20. We therefore rephrased the sentence – please see lines 452-456.

10) Line 397: "these findings are highly clinically relevant as several SAC inhibitors are currently in clinical trials, with one of them being fast-tracked by the FDA (NCT05251714)."

This refers to a phase I and phase II trial that is estimated to complete in 2 years. I cannot find the evidence that the inhibitors are being fast-tracked by the FDA?

Indeed, the MPS1 inhibitor CFI-402257 has been fast-tracked by the FDA, according to this news report:

<https://www.targetedonc.com/view/fda-grants-fast-track-designation-to-novel-ttk1-therapy-for-er-her2--advanced-breast-cancer>

11) In relation to 1D and S1G, how were these assays performed? Legends do not provide details - relative viability after how long of each treatment? Also, what does CDC20 knockout/knockdown do to mitotic duration in the absence of reversine treatment?

We apologize that this was not clear. The cell lines in **Fig. 1D** and **Supplementary Fig. 1E** were seeded and treated with doxycycline on Day 1 to knockdown CDC20. On Day 2, the culture was supplemented with 250nM reversine, and the cells were allowed to grow until Day 6, when they were harvested for the crystal violet assay.

In the absence of reversine treatment, the knockdown of CDC20 either by siRNA or shRNA resulted in an extended duration of metaphase (evaluated as the interval from nuclear envelope breakdown to the onset of anaphase) in 3T3, HCT116, HPT1, HPT2, and aneuploid-HCT116 cells – please see **Fig. 5A-F**. This is now explained in more detail in the Figure legend – please see **Fig. 5**.

Fig. 5A-F:

(A-D) Metaphase duration in mouse 3T3 cells **(A)** or human HCT116 cells **(B,C)** under control conditions or CDC20 depletion by siRNA **(B)** or shRNA **(A,C)**. CDC20 depletion in both cell lines resulted in an increased metaphase duration. Two-sided t-test or One-way ANOVA (N =3, ***, p-value<0.001; ****, p-value < 0.0001). **(D,E,F)** Metaphase duration in the aneuploid HPT1 **(D)**, HPT2 **(E)** or the third HCT116 aneuploid derivative **(F)** under control conditions or CDC20 depletion by siRNA. In all three cell lines CDC20 depletion led to a significant increase in metaphase duration. Two-sided t-test (N =3, ****, p-value < 0.0001).

Dear Uri,

Thank you for the submission of your revised manuscript. As we discussed in person, the two referees of EMBOR-2024-60377-T still have suggestions that I would like to politely request you incorporate into a final revision. The two referee reports and the cross-commenting of ref 1 and ref 2's report is attached below.

For the avoidance of doubt, we will not require a live cell imaging experiment, but encourage you to explain the orthogonal approaches and the technical limitations of sub-M-phase assignment of the current data.

Furtermore, we appreciate the attempts at the functional OS data, but understand the technical challenging nature of these experiment and the limited precedence in the literature. We encourage discussion of this in the paper and display of the data as an 'EV figure' (expanded view figure) in the paper.

Finally, please revisit the text carefully to remove all claims of causality with respect to elevated CDC20 beyond a scholarly discussion of the point in context of the limitations of the current data and other suggestive evidence in the discussion section (see comments of both referees).

I look forward to seeing a new revised version of your manuscript as soon as possible.

It was wonderful to have the opportunity to discuss the revision directly today,

Bernd

~~~~~  
Bernd Pulverer, Ph.D.  
Chief Editor, EMBO Reports  
EMBO

Meyerhofstrasse 1, D-69117 Heidelberg  
Tel: +4962218891501  
bernd.pulverer@embo.org  
~~~~~

Referee #1:

In my opinion, the manuscript is now clearer and less ambiguous with the direct focus on CDC20 instead of differences to APC4. I feel the experiments are performed to a high standard and they make important conclusion that build on previous relevant work on sensitivity to MPS1 inhibitors.

It is a shame that the CDC20 overexpression experiments did not work and I feel that this data should be mentioned very briefly and included in supplementary. This would explain why these key experiments are omitted and it would also help people who may want to try similar experiments in future. Other than this I have no further comments and I congratulate the authors on a nice study.

Referee #2:

The revised manuscript from Zheng et al is substantially improved. There remain a couple of claims that are not sufficiently supported by the data and several minor issues to address.

1. The conclusion that cells undergoing aberrant mitosis exhibit higher levels of CDC20 during metaphase (lines 328-330) is not justified by the current evidence. Fig 4A-B and Sup Fig 5A-B are not sufficient because they are fixed analysis and it is not possible to discriminate whether cells would align their chromosomes before proceeding or missegregate them in fixed cells.

The live cell analysis in Fig 4C-K and Sup Fig 5C-D is also not sufficient to substantiate this claim, since it does not reveal Cdc20 expression in individual cells undergoing various levels of missegregation. This conclusion must be toned down or substantiated, for instance with timelapse analysis of cells with fluorescence chromosomes that also contain Cdc20 alleles that are homozygously tagged with a fluorophore.

Similarly, while I appreciate that the authors attempted to overexpress Cdc20, since they were unable to do so, the conclusion on lines 445-446 about CDC20 overexpression should be tempered.

2. Fig 5G does not show that mitotic aberrations significantly decreased when metaphase duration was extended, as claimed (line 399). Presumably, the cells with the so-called 45 minute metaphase were released from the CDK1 inhibitor for 45 minutes, though the figure legend does not state this. It would be more appropriate to compare the percentage of cells with complete metaphase alignment +/- 60 min treatment with MG-132, rather than to show the fold change.

Minor comments

3. In the legend for Fig 5, please specify how many cells were quantitated per movie and how the data are represented. It would be helpful if cells/averages from independent movies were plotted in different colors so that the variability between movies was apparent.

4. The term "metaphase skipping" is not standard. Is it intended to mean that the cells proceed into anaphase without achieving full chromosome alignment at the metaphase plate? How is this different than metaphase misalignment (when alignment is not achieved prior to anaphase onset)?

5. On line 70, the list of MCC proteins is incorrect. BUBR1 is listed twice (once by its protein name and once by its gene name BUB1B), while BUB3 is omitted. MCC contains proteins encoded by the genes BUB1B, MAD2L1, BUB3 and CDC20.

6. The blue in Fig 1E is very difficult to see. Grayscale single channels for DAPI and EdU would be much easier to visualize. For the overlay, green+red, cyan+red, or green+purple are all preferable to blue+red.

7. Sup Fig 1E would benefit from inclusion of quantitation of the blot (which is shown in the reply to review).

8. To establish the biological importance of the relatively modest correlation in Fig 2A-B, it would be helpful to add the data showing the correlations between CCNE1 and CDK2i, TOP2A and TOP2Ai, etc, (as is shown in the reply to review) to a supplemental figure.

9. Supplementary Table 2 is referenced on line 161, but I could not find this table.

10. Line 248 should be corrected to indicate that nocodazole synchronizes cell in prometaphase, rather than at the G2/M border.

11. The flow cytometry profiles after 20 hr nocodazole treatment in Fig S3 are difficult to interpret without inclusion of profiles from asynchronously cycling cells.

12. Though the data regarding Apc4 and the associated claims about the predictive value of Cdc20 versus core Apc components were largely removed, a claim that CDC20 expression predicts drug sensitivity better than APC/C subunits remains on line 438-439, and another about APC/C subunits not being overexpressed in aneuploid cells remains on lines 456-457. These should be removed.

.....

Referee #1: Cross-commenting on ref #2:

I completely agree with comment 1 and the need to tone down statement related to the causal relationship between CDC20 expression and the degree of CIN
e.g. line 343 "Together, these findings demonstrate a causal relationship between CDC20 expression and the degree of CIN induced by SAC inhibition"

The author have shown very clearly that reducing CDC20 can recover CIN, and this is in line with previous observation on APC inhibition, however the link to overexpression of CDC20 is entirely correlative. It is consistent with the interpretation that high CDC20 induces errors, but without overexpressing CDC20 and demonstrating a causal relationship then these statements need to be toned down. I also agree that fixed analysis is not ideal from Fig4a.b, but I think the suggestion to do this with a fluorescent CDC20 knock-in is not really feasible because I think that tagging the C or N-terminus of CG20 is not possible without affecting its function.

I am also confused by experiment in 5G. The graph and legend are actually very confusing as to how the experiment was performed. I agree that fold change is a strange way to show this and this simply hides data - I would just show the percentage

of cells with/without metaphase alignment as suggested by the reviewer.

In summary, I would agree with reviewer 2 that 1 major key point remain. This issue, of course, is that the really solid data in this manuscript all relates to CDC20 depletion, which is the least novel aspect of the work. The overexpression and potential link to CIN at the most novel part, and the interoperations do really need to be toned down about this without the key overexpression experiment. Personally, if it was me, I would try harder to get that to work given the importance (for example, Inducing expression with dCas9-VPR instead of plasmids).

.....

Response to Reviewers

Referee #1:

In my opinion, the manuscript is now clearer and less ambiguous with the direct focus on CDC20 instead of differences to APC4. I feel the experiments are performed to a high standard and they make important conclusion that build on previous relevant work on sensitivity to MPS1 inhibitors.

We thank the Reviewer for this positive assessment.

It is a shame that the CDC20 overexpression experiments did not work and I feel that this data should be mentioned very briefly and included in supplementary. This would explain why these key experiments are omitted and it would also help people who may want to try similar experiments in future.

We agree with the Reviewer, and we now include these negative results and in the Supplementary data and refer to them from the text. Please see **Supplementary Figure 6 A-G**:

Supplementary figure 6. CDC20 overexpression attempts in HCT116-HPT cells

(A) Western blot quantification of CDC20 overexpression with plasmids from Tsang et al 2023, in 293T cells. CDC20 is overexpressed in these cells. (B-D) Western blot quantification of CDC20 overexpression with plasmids from Tsang et al 2023, in HCT116 (B), HPT1 (C) and HPT2 (D) cells. CDC20 is not overexpressed in

these cells. **(E-G)** Western blot quantification of CDC20 overexpression with lentiviral rTTA inducible system, in HCT116 **(E)**, HPT1 **(F)** and HPT2 **(G)** cells. CDC20 is not overexpressed in this system either.

Other than this I have no further comments and I congratulate the authors on a nice study.

Thank you.

Referee #2:

The revised manuscript from Zheng et al is substantially improved. There remain a couple of claims that are not sufficiently supported by the data and several minor issues to address.

1. The conclusion that cells undergoing aberrant mitosis exhibit higher levels of CDC20 during metaphase (lines 328-330) is not justified by the current evidence. Fig 4A-B and Sup Fig 5A-B are not sufficient because they are fixed analysis and it is not possible to discriminate whether cells would align their chromosomes before proceeding or missegregate them in fixed cells. The live cell analysis in Fig 4C-K and Sup Fig 5C-D is also not sufficient to substantiate this claim, since it does not reveal Cdc20 expression in individual cells undergoing various levels of missegregation. This conclusion must be toned down or substantiated, for instance with time lapse analysis of cells with fluorescence chromosomes that also contain Cdc20 alleles that are homozygously tagged with a fluorophore.

Following the original comments of the Reviewer during the first round of review, we performed three experiments that provide strong support to the association between mitotic aberrations and elevated expression of CDC20: (a) Higher-resolution microscopy to ensure that our image analysis indeed captured cells in metaphase; (b) Western-blot of cells following cell cycle synchronization and their following release into mitosis; and (c) Treatment with a proteasome inhibitor, MG-132, to synchronize the cells in mitosis. All of these experiments support our original conclusion that CDC20 is more highly expressed in cells undergoing aberrant mitosis.

We note that fixed analyses, despite being suboptimal, are considered a common way to estimate mitotic aberrations in the literature – whereas in some cases they may be eventually resolved, as suggested by the Reviewer, there is little doubt that these cells would at least be strongly enriched for aberrations that are not resolved and result in real mitotic aberrations.

We agree that a live-cell imaging experiment would be definitive, but – as Reviewer #1 mentions below – fluorescent CDC20 knock-in is not really feasible because tagging

CDC20 without affecting its function is extremely challenging (if at all possible). We now discuss our orthogonal approaches and their technical limitations, and the experiment proposed by the Reviewer in the Discussion – **Please see the third paragraph of the Discussion (Lines 471-484).** The experiments that support this notion are described in Figure 3C, Figure 4A,B, Figure 3D, and Supplementary Figure 3D:

Figure 3C:

Figure 4A,B:

Figure 3D:

Supplementary Figure 3D:

D

Similarly, while I appreciate that the authors attempted to overexpress Cdc20, since they were unable to do so, the conclusion on lines 445-446 about CDC20 overexpression should be tempered.

We removed any claims for causality between mitotic aberrations and CDC20 expression levels, and we only discuss the strong association between the two. We also discuss and present the negative results from the OE experiment, as was also requested by Reviewer #1. **Please see Lines 496-505 and Supplementary Figure 6 A-G:**

Supplementary figure 6. CDC20 overexpression attempts in HCT116-HPT cells

(A) Western blot quantification of CDC20 overexpression with plasmids from Tsang et al 2023, in 293T cells. CDC20 is overexpressed in these cells. (B-D) Western blot quantification of CDC20 overexpression with plasmids from Tsang et al 2023, in HCT116 (B), HPT1 (C) and HPT2 (D) cells. CDC20 is not overexpressed in these cells. (E-G) Western blot quantification of CDC20 overexpression with lentiviral rTTA inducible system, in HCT116 (E), HPT1 (F) and HPT2 (G) cells. CDC20 is not overexpressed in this system either.

2. Fig 5G does not show that mitotic aberrations significantly decreased when metaphase duration was extended, as claimed (line 399). Presumably, the cells with the so-called 45 minute metaphase were released from the CDK1 inhibitor for 45 minutes, though the figure legend does not state this. It would be more appropriate to compare the percentage of cells with complete metaphase alignment +/- 60 min treatment with MG-132, rather than to show the fold change.

We apologize for the ambiguity in the legend to Fig. 5G – we have rephrased it, and we believe that it is now clearer. We have also added the requested comparison of the percentage of cells with normal vs. aberrant metaphase +/- 60 min of treatment, in addition to showing the fold change. Please see Figure 5G and the new Supplementary Figure 6H.

Figure 5G:

Supplementary Figure 6H:

Minor comments:

3. In the legend for Fig 5, please specify how many cells were quantitated per movie and how the data are represented. It would be helpful if cells/averages from independent movies were plotted in different colors so that the variability between movies was apparent.

This information has been added to the Figure and Legend, by separating the data between two biological experiments and adding the number of cells analysed for each experiment, as requested.

4. The term "metaphase skipping" is not standard. Is it intended to mean that the cells proceed into anaphase without achieving full chromosome alignment at the metaphase plate? How is this different than metaphase misalignment (when alignment is not achieved prior to anaphase onset)?

We use the term “metaphase skipping” when the cells fail to form any kind of metaphase plate, whereas “metaphase misalignment” is milder. We now explain this in the Methods section, and exemplify it with a movie – Please see lines 755-760 and the new Supplementary Movie 17.

5. On line 70, the list of MCC proteins is incorrect. BUBR1 is listed twice (once by its protein name and once by its gene name BUB1B), while BUB3 is omitted. MCC contains proteins encoded by the genes BUB1B, MAD2L1, BUB3 and CDC20.

We apologize for this mistake; it has been corrected. **Please see Line 70.**

6. The blue in Fig 1E is very difficult to see. Grayscale single channels for DAPI and EdU would be much easier to visualize. For the overlay, green+red, cyan+red, or green+purple are all preferable to blue+red.

We have changed the colors as requested – **Please see the revised Fig. 1E:**

7. Sup Fig 1E would benefit from inclusion of quantitation of the blot (which is shown in the reply to review).

We have added the quantification as requested. **Please see the revised Supplementary Fig. 1E:**

8. To establish the biological importance of the relatively modest correlation in Fig 2A-B, it would be helpful to add the data showing the correlations between CCNE1 and CDK2i, TOP2A and TOP2Ai, etc, (as is shown in the reply to review) to a supplemental figure.

These data were actually shown in Supplementary Table 2 of Cohen-Sharir et al. (Nature 2021). We have added a sentence to highlight this point, referring to the Cohen-Sharir et al. paper. **Please see Lines 170-175.**

9. Supplementary Table 2 is referenced on line 161, but I could not find this table.

We apologize for this omission, Table 2 describes the list of genes that belong to the previously reported APC/C signature – it is now provided.

10. Line 248 should be corrected to indicate that nocodazole synchronizes cell in prometaphase, rather than at the G2/M border.

This sentence has been corrected. **Please see Line 258.**

11. The flow cytometry profiles after 20 hr nocodazole treatment in Fig S3 are difficult to interpret without inclusion of profiles from asynchronously cycling cells.

These profiles have been added – **Please see Supplementary Figure 3D:**

D

12. Though the data regarding Apc4 and the associated claims about the predictive value of Cdc20 versus core Apc components were largely removed, a claim that CDC20 expression predicts drug sensitivity better than APC/C subunits remains on line 438-439, and another about APC/C subunits not being overexpressed in aneuploid cells remains on lines 456-457. These should be removed.

We respectfully disagree. These statements accurately represent the findings of our DepMap analyses and CRISPR screen. They are made in the Discussion as part of a

scholarly Discussion of the topic, and they are not in contradiction with previous literature that showed APC/C relevance. Also please note that we define CDC20 as a key driver of the phenotype, not the key driver. Altogether, our data highlight the importance of CDC20 as a potential biomarker for SACi sensitivity, whereas previous literature largely focused on the APC/C subunits, and so we believe that mentioning this in the Discussion is important.

.....

Referee #1: Cross-commenting on ref #2:

I completely agree with comment 1 and the need to tone down statement related to the causal relationship between CDC20 expression and the degree of CIN e.g. line 343 "Together, these findings demonstrate a causal relationship between CDC20 expression and the degree of CIN induced by SAC inhibition".

The authors have shown very clearly that reducing CDC20 can recover CIN, and this is in line with previous observation on APC inhibition, however the link to overexpression of CDC20 is entirely correlative. It is consistent with the interpretation that high CDC20 induces errors, but without overexpressing CDC20 and demonstrating a causal relationship then these statements need to be toned down.

We have removed all claims for causality and now explicitly explain that while we find a strong association between mitotic errors and CDC20 expression, we were unable to perform the overexpression experiment which is required to conclusively demonstrate causality. **Please see lines 495-505.**

I also agree that fixed analysis is not ideal from Fig4a.b, but I think the suggestion to do this with a fluorescent CDC20 knock-in is not really feasible because I think that tagging the C or N-terminus of CC20 is not possible without affecting its function.

We agree with this statement. We now discuss our orthogonal approaches to demonstrate the association between CDC20 expression and mitotic aberrations, and we present their technical limitations, as well as the genetic labeling experiment (which we agree that is not really feasible). **Please see the third paragraph of the Discussion (Lines 471-484).**

I am also confused by the experiment in 5G. The graph and legend are actually very confusing as to how the experiment was performed. I agree that fold change is a strange way to show this and this simply hides data - I would just show the percentage of cells with/without metaphase alignment as suggested by the reviewer.

We have also added the requested comparison of the percentage of cells with normal vs. aberrant metaphase +/- 60 min of treatment, in addition to showing the fold change. Please see Figure 5G and the new Supplementary Figure 6H.

Figure 5G:

Supplementary Figure 6H:

In summary, I would agree with reviewer 2 that 1 major key point remain. This issue, of course, is that the really solid data in this manuscript all relates to CDC20 depletion, which is the least novel aspect of the work. The overexpression and potential link to CIN at the most novel part, and the interoperations do really need to be toned down about this without the key overexpression experiment. Personally, if it was me, I would try harder to get that to work given the importance (for example, Inducing expression with dCas9-VPR instead of plasmids).

We now present and discuss the data from our overexpression experiment (as detailed above). We agree that it would have been very beneficial to include successful results from this experiment, but unfortunately we spent many months doing it without success. We would like to note, however, that the solid results that highlight CDC20 as

a potential biomarker of SACi, albeit not being surprising, are still novel and important, as previous literature heavily focused on the APC/C subunits and not on CDC20 itself. Further, while we were unable to show a causal relationship between mitotic aberrations and CDC20 OE, to the best of our knowledge we are the first to show a strong correlative association between the two, so this aspect of the manuscript is also novel (and quite important from a biomarker standpoint).

Prof. Uri Ben-David
Tel Aviv University
Human Molecular Genetics & Biochemistry
Faculty of Medicine building
Room 728
Tel Aviv, NA 69978
Israel

Dear Uri,

I am pleased to inform you that your manuscript has been accepted for publication in EMBO reports, subject to a minor revision to remedy the issues listed below.

We appreciate the discussion and careful revision to ensure the claims accurately reflect the data provided at this time. In particular, the changed conclusion from 'causality' to 'strong association' between mitotic errors and CDC20 expression. Your manuscript will be processed for publication by EMBO Press as soon as we have the final version which resolves these issues. I am sorry for my slow response due to travel, but please note that if a revision can be supplied in the next couple of days, there is a good chance of publication this year. This chance will decrease exponentially over the next couple of weeks.

The manuscript will be copy edited and you will receive page proofs prior to publication. Please note that you will be contacted by Springer Nature Author Services to complete licensing and payment information. (see <https://www.embopress.org/page/journal/14693178/authorguide#chargesguide>)

Best wishes and thank you for selecting EMBO for your paper,

Bernd

.....

Issues which require further revision prior to publication:

MANUSCRIPT TEXT

- Line 343 states 'These results suggest a link between increased CDC20 expression and chromosome missegregation, which is both a cause and a consequence of the aneuploid state'. >> we appreciate that you changed this sentence substantially and understand the use of 'a' rather than 'the'. However, we recommend to introduce a modifier here to ensure it is clear that causality is plausible and even likely, but not demonstrated experimentally. Thus, we recommend 'These results suggest a likely link between increased CDC20 expression and chromosome missegregation, which may be both a cause and a consequence of the aneuploid state'.

- line 419: pls remove 'All in all,..'

- Please reduce keywords to up to 5
- Main figure legends need to be placed after the References in the manuscript file and then EV figure legends should follow
- Materials and Methods should be Methods
- Dataset availability needs to be renamed to Data Availability; the section should go before Acknowledgments
- Conflicts of interest should be renamed to Disclosure Statement and Competing Interests
- Author contributions need to be removed from the manuscript

FIGURES

- Main and EV need to be removed from the manuscript as we already have the individual production quality Figure files; the legends need to stay in the manuscript
- Nomenclature for EV figures is not correct - instead of Supplementary Fig 1 etc, it should be Figure EV1, etc. - this needs to be corrected in the source file names, individual Figure files and figure legends in the manuscript file
- SI figs 6A-E: Please note that isoforms 2, 3 are induced in 293T cells, for clarity. Please add MW marker(s). Please center +/- signs above the panel and the quantifications. It may be more accurate to state 'No CDC20 overexpression can be detected in these cells'. 'Isoform 3' is not described in the text - please explain the nature of all the isoforms in more detail.
- fig 1E needs scale bars.
- Please include details on size markers in all relevant figure legends e.g. fig 4A, 3E, 1E.

TABLES

- Supplementary Tables 1 is a dataset and needs to be renamed (source file name, title in eJP and ms callout) to Dataset EV1 while the legend needs to be removed from the ms and inserted as a separate sheet/tab in Excel
- Supplementary Table 2 should be renamed to Table EV2 (source file name, title in eJP and ms callout) while the legend needs to be removed from the ms
- Supplementary Table 3 should be renamed to Table EV3 (source file name, title in eJP and ms callout) and the legend should be removed from the ms

MOVIES

- Movie legends need to be removed from the ms file and each should be provided in a readme.txt file and then zipped together with its movie file; the correct nomenclature for the movies (source file names, title in eJP, legends and ms callouts) should be Movie EV1-EV17

SOURCE DATA

- Our source data coordinator has checked your uploaded source data and ticked off everything that was provided in the source data checklist that was added to your submission; The only thing missing are the microscopy images are missing - could you please upload them? If they are deposited in an external repository, we will need the specific URL provided in the data availability section of your manuscript file
- Please also re-group the source data uploaded to our system: source data files need to be submitted as zipped folders, one .zip folder for each figure. Inside each folder, the files should be organized in subfolders, one subfolder for each panel.

OTHER

- The following author's name needs to be corrected as there is a discrepancy: Marica Rosaria Ippolito in the manuscript file vs. Marica Ippolito in the system; if the system needs to be updated, we can correct the name on our end
- Data Availability: we need direct URL for the PRJEB71335 dataset; the dataset should be publicly available latest at the time of publication
- Funding Information: not congruent; the following are missing in eJP and need to be added separately via More funders option, not just copied in the Comments box: the Chinese Scholar Council (CSC), the Israel Cancer Research Fund Project Award, the Azrieli Foundation Faculty Fellowship, BSF Project Grant (grant #2019228), Bridge Grant 2023 - ID. 29228, Fondazione Cariplo, the Rita-Levi Montalcini program from MIUR, supported by an AIRC Fellowship (ID 26738-2021)
- Papers published in EMBO Reports include a 'synopsis' and 'bullet points' to further enhance discoverability. Both are displayed on the html version of the paper and are freely accessible to all readers. The synopsis includes a short standfirst summarizing the study in 1 or 2 sentences (max 35 words) that summarize the paper and are provided by the authors and streamlined by the handling editor. I would therefore ask you to include your synopsis blurb and 3-5 bullet points listing the key experimental findings.
- In addition, please provide an image for the synopsis. This image should provide a rapid overview of the question addressed in the study but still needs to be kept fairly modest since the image size cannot exceed 550 (width) x 300-600 (height) pixels.

ISSUES RAISED BY DATA EDITORS

- Please note that the legend for figure 4c-l is mislabeled as figure 4c-n in the manuscript. This needs to be rectified.
- Please note that the exact p values are not provided in the legends of figures 1d; 2b, c-d, f-g; 3a-b, d, f-k; 4b, d, f, h, j, l, supplementary figures 1d; 2a, c; 3a-c; 4b, f, h; 5a-b, f; 6a.
- Please indicate the statistical test used for data analysis in the legends of figures 1c, supplementary figure 1c.
- Please note that in figures 3b, d; 4d, f; there is a mismatch between the annotated p values in the figure legend and the annotated p values in the figure file that should be corrected.
- Please note that the box plots need to be defined in terms of minima, maxima, centre, bounds of box and whiskers, and percentile in the legends of figures 3a-b, d, j; 5a-f, supplementary figures 3a-b; 5a-b.
- Please note that information related to n is missing in the legends of figures 4d, f, h, j, l, supplementary figure 5d.
- Although 'n' is provided, please describe the nature of entity for 'n' in the legends of figures xxx, supplementary figures 3f; 4f, h.
- Please note that the error bars are not defined in the legends of figures 1d; 2f-g; 3d,f-i, k; 4b; 5g, supplementary figures 1d; 2a, c; 3c; 4b, f, h.
- Please note that scale bar and its definition are missing for figures 1e; 3c, e; 4a.

~~~~~  
Bernd Pulverer, Ph.D.  
Chief Editor, EMBO Reports  
EMBO  
Meyerohofstrasse 1, D-69117 Heidelberg  
Tel: +4962218891501  
bernd.pulverer@embo.org  
~~~~~
